# Hybrid Reinforcement Learning Breaks Sample Size Barriers in Linear MDPs

**Kevin Tan,  Wei Fan,  Yuting Wei**
Department of Statistics and Data Science
The Wharton School, University of Pennsylvania

## Abstract

Hybrid Reinforcement Learning (RL), where an agent learns from both an offline dataset and online explorations in an unknown environment, has garnered significant recent interest. A crucial question posed by Xie et al. (2022b) is whether hybrid RL can improve upon the existing lower bounds established for purely offline or online RL without requiring that the behavior policy visit every state and action the optimal policy does. While Li et al. (2023b) provided an affirmative answer for tabular PAC RL, the question remains unsettled for both the regret-minimizing and non-tabular cases. In this work, building upon recent advancements in offline RL and reward-agnostic exploration, we develop computationally efficient algorithms for both PAC and regret-minimizing RL with linear function approximation, without requiring concentrability on the entire state-action space. We demonstrate that these algorithms achieve sharper error or regret bounds that are no worse than, and can improve on, the optimal sample complexity in offline RL (the first algorithm, for PAC RL) and online RL (the second algorithm, for regret-minimizing RL) in linear Markov decision processes (MDPs), regardless of the quality of the behavior policy. To our knowledge, this work establishes the tightest theoretical guarantees currently available for hybrid RL in linear MDPs.

## 1  Introduction

Reinforcement learning (RL) holds great promise in attaining reliable decision-making in adaptive environments for a broad range of modern applications. Typical RL algorithms often require an enormous number of training samples, motivating a line of recent efforts to study the sample efficiency of RL algorithms. There are two mainstream paradigms of RL, distinguished by how samples are collected: online RL and offline RL. In online RL, an agent learns in a real-time manner, exploring the environment to maximize her cumulative rewards by executing a sequence of adaptively chosen policies (e.g. Azar et al. (2017); Jin et al. (2018); Sutton and Barto (2018); Zhang et al. (2023)). Whereas, in offline RL, an agent has only access to a pre-collected dataset, and tries to figure out how to perform well in a different environment without ever experiencing it (e.g. Jin et al. (2021b); Lange et al. (2012); Levine et al. (2020); Li et al. (2024)). Online methods are often sample-hungry, but offline methods often impose stringent requirements on the quality of the pre-collected data.

To address the limitations of both, the setting of hybrid RL (Song et al., 2023; Xie et al., 2022b) has recently received considerable attention from both theoretical and practical perspectives (Amortila et al., 2024; Ball et al., 2023; Kausik et al., 2024; Li et al., 2023b; Nair et al., 2020; Nakamoto et al., 2023; Song et al., 2023; Tan and Xu, 2024; Vecerik et al., 2017; Wagenmaker and Pacchiano, 2023; Zhou et al., 2023). In hybrid RL, an agent learns from a combination of both offline and online data, extracting information from offline data to enhance online exploration. Theoretical guarantees for hybrid RL algorithms can be categorized on: (1) the type of function approximation considered, (2) the level of coverage required by the behavior policy, (3) whether it improves on the *minimax lower bounds* for online-only and offline-only learning, and (4) whether they minimize regret or obtain a PAC guarantee. We elaborate below, and summarize the prior art in Table 1.

38th Conference on Neural Information Processing Systems (NeurIPS 2024).

| Paper | Function Type | Concentrability? | Improvement? | Regret or PAC? |
|---|---|---|---|---|
| Song et al. (2023) Nakamoto et al. (2023) | General | Required | No | Regret |
| Tan and Xu (2024) Amortila et al. (2024) | General | Not Required | No | Regret |
| Wagenmaker and Pacchiano (2023) | Linear | Not Required | No | PAC |
| Li et al. (2023b) | Tabular | Not Required | Yes | PAC |
| This work | Linear | Not Required | Yes | Regret, PAC |

Table 1: Comparison of our contributions to previous work in hybrid RL.

While much of the prior literature (Amortila et al., 2024; Nakamoto et al., 2023; Song et al., 2023; Tan and Xu, 2024; Zhou et al., 2023) tackles general function approximation in hybrid RL, they either require stringent concentrability assumptions on behavior policy quality, or fail to obtain tight theoretical guarantees. Under such single-policy concentrability assumptions (explained below), Xie et al. (2022b) show the optimal RL algorithm is either a purely offline reduction or a purely online algorithm if the agent can choose the ratio of offline to online samples, rendering the benefits of hybrid RL questionable. Without this assumption, Li et al. (2023b) show guarantees for PAC RL that improve over lower bounds for offline-only and online-only RL, but only for tabular MDPs.

This paper focuses on obtaining sharper theoretical guarantees in the setting of linear function approximation in *linear MDPs*. First proposed in Jin et al. (2019); Yang and Wang (2019), linear MDPs parameterize the transition probability kernel and reward function by linear functions of known features (e.g. pre-trained neural embeddings). It has been extensively studied due to its benefits in dimension reduction and mathematical tractability in both the online and offline settings (Du et al., 2019; Duan and Wang, 2020; He et al., 2023; Hu et al., 2023; Jin et al., 2019; Li et al., 2021; Min et al., 2021; Qiao and Wang, 2022; Xiong et al., 2023; Yang and Wang, 2019; Yin et al., 2022; Zanette et al., 2021). Despite these efforts, hybrid RL algorithms for linear MDPs (Amortila et al., 2024; Nakamoto et al., 2023; Song et al., 2023; Tan and Xu, 2024; Wagenmaker and Pacchiano, 2023) have suboptimal worst-case guarantees (Table 2), which raises the question:

*Is it possible to develop sample efficient RL algorithms in the setting of hybrid RL that are provably better than online-only and offline-only algorithms for linear MDPs?*

## 1.1 Hybrid RL: two approaches

To answer the question above, we introduce two types of approaches widely-adopted in hybrid RL.

**The offline-to-online approach:** Most of the current literature (e.g. Amortila et al. (2024); Nakamoto et al. (2023); Song et al. (2023); Tan and Xu (2024)) initializes the online dataset with offline samples to perform regret-minimizing online RL. We refer to this as the *offline-to-online* approach. This method is simple and natural, and as the algorithm optimizes the reward during each online episode, it is suitable when the agent has to perform well during online exploration.

**The online-to-offline approach:** However, if our goal is to output a near-optimal policy, especially in real-world situations in medicine and defense, randomizing between policies can be suboptimal or even unethical. Recently, Wagenmaker and Pacchiano (2023) and Li et al. (2023b) propose using reward-agnostic online exploration to explore parts of the state space unseen by the behavior policy, to construct a dataset that is especially amenable to leverage the sharp performance guarantees of offline RL. We refer to this as the *online-to-offline approach*. While this approach does not optimize the "true reward" during online exploration, it avoids the need to deploy mixed policies to achieve a PAC bound, allowing for the deployment of fixed, and thus more interpretable, policies.

## 1.2 Our contributions

- We propose an online-to-offline method called *Reward-Agnostic Pessimistic PAC Exploration-initialized Learning (RAPPEL)* in Algorithm 1. It employs reward-agnostic online exploration to enhance the offline dataset, then learns a policy through a pessimistic offline RL algorithm. Algorithm 1 significantly improves upon the sample complexity of the only dedicated hybrid RL algorithm for linear MDPs (Wagenmaker and Pacchiano, 2023) by a factor of at least $H^3$. This performs no worse than the offline-only minimax-optimal error bound from Xiong et al. (2023), with the potential of significant gains from online data. This is the first work to explore the online-to-offline approach in linear MDPs.
- In addition, we propose an offline-to-online method called *Hybrid Regression for Upper-Confidence Reinforcement Learning (HYRULE)* in Algorithm 2, where one warm-starts an

| | Upper Bound | Lower Bound |
|---|---|---|
| Offline (Error) | $\sqrt{d} \cdot \sum_{h=1}^{H} \mathbb{E}_{\pi*} \|\phi(s_h, a_h)\|_{\Sigma_{\text{off},h}^{*-1}}$ $\leqslant \sqrt{C^* d^2 H^4 / N_{\text{off}}}$ (Xiong et al., 2023) | $\sqrt{d} \cdot \sum_{h=1}^{H} \mathbb{E}_{\pi*} \|\phi(s_h, a_h)\|_{\Sigma_{\text{off},h}^{*-1}}$ $\geqslant \sqrt{C^* d^2 H^2 / N_{\text{off}}}$ (Xiong et al., 2023) |
| Online (Regret) | $\sqrt{d^2 H^3 T}$ (He et al., 2023) | $\sqrt{d^2 H^3 T}$ (Zhou et al., 2021) |
| | Result | |
| Hybrid (Online-to-offline Error) | $\sqrt{d^2 H^7 / N}$ (Wagenmaker and Pacchiano, 2023) $\sqrt{c_{\text{off}}(\mathcal{X}_{\text{off}}) d H^3 \min\{c_{\text{off}}(\mathcal{X}_{\text{off}}), H\}/N_{\text{off}}} + \sqrt{d_{\text{on}} d H^3 \min\{d_{\text{on}}, H\}/N_{\text{on}}}$ (Alg. 1) | |
| Hybrid (Offline-to-online Regret) | $C^* \sqrt{d^2 H^6 N_{\text{on}}}$ (Nakamoto et al., 2023; Song et al., 2023) $\sqrt{(C^* + c_{\text{on}}(\mathcal{X})) d^3 H^6 N_{\text{on}}}$ (Amortila et al., 2024) $\sqrt{c_{\text{off}}(\mathcal{X}_{\text{off}}) d H^5 N_{\text{on}}^2 / N_{\text{off}}} + \sqrt{d_{\text{on}} d H^5 N_{\text{on}}}$ (Tan and Xu, 2024) $\sqrt{c_{\text{off}}(\mathcal{X}_{\text{off}})^2 d H^3 N_{\text{on}}^2 / N_{\text{off}}} + \sqrt{d_{\text{on}} d H^3 N_{\text{on}}}$ (Alg. 2) | |

Table 2: Comparisons of our results to the best upper and lower bounds for offline and online RL, and existing results for hybrid RL, in linear MDPs. Often, offline data is cheaper or easier to obtain. When this happens, $N_{\text{off}} \gg N_{\text{on}}$, and the online term in our results (depending on $N_{\text{on}}$) dominates.

online RL algorithm with parameters estimated from offline data. In addition to improving the ambient dimension dependence, this algorithm enjoys a regret (or sample complexity) bound that is no worse than the online-only minimax optimal bound, with the potential of significant gains if the offline dataset is of high quality (Agarwal et al., 2022; He et al., 2023; Hu et al., 2023; Zhou et al., 2021). Our result demonstrates the provable benefits of hybrid RL in scenarios where offline samples are much cheaper or much easier to acquire.

To the best of our knowledge, we are the first to show improvements over the aforementioned lower bounds of hybrid RL algorithms (in the same vein as Li et al. (2023b)) in the presence of function approximation, without any explicit requirements on the quality of the behavior policy, and with both the offline-to-online and online-to-offline approaches. Our results are also, at the point of writing, the best bounds available in the literature for hybrid RL in linear MDPs (see Table 2).

**Technical contributions.** In this work, we build on recent advancements in offline and online RL, demonstrating that intuitive modifications suffice to achieve state-of-the-art sample complexity for hybrid RL in linear MDPs. At a high level, our sample efficiency gains are achieved by decomposing the error of interest into offline and online partitions, and optimizing them respectively, following the same idea in Tan and Xu (2024). Below, we summarize our specific technical contributions.

1. We sharpen the dimensional dependence from $d$ to $d_{\text{on}}$ and $c_{\text{off}}(\mathcal{X}_{\text{off}})$ via projections onto those partitions. The former is accomplished in Algorithm 1 by Kiefer-Wolfowitz in Lemma 1, and in Algorithm 2 by proving a sharper variant of Lemma B.1 from Zhou and Gu (2022) in Lemma 18, using this in Lemma 14 to reduce the dimensional dependence in the summation of bonuses, which helps achieve the desired result.
2. We maintain a $H^3$ dependence for the error or regret for both algorithms, which is non-trivial, in Algorithm 1 and for the offline partition in Algorithm 2 by combining the total variance lemma with a novel truncation argument for "bad" trajectories in Lemma 17.

## 2 Preliminaries

### 2.1 Basics of Markov decision processes

An episodic MDP is a tuple $\mathcal{M} = (\mathcal{S}, \mathcal{A}, H, (\mathbb{P}_h)_{h=1}^{H}, (r_h)_{h=1}^{H})$, where $\mathcal{S}$ is the state space, $\mathcal{A}$ the action space, $H$ the horizon, $(\mathbb{P}_h)_{h=1}^{H}$ the collection of transition probability kernels $\mathbb{P}_h : \mathcal{S} \times \mathcal{A} \to \Delta(\mathcal{S})$, and $(r_h)_{h=1}^{H}$ the collection of reward functions $r_h : \mathcal{S} \times \mathcal{A} \to [0, 1]$. $\Delta(\cdot)$ is the collection of distributions over a set. At each $h \in [H] = \{1, ..., H\}$, an agent observes the current state $s_h \in \mathcal{S}$, takes an action $a_h \in \mathcal{A}$ according to $\pi_h : \mathcal{S} \to \Delta(\mathcal{A})$, and observes the reward $r_h$ and next state $s_{h+1} \sim \mathbb{P}_h(\cdot \mid s_h, a_h)$. We write $\Pi$ for the set of policies $\pi = \{\pi_h\}_{h=1}^{H}$, with value and Q-functions

$$\text{for every } (s, h) \in \mathcal{S} \times [H] : \quad V_h^\pi(s) := \mathbb{E}_\pi[\textstyle\sum_{h'=h}^{H} r_{h'} | s_h = s], \tag{1}$$

$$\text{and for every } (s, a, h) \in \mathcal{S} \times \mathcal{A} \times [H] : \quad Q_h^\pi(s, a) := \mathbb{E}_\pi[\textstyle\sum_{h'=h}^{H} r_{h'} | s_h = s, a_h = a]. \tag{2}$$

$\pi^* = \{\pi^*\}_{h=1}^{H}$ is the optimal policy attaining the highest value and Q-functions, and we write $V^* = \{V_h^*\}_{h=1}^{H}$ and $Q^* = \{Q_h^*\}_{h=1}^{H}$ for the optimal value and Q-functions. We consider the setting of hybrid RL, where an agent has access to two sources of data:

- $N_{\text{off}}$ independent episodes of length $H$ collected by a behavior policy $\pi_b$ where the $n$-th sample trajectory is a sequence of data $(s_1^{(n)}, a_1^{(n)}, r_1^{(n)}, ..., s_H^{(n)}, a_H^{(n)}, r_H^{(n)}, s_{H+1}^{(n)})$;
- $N_{\text{on}}$ sequential episodes of online data, where at each episode $n = 1, ..., N_{\text{on}}$, the algorithm has knowledge of the $N_{\text{off}}$ offline episodes and the previous online episodes $1, ..., n-1$.

The quality of the behavior policy $\pi_b$ is measured by the all-policy and single-policy concentrability coefficients proposed by Xie et al. (2023); Zhan et al. (2022):

**Definition 1** (Occupancy Measure). *For a policy $\pi = \{\pi_h\}_{h=1}^H$, its occupancy measure $d^\pi = \{d_h^\pi\}_{h=1}^H$ corresponds to the collection of distributions over states and actions induced by running $\pi$ within $\mathcal{M}$, where for some initial distribution $\rho$ and $s_1 \sim \rho$, we have*

$$d_h^\pi(s, a) := \mathbb{P}(s_h = s, a_h = a \mid s_1 \sim \rho, \pi). \tag{3}$$

**Definition 2** (Concentrability Coefficient). *The all-policy and single-policy concentrability coefficients of $\pi$ with regard to the occupancy measure $\mu = \{\mu_h\}_{h=1}^H$ of a behavior policy $\pi_b$ are*

$$C_{all} := \sup_\pi \sup_{h,s,a} \frac{d_h^\pi(s,a)}{\mu_h(s,a)} \quad and \quad C^* := \sup_{h,s,a} \frac{d_h^*(s,a)}{\mu_h(s,a)}, \tag{4}$$

**Policy learning and regret minimization.** Hybrid RL aims to either learn an $\epsilon$-optimal policy $\hat{\pi}$ such that $V^* - V^{\hat{\pi}} \leqslant \epsilon$ with high probability, or to minimize the regret. Here, the regret of an online algorithm $\mathcal{L} : \mathcal{H} \to \Pi$ is $\text{Reg}_{\mathcal{L}}(T) := \mathbb{E}[\sum_{t=1}^T (V_1^*(s_1^{(t)}) - \sum_{h=1}^H r_h^{(t)})]$. We write $T = N_{\text{on}}$ interchangeably for the number of episodes taken by a regret-minimizing online RL algorithm.

## 2.2 Linear MDPs

Throughout this paper, we study linear MDPs, where the transition probabilities and rewards are linearly parametrizable as functions of known features. This was first proposed by Jin et al. (2019); Yang and Wang (2019), and further studied in He et al. (2023); Hu et al. (2023); Wagenmaker and Jamieson (2023); Wagenmaker and Pacchiano (2023); Xiong et al. (2023); Zanette et al. (2021).

**Assumption 1** (Linear MDP, Jin et al. (2019)). *There exists a known feature map $\phi : \mathcal{S} \times \mathcal{A} \to \mathbb{R}^d$, $d$ unknown signed measures $\mu_h = (\mu_h^{(1)}, \cdots, \mu_h^{(d)})$ over $\mathcal{S}$ for each $h$, and an unknown vector $\theta_h \in \mathbb{R}^d$, such that for any $s, a, h$ we have $\mathbb{P}_h(\cdot \mid s, a) = \langle \phi(s,a), \mu_h(\cdot) \rangle$, $r_h(s,a) = \langle \phi(s,a), \theta_h \rangle$. Assume $\|\phi(s,a)\| \leqslant 1$ for all $s, a$, and $\max\{\|\mu_h(\mathcal{S})\|, \|\theta_h\|\} \leqslant \sqrt{d}$ for all $h$.*

This allows for sample-efficient RL for a few reasons. Firstly, linear MDPs are Bellman complete (Jin et al., 2021a), a common assumption for sample-efficient RL in the literature (Duan and Wang, 2020; Fan et al., 2020; Munos and Szepesvári, 2008). Secondly, the value and Q-functions are linearly parametrizable in the features, allowing one to learn them via ridge regression. This allows for sample-efficient online (He et al., 2023; Hu et al., 2023) and offline (Xiong et al., 2023; Yin et al., 2022) RL with function approximation. However, existing guarantees for hybrid RL in linear MDPs (Wagenmaker and Pacchiano, 2023) are loose (Li et al., 2023b), inspiring our work.

**Further notation.** Write $\phi_{n,h} = \phi(s_h^{(n)}, a_h^{(n)})$ for the feature vector at episode $n$ and horizon $h$. Let $\boldsymbol{\Lambda}_h = \sum_{n=1}^N \phi_{n,h}\phi_{n,h}^\top + \lambda\mathbf{I}$ and $\boldsymbol{\Lambda}_{\text{off},h} = \sum_{n=1}^{N_{\text{off}}} \phi_{n,h}\phi_{n,h}^\top + \lambda\mathbf{I}$ be the covariance matrices of the entire dataset and the offline dataset respectively, and $\boldsymbol{\Omega}$ the set of all covariates. We consider two kinds of variance-weighted covariance matrices, namely $\boldsymbol{\Sigma}_{n,h}^* = \sum_{n=1}^N \phi_{n,h}\phi_{n,h}^\top / [\mathbb{V}_h V_{h+1}^*](s_h^\tau, a_h^\tau) + \lambda\mathbf{I}$ and $\boldsymbol{\Sigma}_{n,h} = \sum_{n=1}^N \bar{\sigma}_{n,h}^{-2}\phi_{n,h}\phi_{n,h}^\top + \lambda\mathbf{I}$, where $[\mathbb{V}_h V_{h+1}^*](s_h^\tau, a_h^\tau) = \max\{1, [\text{Var}_h V_{h+1}^*](s,a)\}$ is the truncated variance of the optimal value function (where $s, a$ are random variables) and $\bar{\sigma}_{n,h}^{-2}$ is the variance estimator from He et al. (2023).

## 2.3 Exploring the state-action space

We aim to develop efficient hybrid RL algorithms for linear MDPs that do not rely on single-policy concentrability over the entire state-action space, which entails that the behavior policy covers every state-action pair that $\pi^*$ visits. A natural idea from Li et al. (2023b) is to partition this space into a component that is well-covered by the behavior policy, which we call the offline partition $\mathcal{X}_{\text{off}}$, and a component requiring further exploration, which we call the online partition $\mathcal{X}_{\text{on}}$. Based on this partition, similarly to Tan and Xu (2024), the estimation error or regret of a hybrid RL algorithm can be analyzed on each component separately. We define $\mathcal{X}_{\text{on}} \cup \mathcal{X}_{\text{off}} = [H] \times \mathcal{S} \times \mathcal{A}$, with their images under the feature map $\Phi_{\text{off}} = \text{Span}(\phi(\mathcal{X}_{\text{off},h}))_{h\in[H]} \subseteq \mathbb{R}^d$ and $\Phi_{\text{on}} = \text{Span}(\phi(\mathcal{X}_{\text{on},h}))_{h\in[H]} \subseteq \mathbb{R}^d$ being subspaces of dimension $d_{\text{off}}$ and $d_{\text{on}}$ respectively. Write $\mathcal{P}_{\text{off}}, \mathcal{P}_{\text{on}}$ for the orthogonal projection operators onto these subspaces respectively. Let $\lambda_k(M)$ denote the $k$-th largest eigenvalue

---

**Algorithm 1** Reward-Agnostic Pessimistic PAC Exploration-initialized Learning (RAPPEL)

---

1: **Input:** Offline dataset $\mathcal{D}_{\text{off}}$, samples sizes $N_{\text{on}}$, $N_{\text{off}}$, feature maps $\phi_h$, tolerance parameter for reward-agnostic exploration $\tau$.

2: **Initialize:** $\mathcal{D}_h^{(0)} \leftarrow \varnothing \ \ \forall h \in [H]$, $\lambda = 1/H^2$, $\beta_2 = \tilde{O}(\sqrt{d})$.

3: **for** horizon $h = 1, ..., H$ **do**

4:   Run an exploration algorithm (OPTCOV, Wagenmaker and Jamieson (2023)) to collect covariates $\mathbf{\Lambda}_h$ such that $\max_{\phi_h \in \Phi} \phi_h^\top (\mathbf{\Lambda}_h + \lambda \mathbf{I} + \mathbf{\Lambda}_{\text{off},h})^{-1} \phi_h \leqslant \tau$.

5: **end for**

6: **Output:** $\widehat{\pi}$ from running a pessimistic offline RL algorithm (LinPEVI-ADV+, Xiong et al. (2023)) with hyperparameters $\lambda, \beta_2$ on the combined dataset $\mathcal{D}_{\text{off}} \cup \{\mathcal{D}_h^{(N_{\text{on}})}\}_{h \in [H]}$.

---

of a symmetric matrix $M$. We borrow the definition of partial offline all-policy concentrability,[1]

$$c_{\text{off}}(\mathcal{X}_{\text{off}}) := \max_h \ 1/\lambda_{d_{\text{off}}}(\mathbb{E}_{\mu_h}[(\mathcal{P}_{\text{off}}\phi_h)(\mathcal{P}_{\text{off}}\phi_h)^\top]), \tag{5}$$

from Tan and Xu (2024), where we use the convention that $c_{\text{off}}(\varnothing) = 0$. This corresponds to the inverse of the $d_{\text{off}}$-th largest eigenvalue of the covariance matrix of the projected feature maps. Similarly, the partial all-policy analogue of the coverability coefficient from Xie et al. (2022a) is

$$c_{\text{on}}(\mathcal{X}_{\text{on}}) := \inf_\pi \max_h \ 1/\lambda_{d_{\text{on}}}(\mathbb{E}_{d_h^\pi}[(\mathcal{P}_{\text{on}}\phi_h)(\mathcal{P}_{\text{on}}\phi_h)^\top]). \tag{6}$$

As we shall see, these quantities characterize the estimation error of our proposed algorithms.

## 3 Algorithms and main results

We provide two algorithms with improved statistical guarantees to tackle the unsolved (Table 2) problem of achieving sharp guarantees with hybrid RL in linear MDPs, with different approaches:

1. Performing reward-agnostic online exploration (Wagenmaker and Pacchiano, 2023) to augment the offline data, then invoking offline RL (Xiong et al., 2023) to learn an $\epsilon$-optimal policy on the combined dataset, in the same vein of Li et al. (2023b). This is Algorithm 1.
2. Warm-starting an online RL algorithm (He et al., 2023) with parameters estimated from an offline dataset to minimize regret, as in Song et al. (2023), with details in Algorithm 2.

### 3.1 Offline RL after online exploration

Algorithm 1 collects online samples informed by the degree of coverage (or lack thereof) of the offline dataset $\mathcal{D}_{\text{off}}$ with a reward-agnostic online exploration algorithm called OPTCOV from Wagenmaker and Jamieson (2023). OPTCOV explores so that the smallest eigenvalue of the covariance matrix, $\lambda_{\min}(\mathbf{\Lambda}_h)$, is no smaller than a tolerance parameter $1/\tau$. We then learn a policy from the combined dataset using a minimax-optimal pessimistic offline RL algorithm from Xiong et al. (2023), LinPEVI-ADV+. To employ OPTCOV, one requires a modified analogue of the full-rank covariate assumption from Wagenmaker and Pacchiano (2023) that ensures that the MDP is "explorable" enough. This assumption is only imposed for Algorithm 1.

**Assumption 2** (Full Rank Projected Covariates). *For any partition $\mathcal{X}_{\text{on}} \cup \mathcal{X}_{\text{off}} = [H] \times \mathcal{S} \times \mathcal{A}$,*

$$c_{\text{on}}(\mathcal{X}_{\text{on}}) < \infty, \text{ or equivalently that } \inf_\pi \min_h \lambda_{d_{\text{on}}}(\mathbb{E}_{d_h^\pi}[(\mathcal{P}_{\text{on}}\phi_h)(\mathcal{P}_{\text{on}}\phi_h)^\top]) = \lambda_{d_{\text{on}}}^* > 0.$$

Informally, this states that for any partition, there exists some "optimal exploration policy" that ensures that the projected covariates onto the online partition have the same rank as its dimension at every timestep. In practice, this is achievable for any linear MDP via projecting the features onto the eigenspace corresponding to the nonzero singular values. We can then establish the following:

**Lemma 1** (Partial Coverability Is Bounded In Linear MDPs). *For any partition $\mathcal{X}_{\text{off}}, \mathcal{X}_{\text{on}}$, it satisfies that $c_{\text{on}}(\mathcal{X}_{\text{on}}) \leqslant d_{\text{on}}$. Also, there exists at least one partition such that $c_{\text{off}}(\mathcal{X}_{\text{off}}) = O(d)$.*

The proof of this lemma is deferred to Appendix D. This result allows us to bound the error on the offline and online partitions by the dimensionality of the partitions, instead of the coverability coefficient. Define $\alpha_{\text{off}} := N_{\text{off}}/N$, $\alpha_{\text{on}} := N_{\text{on}}/N$, and the minimal online samples for exploration

$$N^*(\tau) := \min_N N \quad \text{s.t.} \quad \inf_{\mathbf{\Lambda} \in \mathbf{\Omega}} \max_{\phi \in \Phi} \phi^\top \left(N(\mathbf{\Lambda} + \bar{\lambda}I) + \mathbf{\Lambda}_{\text{off}}\right)^{-1} \phi \leqslant \tau.$$

We now have, with full proof in Appendix B and sketch at the end of the subsection, the following:

---

[1]Some authors regard "partial" as a synonym of "single-policy" and "full" as a synonym of "all-policy". We use the term "partial" to refer to concentrability or coverability over only part of the state-action space. There is always a partition so that the partial all-policy concentrability coefficient is less than the full single-policy concentrability coefficient (with a larger online partition), and the former can be finite when the latter is infinite.

**Theorem 1** (Error Bound for RAPPEL, Algorithm 1). *For every $\delta \in (0,1)$ and any partition $\mathcal{X}_{\text{off}}, \mathcal{X}_{\text{on}}$, when choosing $\tau \leqslant \tilde{O}(\max\{d_{\text{on}}/N_{\text{on}}, c_{\text{off}}(\mathcal{X}_{\text{off}})/N_{\text{off}}\})$, RAPPEL achieves w.p. $1 - \delta$:*

$$V_1^* - V_1^{\hat{\pi}} \lesssim \sqrt{d} \sum_{h=1}^{H} \mathbb{E}_{\pi*} ||\phi(s_h, a_h)||_{(\boldsymbol{\Sigma}_{\text{off},h}^* + \boldsymbol{\Sigma}_{\text{on},h}^*)^{-1}} \leqslant \sqrt{d} \sum_{h=1}^{H} \mathbb{E}_{\pi*} ||\phi(s_h, a_h)||_{\boldsymbol{\Sigma}_{\text{off},h}^{*-1}}, \quad (7)$$

$$V_1^* - V_1^{\hat{\pi}} \lesssim \min \left\{ \sqrt{\frac{c_{\text{off}}(\mathcal{X}_{\text{off}})dH^4}{N_{\text{off}}}} + \sqrt{\frac{d_{\text{on}}dH^4}{N_{\text{on}}}}, \sqrt{\frac{c_{\text{off}}(\mathcal{X}_{\text{off}})^2 dH^3}{N_{\text{off}}\alpha_{\text{off}}}} + \sqrt{\frac{d_{\text{on}}^2 dH^3}{N_{\text{on}}\alpha_{\text{on}}}} \right\}, \quad (8)$$

*given $N \geqslant \max\left\{\alpha_{\text{on}}^4 d_{\text{on}}^{-4}, \alpha_{\text{off}}^4 c_{\text{off}}(\mathcal{X}_{\text{off}})^{-4}\right\} \max\{N^*(\tau), poly(d, H, c_{\text{off}}(\mathcal{X}_{\text{off}}), \log 1/\delta)\}$.*

This result, when applied to tabular MDPs with finite states and actions, yields:

**Corollary 1.** *In tabular MDPs, for every $\delta \in (0,1)$, it satisfies that with probability at least $1 - \delta$,*

$$V_1^{\star}(s) - V_1^{\hat{\pi}}(s) \lesssim \sqrt{H^3 |\mathcal{S}|^2 |\mathcal{A}|} \left( \sqrt{c_{\text{off}}(\mathcal{X}_{\text{off}})/N_{\text{off}}} + \sqrt{d_{\text{on}}/N_{\text{on}}} \right). \quad (9)$$

In sum, Theorem 1 shows that with a $poly(d, H)$ burn-in cost that is no smaller than $N^*$ (the minimal online samples for any algorithm to achieve our choice of OPTCOV tolerance), we require only

$$c_{\text{off}}(\mathcal{X}_{\text{off}})dH^3 \min\{c_{\text{off}}(\mathcal{X}_{\text{off}}), H\}/\epsilon^2 + d_{\text{on}}dH^3 \min\{d_{\text{on}}, H\}/\epsilon^2$$

trajectories to learn an $O(\epsilon)$-optimal policy. $N^*$, from Wagenmaker and Pacchiano (2023), is essentially unavoidable in reward-agnostic exploration for linear MDPs. To compare with prior literature, our result leads to a better worst-case guarantee than the error bound of $\sqrt{d^2 H^7/N}$ attained in Wagenmaker and Pacchiano (2023) (by at least a factor of $H^{3/2}$), the only other work on hybrid RL in linear MDPs thus far. While we employ the same online exploration procedure, we combine our exploration phase with an offline learning algorithm LinPEVI-ADV+ from Xiong et al. (2023) and conduct a careful analysis. When comparing with the offline-only and online-only settings, Theorem 1 improves upon the offline-only minimax-optimal error bound of $\sqrt{d} \sum_{h=1}^{H} \mathbb{E}_{\pi*} ||\phi(s_h, a_h)||_{\Sigma_{\text{off},h}^{*-1}}$ from Xiong et al. (2023) as a consequence of $\Sigma_{\text{off},h}^* + \Sigma_{\text{on},h}^* \geqslant \Sigma_{\text{off},h}^*$; the best offline-only error bound is $\sqrt{d^2 H^4/N_{\text{off}}}$ obtained under the "well-covered" assumption (Corollary 4.6, Jin et al. (2021b)) that $\lambda_{\min}(\boldsymbol{\Lambda}_{h,\text{off}}) \geqslant \Omega(1/d)$, Theorem 1 enjoys better dimension and horizon dependence as there is always a partition such that $d_{\text{on}}, c_{\text{off}}(\mathcal{X}_{\text{off}}) \leqslant d$ and $d_{\text{on}}H^3 \min\{d_{\text{on}}, H\} \leqslant d^2 H^4$.

The literature has experienced considerable difficulty in sharpening the horizon dependence to $H^3$ in offline RL for linear MDPs. While Yin et al. (2022) and Xiong et al. (2023) provide minimax-optimal algorithms for offline RL in linear MDPs, both only manage to achieve a $H^3$ horizon dependence in the special case of tabular MDPs, even under the aforementioned "well-covered" assumption. We provide the same result in Corollary 1 with proof deferred to Appendix C, but encouragingly, hybrid RL lets us bypass the "well-covered" assumption. In Appendix B and G, we use a novel truncation argument and the total variance lemma (Lemma C.5 of Jin et al. (2018)) to improve the dependence on $H$, but our result falls slightly short of $\sqrt{c_{\text{off}}(\mathcal{X}_{\text{off}})dH^3/N_{\text{off}}} + \sqrt{d_{\text{on}}dH^3/N_{\text{on}}}$.

**Computational efficiency.** In terms of computational efficiency, Algorithm 1 inherits the computational costs of the previous proposed algorithms OPTCOV and LinPEVI-ADV+ (Wagenmaker and Jamieson (2023); Xiong et al. (2023). OPTCOV runs in polynomial time $poly(d, H, c_{\text{on}}(\mathcal{X}_{\text{on}}), \log 1/\delta)$, and LinPEVI-ADV+ runs in $\tilde{O}(d^3 HN|\mathcal{A}|)$ time when the action space is discrete. Algorithm 1 therefore remains computationally efficient in this case.

**Requirement of choosing $d_{\text{on}}$.** There is the caveat that we require the user to choose the tolerance for OPTCOV. In practice, one can achieve this by performing SVD on the offline dataset and looking at the plot of eigenvalues. One can also choose a tolerance of $O(d/\min\{N_{\text{off}}, N_{\text{on}}\})$, but this would not achieve the reduction in the dependence on dimension from $d^2$ to $c_{\text{off}}(\mathcal{X}_{\text{off}})d, d_{\text{on}}d$.

**Practical benefits of the online-to-offline approach.** Algorithm 1 outputs a fixed policy satisfying a PAC bound. This enables policies to be deployed in critical real-world applications, such as in medicine or defense, where randomized policies from regret minimization are unacceptable.

**Algorithm 1 works for reward-agnostic hybrid RL.** The use of reward-agnostic online exploration in Algorithm 1 enables one to use the hybrid dataset $\mathcal{D}$ to learn policies for different reward functions offline. As the online exploration is not influenced by any single reward function, the collected data satisfies good coverage for any possible reward function even if it is revealed only after exploration, enabling one to use a single dataset to achieve success on many different tasks. This therefore also serves as an algorithm for the related setting of *reward-agnostic hybrid RL*, where the reward function is unknown during online exploration and only revealed to the agent after it.

---

**Algorithm 2** Hybrid Regression for Upper-Confidence Reinforcement Learning (HYRULE)

1: **Input:** Offline dataset $\mathcal{D}_{\text{off}}$, samples sizes $N_{\text{on}}$, $N_{\text{off}}$, feature maps $\phi_h$. Regularization parameter $\lambda > 0$, confidence radii $\beta, \bar{\beta}, \tilde{\beta}$, $t_{\text{last}} = 0$.

2: **Initialize:** For $h \in [H]$, estimate $\widehat{\mathbf{w}}_{1,h}, \check{\mathbf{w}}_{1,h}, Q_{1,h}, \check{Q}_{1,h}, \sigma_{1,h}, \bar{\sigma}_{1,h}$ from $\mathcal{D}_{\text{off}}$, and assign $\boldsymbol{\Sigma}_{0,h} = \boldsymbol{\Sigma}_{1,h} = \boldsymbol{\Sigma}_{\text{off}} + \lambda \mathbf{I} = \sum_{n=1}^{N_{\text{off}}} \bar{\sigma}_{n,h}^{-2} \phi_{n,h} \phi_{n,h}^{\top} + \lambda \mathbf{I}$.

3: **for** episodes $t = 1, ..., T$ **do**

4:     Update optimistic and pessimistic weights $\widehat{\mathbf{w}}_{t,h}, \check{\mathbf{w}}_{t,h}$ for all $h$.

5:     **if** there exists a stage $h' \in [H]$ such that $\det(\boldsymbol{\Sigma}_{t,h'}) \geqslant 2 \det(\boldsymbol{\Sigma}_{t_{\text{last}},h'})$ **then**

6:         Update optimistic and pessimistic Q-functions $Q_{t,h}(s,a), \check{Q}_{t,h}(s,a)$, set $t_{\text{last}} = t$.

7:     **end if**

8:     **for** horizon $h = 1, ..., H$ **do**

9:         Play action $a_h^{(t)} \leftarrow \arg\max_a Q_{t,h}(s_h^{(t)}, a)$, receive reward $r_h^{(t)}$, next state $s_{h+1}^{(t)}$

10:         Estimate $\sigma_{t,h}, \bar{\sigma}_{t,h} \leftarrow \max\{\sigma_{t,h}, \sqrt{H}, 2d^3 H^2 \|\phi(s_h^{(t)}, a_h^{(t)})\|_{\boldsymbol{\Sigma}_{t,h}^{-1}}^{1/2}\}^2$, update $\boldsymbol{\Sigma}_{t+1,h}$.

11:     **end for**

12: **end for**

13: **Output:** Greedy policy $\widehat{\pi} = \pi^{Q_{T,h}}$, $\text{Unif}(\pi^{Q_{1,h}}, ..., \pi^{Q_{T,h}})$ for PAC guarantee.

---

**Proof sketch.** The relation (7) in Theorem 1 follows from invoking Theorem 2 from Xiong et al. (2023) with $N > \Omega(d^2 H^6), \lambda = 1/H^2, \beta_1 = O(\sqrt{d})$. To establish (8), we first bound

$$V_1^*(s) - V_1^{\widehat{\pi}}(s) \leqslant \sqrt{d} \sum_{h=1}^{H} \max_{\phi_h \in \Phi_{\text{on}}} \sqrt{\phi_h^{\top} \boldsymbol{\Sigma}_h^{*-1} \phi_h} + \sqrt{d} \sum_{h=1}^{H} \max_{\phi_h \in \Phi_{\text{off}}} \sqrt{\phi_h^{\top} \boldsymbol{\Sigma}_h^{*-1} \phi_h}.$$

As $\boldsymbol{\Sigma}_h^{*-1} \leqslant H^2 \boldsymbol{\Lambda}_h^{-1}$ (see Xiong et al. (2023)), it therefore boils down to controlling $\max_{\phi_h \in \Phi} \phi_h^{\top} \boldsymbol{\Lambda}_h^{-1} \phi_h$. Towards this, first, we make the observation that Lemma 1 suggests that $c_{\text{on}}(\mathcal{X}_{\text{on}}) \leqslant d_{\text{on}}$. If we run OPTCOV with tolerance $\tilde{O}(\max\{d_{\text{on}}/N_{\text{on}}, c_{\text{off}}(\mathcal{X}_{\text{off}})/N_{\text{off}}\})$ on partitions where the above hold, in Lemma 5, we prove that $\max_{\phi_h \in \Phi} \phi_h^{\top} \boldsymbol{\Lambda}_h^{-1} \phi_h \lesssim \max\{c_{\text{off}}(\mathcal{X}_{\text{off}})/N_{\text{off}}, d_{\text{on}}/N_{\text{on}}\}$. This yields the $c_{\text{off}}(\mathcal{X}_{\text{off}})dH^4, d_{\text{on}}dH^4$ result.

To tighten the horizon dependence to $H^3$, we employ an useful truncation argument. More specifically, from the total variance lemma (Lemma C.5 of Jin et al. (2018)), the average variance $\mathbb{V}_h V_{h+1}^*$ is asymptotically on the order of $H$. We therefore define the sets of trajectories $\mathcal{E}_h(\delta_h) = \{\tau \in \mathcal{D} : [\mathbb{V}_h V_{h+1}^*](s_h^{\tau}, a_h^{\tau}) \geqslant H^{1+\delta_h}\}$. The cardinality of each set can be bounded by $|\mathcal{E}_h(\delta_h)| \lesssim N H^{1-\delta_h}$, and so truncating at the level where $N H^{1-\delta_h} \approx \min(\frac{N_{\text{off}}}{c_{\text{off}}(\mathcal{X}_{\text{off}})}, \frac{N_{\text{on}}}{d_{\text{on}}})$ leads to $\min_{\phi_h \in \Phi} \phi_h^{\top} \Sigma_h^* \phi_h \gtrsim \frac{1}{NH^2} \min(\frac{N_{\text{off}}}{c_{\text{off}}(\mathcal{X}_{\text{off}})}, \frac{N_{\text{on}}}{d_{\text{on}}})^2$. Putting things together yields the last $c_{\text{off}}(\mathcal{X}_{\text{off}})^2 dH^3, d_{\text{on}}^2 dH^3$ result needed, and the theorem then follows.

### 3.2 Online regret minimization

Thus far, we described an online-to-offline strategy which collects online samples to augment the offline dataset. However, in certain critical cases, such as with a doctor treating patients, performance-agnostic online exploration is untenable. One may wish to minimize the regret of the online actions taken while learning a policy. We therefore explore another approach inspired by the work of Song et al. (2023); Tan and Xu (2024) – that of warm-starting an online RL algorithm with parameters estimated from an offline dataset. We describe this in Algorithm 2, and show that hybrid RL enables provable gains over minimax-optimal online-only regret bounds in the offline-to-online case as well.

In order to warm-start an online RL algorithm with an offline dataset, we modify LSVI-UCB++ from He et al. (2023) by estimating its parameters from $\mathcal{D}_{\text{off}}$ with the same formulas it would use as if it had experienced the $N_{\text{off}}$ offline episodes itself. As Tan and Xu (2024) suggest, this can be understood as including the offline episodes in the "experience replay buffer" that the algorithm uses to learn parameters. The full version can be found in Appendix E as Algorithm 4. Doing so allows us prove a regret bound depending on the partial all-policy concentrability coefficient. Below we state our theoretical guarantees for this algorithm. The proof of this result is deferred to Appendix E, and a brief proof sketch is provided at the end of this subsection.

---

[2]He et al. (2023) write $\bar{\sigma}_{t,h} \leftarrow \max\{\sigma_{t,h}, H, ...\}$, not $\sqrt{H}$, which we believe is a typo. In the proof of Lemma B.1, after equation D.7, they state that $0 \leqslant \bar{\sigma}_{ih}^{-1} \leqslant 1/\sqrt{H}$. In the proof of Lemma B.5, the argument after equation D.22, particularly that $\left\| \bar{\sigma}_{i,h}^{-1} \phi(s_h^i, a_h^i) \right\|_2 \leqslant \left\| \phi(s_h^i, a_h^i) \right\|_2 / \sqrt{H}$, only holds if this is $\sqrt{H}$.

**Theorem 2** (Regret Bound for HYRULE, Algorithm 2)**.** *Given any $\delta \in (0,1)$, for every partition $\mathcal{X}_{\mathrm{off}}, \mathcal{X}_{\mathrm{on}}$, if $N_{\mathrm{on}}, N_{\mathrm{off}} = \tilde{\Omega}(d^{13}H^{14})$, the regret of HYRULE is bounded w.p. at least $1 - \delta$ by*

$$Reg(N_{\mathrm{on}}) \lesssim \inf_{\mathcal{X}_{\mathrm{off}}, \mathcal{X}_{\mathrm{on}}} \sqrt{c_{\mathrm{off}}(\mathcal{X}_{\mathrm{off}})^2 dH^3 N_{\mathrm{on}}^2 / N_{\mathrm{off}}} + \sqrt{d_{\mathrm{on}} dH^3 N_{\mathrm{on}}}.$$

**Corollary 2.** *By the regret-to-PAC conversion, Algorithm 2 achieves a sub-optimality gap w.p. $1 - \delta$:*

$$V_1^*(s) - V_1^{\hat{\pi}}(s) \lesssim \inf_{\mathcal{X}_{\mathrm{off}}, \mathcal{X}_{\mathrm{on}}} \sqrt{c_{\mathrm{off}}(\mathcal{X}_{\mathrm{off}})^2 dH^3 / N_{\mathrm{off}}} + \sqrt{d_{\mathrm{on}} dH^3 / N_{\mathrm{on}}}.$$

To understand this result, we first note that bounding the regret over all possible partitions yields an improvement over the $\sqrt{d^2 H^3 N_{\mathrm{on}}}$ regret bound obtained by He et al. (2023), as we can take $\mathcal{X}_{\mathrm{on}} = \mathcal{X}, \mathcal{X}_{\mathrm{off}} = \varnothing$ to recover the $\sqrt{d^2 H^3 N_{\mathrm{on}}}$ bound. In the scenario where offline samples are abundant (where $N_{\mathrm{off}} \gg N_{\mathrm{on}}$), it is possible to achieve significant improvements over online-only learning. Furthermore, in view of Lemma 1, there always exists a partition such that $c_{\mathrm{off}}(\mathcal{X}_{\mathrm{off}}), d_{\mathrm{on}} \leq d$. This result therefore yields provable improvements over the minimax-optimal online regret bound in linear MDPs (Agarwal et al., 2022; He et al., 2023; Hu et al., 2023; Zhou et al., 2021).

Additionally, Theorem 2 shows that Algorithm 2 attains the best known regret bound in hybrid RL for linear MDPs, as we illustrate in Table 2. The current best known result is that of Tan and Xu (2024), with a dependence of $\sqrt{c_{\mathrm{off}}(\mathcal{X}_{\mathrm{off}}) dH^5 N_{\mathrm{on}}^2 / N_{\mathrm{off}}} + \sqrt{d_{\mathrm{on}} dH^5 N_{\mathrm{on}}}$. Notably, we achieve the same a reduction in the dimension dependence on the online partition from $d^2$ to $d_{\mathrm{on}} d$ that Tan and Xu (2024) do by proving a sharper variant of Lemma B.1 from Zhou and Gu (2022) in Lemma 18, using this in Lemma 14 to reduce the dimensional dependence in the summation of bonuses. Song et al. (2023) and Amortila et al. (2024), on the other hand, have bounds on the order of $C^* \sqrt{d^2 H^6 N_{\mathrm{on}}}$ and $\sqrt{(C^* + c_{\mathrm{on}}(\mathcal{X})) d^3 H^6 N_{\mathrm{on}}}$ respectively. We produce a better bound than Amortila et al. (2024); Song et al. (2023); Tan and Xu (2024) by at least a factor of $H^2$ by combining the total variance lemma and a novel truncation argument that rules out "bad" trajectories in Lemma 17, which allows us to maintain a desirable $H^3$ dependence on both partitions.

**Computational efficiency.** When the action space is finite and of cardinality $|\mathcal{A}|$, the computational complexity of Algorithm 2 is of order $\widetilde{O}\left(d^4 H^3 N |\mathcal{A}|\right)$, as outlined in He et al. (2023). Algorithm 2 is therefore computationally efficient and runs in polynomial time in this case. When the action space is continuous, one may need to solve an optimization problem over the continuous action space, making the computational complexity highly problem-dependent.

**Algorithm 2 is unaware of the partition.** Unlike Algorithm 1, Algorithm 2 is fully unaware of the choice of partition, and there is therefore no need to estimate $d_{\mathrm{on}}$ or any relevant analogue to the choice of tolerance for OPTCOV. The regret bound therefore automatically adapts to the best possible partition, even though Algorithm 2 is unaware of it.

**Practical benefits of the offline-to-online approach.** While Algorithm 2 only satisfies a PAC bound with a randomized policy, it minimizes the regret of the actions it takes. This enables the algorithm to be deployed in situations where its performance during online exploration is of critical importance, e.g. in applications like mobile health (Nahum-Shani et al., 2017).

**Technical challenges.** Although Algorithm 2 is a straightforward generalization of LSVI-UCB++ in He et al. (2023), with $\Sigma_0$ initialized with the offline dataset, we had to decompose the regret into the regret on the offline and online partitions to achieve the regret guarantee in Theorem 2. In the process, we faced the following challenges:

- Bounding the regret on the offline partition was challenging, as the argument of He et al. (2023) was not applicable. Instead, we used a truncation argument in Lemma 17 to bound the maximum eigenvalue of $\Sigma_{\mathrm{off},h}^{-1}$, maintaining a $H^3$ dependence on the offline partition.
- Bounding the regret on the online partition allowed us to use an analysis that was close to that of He et al. (2023). However, directly following their argument would have left us with a $d^2 H^3$ dependence. To reduce the dimensional dependence to $d_{\mathrm{on}} d$, we prove a sharper variant of Lemma B.1 from Zhou and Gu (2022) in Lemma 18. We use this in Lemma 14 to reduce the dimensional dependence in the sum of bonuses, achieving the desired result.
- Without the above two techniques, one could have used a simpler analysis to achieve a far looser $\sqrt{c_{\mathrm{off}}(\mathcal{X}_{\mathrm{off}})^2 d^6 H^8 N_{\mathrm{on}}^2 / N_{\mathrm{off}}} + \sqrt{d^2 H^3}$ regret bound by using the maximum magnitude of the variance weights for the offline partition and the analysis from He et al. (2023) verbatim for the online partition, but this would not have yielded the same improvement.

We accordingly provide a proof sketch below.

**Proof sketch.** We first adopt the regret decomposition as in He et al. (2023) and bound

$$\text{Reg}(T) \lesssim \sqrt{H^3 T} + \sum_{h,t} \beta \|\boldsymbol{\Sigma}_{t,h}^{-1/2} \phi_h(s_h^{(t)}, a_h^{(t)}) \mathbb{1}_{\mathcal{X}_{\text{off}}}\|_2 + \sum_{h,t} \beta \|\boldsymbol{\Sigma}_{t,h}^{-1/2} \phi_h(s_h^{(t)}, a_h^{(t)}) \mathbb{1}_{\mathcal{X}_{\text{on}}}\|_2.$$

It then boils down to controlling the second and third terms separately. We prove in Lemma 12 that the sum of bonuses on the offline partition can be bounded by $\sum_h \sqrt{d N_{\text{on}} \frac{N_{\text{on}}}{N_{\text{off}}} \max_{\phi_h \in \Phi_{\text{off}}} \phi_h^\top \bar{\boldsymbol{\Sigma}}_{\text{off},h}^{-1} \phi_h}$. Further, $\sum_h \max_{\phi_h \in \Phi_{\text{off}}} \sqrt{\phi_h^\top \bar{\boldsymbol{\Sigma}}_{\text{off},h}^{-1} \phi_h} \lesssim c_{\text{off}}(\mathcal{X}_{\text{off}})^2 H^3$ by Lemma 13. Putting things together, the second term can be controlled as

$$\beta \sum_{h,t} \|\boldsymbol{\Sigma}_{t,h}^{-1/2} \phi_h(s_h^{(t)}, a_h^{(t)}) \mathbb{1}_{\mathcal{X}_{\text{off}}}\|_2 \lesssim \sqrt{c_{\text{off}}(\mathcal{X}_{\text{off}})^2 d H^3 N_{\text{on}}^2 / N_{\text{off}}}.$$

With respect to the third term, Lemma 14 (a sharpened version of Lemma E.1 in He et al. (2023)), combined with the Cauchy-Schwartz inequality, yields

$$\beta \sum_{h,t} \|\boldsymbol{\Sigma}_{t,h}^{-1/2} \phi_h(s_h^{(t)}, a_h^{(t)}) \mathbb{1}_{\mathcal{X}_{\text{on}}}\|_2 \lesssim d^4 H^8 + \beta d^7 H^5 + \beta \sqrt{d_{\text{on}} H T + d_{\text{on}} H \sum_{h,t} \sigma_{t,h}^2}.$$

By the total variance lemma (Appendix B, He et al. (2023)), $\sum_{h,t} \sigma_{t,h}^2 \leq \widetilde{O}\left(H^2 T + d^{10.5} H^{16}\right)$. Taking everything collectively establishes the desired result.

## 4 Numerical experiments

To demonstrate the benefits of hybrid RL in the offline-to-online and online-to-offline settings, we implement Algorithms 1 and 2 on a scaled-down Tetris environment (as in Tan and Xu (2024)). For the purposes of brevity, we defer the details of the environment to Appendix H.[3]

Figure 1 depicts the coverage (defined by $1/\lambda_{\min}(\boldsymbol{\Lambda}), 1/\lambda_{d_{\text{off}}}(\boldsymbol{\Lambda}_{\text{off}}), 1/\lambda_{d_{\text{on}}}(\boldsymbol{\Lambda}_{\text{on}})$) achieved by the reward-agnostic exploration algorithm, OPTCOV, when initialized with 200 trajectories from (1) a uniform behavioral policy, (2) an adversarial behavior policy obtained by the negative of the weights of a fully-trained agent under Algorithm 1, and (3) no offline trajectories at all. Although hybrid RL with the uniform behavior policy achieves the best coverage throughout as expected, hybrid RL with even adversarially collected offline data achieves better coverage than online-only exploration. This demonstrates the potential of hybrid RL as a tool for taking advantage of poor quality offline data.

Figure 2 shows the benefits of hybrid RL in the online-to-offline setting when the behavior policy is of poor quality. When applying LinPEVI-ADV to the hybrid dataset of 200 trajectories and 100 online trajectories, 300 trajectories of adversarially collected offline data, and 300 trajectories of online data under reward-agnostic exploration, we see that the hybrid dataset is most conducive for learning. Additionally, without a warm-start from offline data, online-only reward-agnostic exploration performs worse than the adversarially collected offline data due to significant burn-in costs. Hybrid RL, in this instance, performs better than both offline-only and online-only learning alone. Figure 3 compares the performances of LSVI-UCB++ and Algorithm 2. Initializing a regret-minimizing online algorithm (LSVI-UCB++, (He et al., 2023)) with an offline dataset as in Algorithm 2 yields lower regret than LSVI-UCB++ without an offline dataset. This shows that even a nearly minimax-optimal online learning algorithm can stand to benefit from offline data.

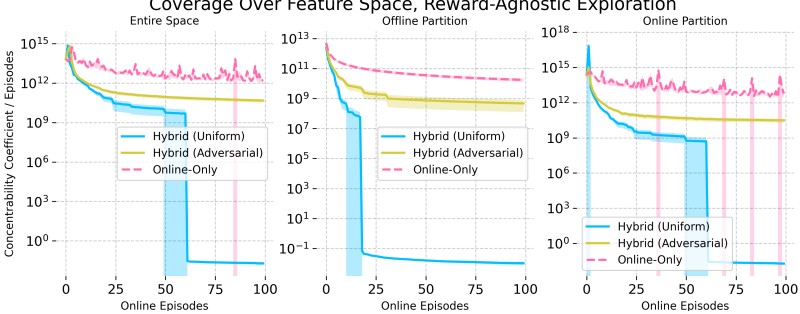

Figure 1: Coverage achieved by OPTCOV with 200 trajectories of offline data collected under a uniform and an adversarial behavior policy, and with no offline data. Results averaged over 30 trials, with the shaded area depicting 1.96-standard errors. Lower is better.

---

[3]For ease of implementation, we implement LSVI-UCB++ (He et al., 2023) for Algorithm 2 as-is, while substituting LSVI-UCB (Jin et al., 2019) for FORCE (Wagenmaker et al., 2022) within OPTCOV and LinPEVI-ADV for LinPEVI-ADV+ (Xiong et al., 2023). The code can be found at github.com/hetankevin/hybridlin.

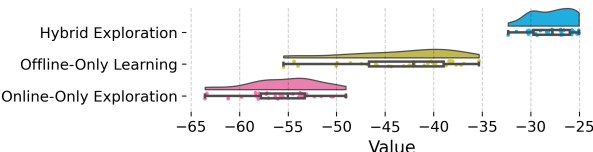

Figure 2: Value of policies learned by applying LinPEVI-ADV to the hybrid, offline, and online datasets, with an adversarial behavior policy. The reward is negative as it is the negative of the excess height. Results over 30 trials. Higher is better.

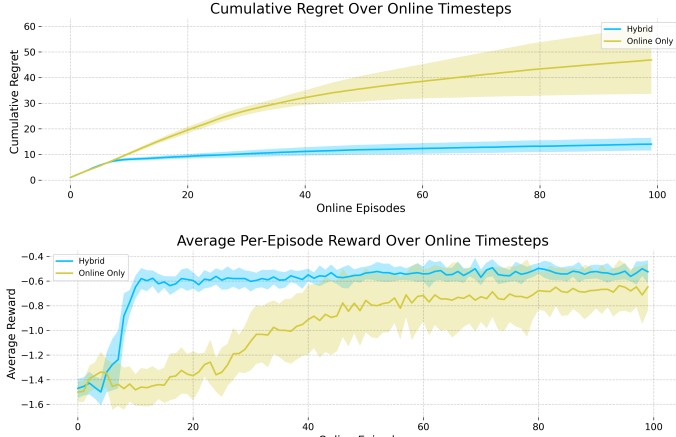

Figure 3: Comparison of LSVI-UCB++ and Algorithm 2 over 10 trials, with 1 s.d. error bars.

## 5 Discussion, limitations and future work

In this paper, we develop two hybrid RL algorithms for linear MDPs with desirable statistical guarantees for the online-to-offline and offline-to-online settings. Both algorithms demonstrate provable gains over the minimax-optimal rates in offline or online-only reinforcement learning, and provide the sharpest worst-case bounds for the performance of hybrid RL in linear MDPs thus far.

Throughout this paper, we have used both optimism and pessimism in our algorithm design. Other work in hybrid RL (Amortila et al., 2024; Li et al., 2023b; Nakamoto et al., 2023; Song et al., 2023; Tan and Xu, 2024; Wagenmaker and Pacchiano, 2023) uses optimism, pessimism, or sometimes even neither. We conjecture that optimism is still helpful in aiding online exploration within hybrid RL and that pessimism helps in hybrid RL when learning from a combined dataset. However, determining if or when optimism or pessimism is beneficial in hybrid RL remains an open question.

Achieving a $H^3$ horizon dependence in offline RL for linear MDPs has proven challenging. Even under strong coverage assumptions, Yin et al. (2022) and Xiong et al. (2023) only manage to achieve a $H^3$ horizon dependence for tabular MDPs. Obtaining a $\sqrt{d^2 H^3/N}$ bound is an open problem.

A result depending on a partial single-policy concentrability coefficient would be desirable, but may provide only limited benefits as we take the infimum over partitions. A good offline partition for the partial all-policy concentrability contains the portion of the state-action space well-covered by the offline dataset, while the same for the partial single-policy concentrability would be well-covered by both the offline dataset and the optimal policy. The smaller size of the latter offline partition may be offset by the larger size of the latter's online partition, and as such any gains may be limited.

Furthermore, while Algorithm 1 improves upon the offline-only error lower bound in Xiong et al. (2023) and Algorithm 2 improves upon the online-only regret lower bound in Zhou et al. (2021), we still desire a single algorithm that improves upon both the best possible offline-only and online-only rates at once. Additionally, the burn-in costs for Algorithms 1 and 2 are nontrivial. The former is inherited from OPTCOV (Wagenmaker and Jamieson, 2023), while the latter is inherited from He et al. (2023) and the truncation argument. Improving the former by devising new reward-agnostic exploration algorithms for linear MDPs, perhaps in the vein of Li et al. (2023a), would be welcome.

While we tackle the setting of linear MDPs, it remains a first step towards showing that hybrid RL breaks minimax-optimal barriers in the presence of function approximation. Further work in this vein on other types of function approximation would be an interesting contribution to the literature.

## Acknowledgements

Y. Wei is supported in part by the NSF grant CCF-2106778, CCF-2418156 and the CAREER award DMS-2143215.

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

# A Unabridged versions of our algorithms

---

**Algorithm 3** Reward-Agnostic Exploration-initialized Pessimistic PAC Learning (RAPPEL, Full)

---

1: **Input:** Offline dataset $\mathcal{D}_{\mathrm{off}}$, samples sizes $N_{\mathrm{on}}$, $N_{\mathrm{off}}$, feature maps $\phi_h$, , tolerance parameter for reward-agnostic exploration $\tau$.

2: **Initialize:** $\mathcal{D}_h^{(0)} \leftarrow \varnothing \quad \forall h \in [H]$, $\lambda = 1/H^2$, $\beta_2 = \tilde{O}(\sqrt{d})$. Set functions to optimize $f_i(\boldsymbol{\Lambda}) = \eta_i^{-1} \log\left(\sum_{\phi \in \Phi} \exp(\eta_i \|\phi\|^2_{\mathbf{A}_i(\boldsymbol{\Lambda})^{-1}})\right)$, $\mathbf{A}_i(\boldsymbol{\Lambda}) = \boldsymbol{\Lambda} + (T_i K_i)^{-1}(\boldsymbol{\Lambda}_{0,i} + \boldsymbol{\Lambda}_{\mathrm{off}})$ for some $\boldsymbol{\Lambda}_{0,i}$ satisfying $\boldsymbol{\Lambda}_{0,i} \geq \boldsymbol{\Lambda}_0$ for all $i$, and $\eta_i = 2^{2i/5}$.

    **Exploration Phase:** Run an exploration algorithm (OPTCOV, Wagenmaker and Jamieson (2023)) to collect covariates $\boldsymbol{\Lambda}_h$ such that $\max_{\phi_h \in \Phi} \phi_h^\top (\boldsymbol{\Lambda}_h + \lambda \mathbf{I} + \boldsymbol{\Lambda}_{\mathrm{off},h})^{-1} \phi_h \leq \tau$.

3: **for** $i = 1, 2, 3, \dots$ **do**

4:     Set the number of iterates $T_i \leftarrow 2^i$, episodes per iterate $K_i \leftarrow 2^i$.

5:     Play any policy for $K_i$ episodes to collect covariates $\boldsymbol{\Gamma}_0$ and data $\mathfrak{D}_0$.

6:     Initialize covariance matrix $\boldsymbol{\Lambda}_1 \leftarrow \boldsymbol{\Gamma}_0/K$.

7:     **for** $t = 1, \dots, T_i$ **do**

8:         **if** $\sum_{j=1}^i T_j K_j \geq N_{\mathrm{on}}$ **then**

9:             **break**

10:         **end if**

11:         Run FORCE (Wagenmaker et al., 2022) or another regret-minimizing algorithm on the exploration-focused synthetic reward $g_h^{(t)}(s,a) \propto \mathrm{tr}(-\nabla_{\boldsymbol{\Lambda}} f_i(\boldsymbol{\Lambda})|_{\boldsymbol{\Lambda} = \boldsymbol{\Lambda}_t} \phi(s,a)\phi(s,a)^\top)$.

12:         Collect covariates $\boldsymbol{\Gamma}_t$, data $\mathfrak{D}_t$.

13:         Perform Frank-Wolfe update: $\boldsymbol{\Gamma}_{t+1} \leftarrow (1 - \frac{1}{t+1})\boldsymbol{\Lambda}_t + \frac{1}{t+1}\boldsymbol{\Gamma}_t/K_i$.

14:     **end for**

15:     Assign $\widehat{\boldsymbol{\Lambda}_{i,h}} \leftarrow \boldsymbol{\Lambda}_{T_i+1}, \mathfrak{D}_i \leftarrow \cup_{t=0}^{T_i}\mathfrak{D}_t$.

16:     Set $\boldsymbol{\Lambda}_h = \widehat{\boldsymbol{\Lambda}_{i,h}}, \mathcal{D}_{\mathrm{on}} = \mathfrak{D}_i$.

17:     **if** $f_i(\widehat{\boldsymbol{\Lambda}_i}) \leq K_i T_i \tau$ **then**

18:         **break**

19:     **end if**

20: **end for**

    **Planning Phase:** Estimate $\widehat{\pi}$ using a pessimistic offline RL algorithm (LinPEVI-ADV+, Xiong et al. (2023)) with hyperparameters $\lambda, \beta_2$ on the combined dataset $\mathcal{D}_{\mathrm{off}} \cup \{\mathcal{D}_h^{(N_{\mathrm{on}})}\}_{h \in [H]}$.

21: Split the dataset $\mathcal{D}_{\mathrm{off}} \cup \{\mathcal{D}_h^{(N_{\mathrm{on}})}\}_{h \in [H]}$ into $\mathcal{D}$ and $\mathcal{D}'$. Estimate, on $\mathcal{D}'$,

$$\widetilde{\beta}_{h,2} = \underset{\beta \in \mathbb{R}^d}{\operatorname{argmin}} \sum_{\tau \in \mathcal{D}'} \left[ \langle \phi(s_h^\tau, a_h^\tau), \beta \rangle - (\widehat{V}_{h+1}')^2(s_{h+1}^\tau) \right]^2 + \lambda \|\beta\|_2^2,$$

$$\widetilde{\beta}_{h,1} = \underset{\beta \in \mathbb{R}^d}{\operatorname{argmin}} \sum_{\tau \in \mathcal{D}'} \left[ \langle \phi(s_h^\tau, a_h^\tau), \beta \rangle - \widehat{V}_{h+1}'(s_{h+1}^\tau) \right]^2 + \lambda \|\beta\|_2^2.$$

$$\widehat{\sigma}_h^2(s,a) := \max\left\{ 1, \left[\phi(s,a)^\top \widetilde{\beta}_{h,2}\right]_{[0,H^2]} - \left[\phi(s,a)^\top \widetilde{\beta}_{h,1}\right]_{[0,H]}^2 - \tilde{O}\left(\frac{dH^3}{\sqrt{N\kappa}}\right) \right\}.$$

22: **for** $h = 1, \dots, H$ **do**

23:     Compute covariance matrix $\boldsymbol{\Sigma}_h = \sum_{\tau \in \mathcal{D}} \phi(s_h^\tau, a_h^\tau) \phi(s_h^\tau, a_h^\tau)^\top / \widehat{\sigma}_h^2(s_h^\tau, a_h^\tau) + \lambda \mathbf{I_d}$.

24:     Compute weights $\widehat{w}_h = \boldsymbol{\Sigma}_h^{-1}\left(\sum_{\tau \in \mathcal{D}} \phi(s_h^\tau, a_h^\tau) \frac{r_h^\tau + \widehat{V}_{h+1}(s_{h+1}^\tau)}{\widehat{\sigma}_h^2(s_h^\tau, a_h^\tau)}\right)$.

25:     Compute pessimistic penalty $\Gamma_h(\cdot, \cdot) \leftarrow \beta_2 \|\phi(\cdot, \cdot)\|_{\boldsymbol{\Sigma}_h^{-1}}$.

26:     Compute pessimistic Q-function $\widehat{Q}_h(\cdot, \cdot) \leftarrow \left\{\phi(\cdot, \cdot)^\top \widehat{w}_h - \Gamma_h(\cdot, \cdot)\right\}_{[0, H-h+1]}$.

27:     Set $\widehat{\pi}_h(\cdot \mid \cdot) \leftarrow \arg\max_{\pi_h} \langle \widehat{Q}_h(\cdot, \cdot), \pi_h(\cdot \mid \cdot)\rangle_{\mathcal{A}}, \widehat{V}_h(\cdot) \leftarrow \langle \widehat{Q}_h(\cdot, \cdot), \widehat{\pi}_h(\cdot \mid \cdot)\rangle_{\mathcal{A}}$.

28: **end for**

29: **Output:** $\widehat{\pi}$.

---

**Algorithm 4** Hybrid Regression for Upper-Confidence Reinforcement Learning (HYRULE, Full)

1: **Input:** Offline dataset $\mathcal{D}_{\text{off}}$, samples sizes $N_{\text{on}}$, $N_{\text{off}}$, feature maps $\phi_h$. Regularization parameter $\lambda > 0$, confidence radii $\beta, \bar{\beta}, \tilde{\beta}, t_{\text{last}} = 0$.

2: **Initialize:** For $h \in [H]$, estimate $\widehat{\mathbf{w}}_{1,h}, \widecheck{\mathbf{w}}_{1,h}, Q_{1,h}, \widecheck{Q}_{1,h}, \sigma_{1,h}, \bar{\sigma}_{1,h}$ from $\mathcal{D}_{\text{off}}$ with the same formulas outlined below, and assign $\boldsymbol{\Sigma}_{0,h} = \boldsymbol{\Sigma}_{1,h} = \boldsymbol{\Sigma}_{\text{off}} + \lambda\mathbf{I} = \sum_{n=1}^{N_{\text{off}}} \bar{\sigma}_{n,h}^{-2} \phi_{n,h} \phi_{n,h}^{\top} + \lambda\mathbf{I}$.

3: **for** episodes $t = 1, ..., T$ **do**

4:      Receive the initial state $s_1^{(t)}$.

5:      **for** horizon $h = 1, ..., H$ **do**

6:          $\widehat{\mathbf{w}}_{k,h} = \boldsymbol{\Sigma}_{t,h}^{-1} \sum_{i=1}^{t-1} \bar{\sigma}_{i,h}^{-2} \phi(s_h^{(i)}, a_h^{(i)}) V_{t,h+1}(s_{h+1}^{(i)})$.

7:          $\widecheck{\mathbf{w}}_{t,h} = \boldsymbol{\Sigma}_{t,h}^{-1} \sum_{i=1}^{t-1} \bar{\sigma}_{i,h}^{-2} \phi(s_h^{(i)}, a_h^{(i)}) \widecheck{V}_{t,h+1}(s_{h+1}^{(i)})$.

8:          **if** there exists a stage $h' \in [H]$ such that $\det(\boldsymbol{\Sigma}_{t,h'}) \geqslant 2\det(\boldsymbol{\Sigma}_{t_{\text{last}},h'})$ **then**

9:              $Q_{t,h}(s,a) = \min\left\{ r_h(s,a) + \widehat{\mathbf{w}}_{t,h}^{\top} \phi(s,a) + \beta\sqrt{\phi(s,a)^{\top} \boldsymbol{\Sigma}_{t,h}^{-1} \phi(s,a)}, Q_{t-1,h}(s,a), H \right\}$.

10:             $\widecheck{Q}_{t,h}(s,a) = \max\left\{ r_h(s,a) + \widecheck{\mathbf{w}}_{t,h}^{\top} \phi(s,a) - \bar{\beta}\sqrt{\phi(s,a)^{\top} \boldsymbol{\Sigma}_{t,h}^{-1} \phi(s,a)}, \widecheck{Q}_{t-1,h}(s,a), 0 \right\}$.

11:             Set the last updating episode $t_{\text{last}} = t$.

12:          **else**

13:             $Q_{t,h}(s,a) = Q_{t-1,h}(s,a), \widecheck{Q}_{t,h}(s,a) = \widecheck{Q}_{t-1,h}(s,a)$.

14:          **end if**

15:          $V_{t,h}(s) = \max_a Q_{t,h}(s,a), \widecheck{V}_{t,h}(s) = \max_a \widecheck{Q}_{t,h}(s,a)$.

16:      **end for**

17:      **for** horizon $h = 1, ..., H$ **do**

18:          Play action $a_h^{(t)} \leftarrow \arg\max_a Q_{t,h}(s_h^{(t)}, a)$.

19:          Estimate $\sigma_{t,h} = \sqrt{\left[ \overline{\mathbb{V}}_{t,h} V_{t,h+1} \right]\left( s_h^{(t)}, a_h^{(t)} \right) + E_{t,h} + D_{t,h} + H}$, setting $E_{t,h}$ and $D_{t,h}$:

$$E_{t,h} = \min\left\{ \tilde{\beta} \left\| \boldsymbol{\Sigma}_{t,h}^{-1/2} \phi(s_h^{(t)}, a_h^{(t)}) \right\|_2, H^2 \right\} + \min\left\{ 2H\bar{\beta} \left\| \boldsymbol{\Sigma}_{t,h}^{-1/2} \phi(s_h^{(t)}, a_h^{(t)}) \right\|_2, H^2 \right\},$$

$$D_{t,h} = \min\left\{ 4d^3 H^2 \left( \widehat{\mathbf{w}}_{t,h}^{\top} \phi(s_h^{(t)}, a_h^{(t)}) - \widecheck{\mathbf{w}}_{t,h}^{\top} \phi(s_h^{(t)}, a_h^{(t)}) \right.\right.$$

$$\left.\left. + 2\bar{\beta}\sqrt{\phi(s_h^{(t)}, a_h^{(t)})^{\top} \boldsymbol{\Sigma}_{t,h}^{-1} \phi(s_h^{(t)}, a_h^{(t)})} \right), d^3 H^3 \right\}.$$

20:          $\bar{\sigma}_{t,h} \leftarrow \max\left\{ \sigma_{t,h}, \sqrt{H}, 2d^3 H^2 \left\| \phi\left( s_h^{(t)}, a_h^{(t)} \right) \right\|_{\boldsymbol{\Sigma}_{t,h}^{-1}}^{1/2} \right\}$[4].

21:          $\boldsymbol{\Sigma}_{t+1,h} = \boldsymbol{\Sigma}_{t,h} + \bar{\sigma}_{t,h}^{-2} \phi\left( s_h^{(t)}, a_h^{(t)} \right) \phi\left( s_h^{(t)}, a_h^{(t)} \right)^{\top}$.

22:          Receive reward $r_h^{(t)}$, next state $s_{h+1}^{(t)}$.

23:      **end for**

24: **end for**

25: **Output:** Greedy policy $\widehat{\pi} = \pi^{Q_{T,h}}$, $\text{Unif}(\pi^{Q_{1,h}}, ..., \pi^{Q_{T,h}})$ for PAC guarantee.

---

[4] He et al. (2023) write $\bar{\sigma}_{t,h} \leftarrow \max\{\sigma_{t,h}, H, ...\}$ instead of $\sqrt{H}$. We believe that this is a typo in their paper, given that in the proof of Lemma B.1, they state right after equation D.7 that $0 \leqslant \bar{\sigma}_{i,h}^{-1} \leqslant 1/\sqrt{H}$. Moreover, in the proof of Lemma B.5 the array of equations right after equation D.22, particularly $\left\| \bar{\sigma}_{i,h}^{-1} \phi\left( s_h^i, a_h^i \right) \right\|_2 \leqslant \left\| \phi\left( s_h^i, a_h^i \right) \right\|_2 / \sqrt{H}$, only holds true if this is $\sqrt{H}$.

# B  Proofs for Theorem 1

The proof of Theorem 1 follows from a series of distinct results, presented as three lemmas below. The first lemma demonstrates that RAPPEL achieves no higher error than LinPEVI-ADV+ itself, the second produces a $d_{\text{on}} d H^4$ error bound, while the third produces a $d_{\text{on}}^2 d H^3$ error bound via a slightly different truncation argument. We will prove Equation 7 in Lemma 2, which act as a general statistical guarantee for RAPPEL. We show the validity of the instance-dependent bound developed from Equation 7 in Lemmas 3 and 4. We observe that Theorem 1 follows immediately after.

**Lemma 2** (General Statistical Guarantee for RAPPEL, Algorithm 1). *For every $\delta \in (0,1)$ and any partition $\mathcal{X}_{\text{off}}, \mathcal{X}_{\text{on}}$, with probability at least $1 - \delta$, RAPPEL achieves*

$$V_1^*(s) - V_1^{\widehat{\pi}}(s) \lesssim \sqrt{d} \sum_{h=1}^{H} \mathbb{E}_{\pi*} ||\phi(s_h, a_h)||_{(\Sigma_{\text{off},h}^* + \Sigma_{\text{on},h}^*)^{-1}} \leqslant \sqrt{d} \sum_{h=1}^{H} \mathbb{E}_{\pi*} ||\phi(s_h, a_h)||_{\Sigma_{\text{off},h}^{*-1}}.$$

*Proof.* Before we proof the desired result, we first recall that

$$\Lambda_h = \sum_{\tau \in \mathcal{D}} \phi\left(s_h^\tau, a_h^\tau\right) \phi\left(s_h^\tau, a_h^\tau\right)^\top + I_d, \tag{10}$$

$$\Sigma_h^* = \sum_{\tau \in \mathcal{D}} \phi\left(s_h^\tau, a_h^\tau\right) \phi\left(s_h^\tau, a_h^\tau\right)^\top / \left[\mathbb{V}_h V_{h+1}^*\right]\left(s_h^\tau, a_h^\tau\right) + \lambda I_d. \tag{11}$$

Then, by invoking Theorem 2 from Xiong et al. (2023) with $N > \Omega(d^2 H^6), \lambda = 1/H^2, \beta_1 = O(\sqrt{d})$, we see that

$$V_1^*(s) - V_1^{\widehat{\pi}}(s) \lesssim \sqrt{d} \sum_{h=1}^{H} \mathbb{E}_{\pi*} \left[ \|\phi(s_h, a_h)\|_{\Sigma_h^{*-1}} \mid s_1 = s \right]$$

$$= \sqrt{d} \sum_{h=1}^{H} \mathbb{E}_{\pi*} \left[ \|\phi(s_h, a_h)\|_{(\Sigma_{\text{off},h}^* + \Sigma_{\text{on},h}^*)^{-1}} \mid s_1 = s \right],$$

as $\Sigma_h = \Sigma_{\text{off},h}^* + \Sigma_{\text{on},h}^*$. Noting that $\Sigma_{\text{on},h}^*$ is positive semi-definite, it then follows $\Sigma_{\text{off},h}^* \preceq \Sigma_{\text{off},h}^* + \Sigma_{\text{on},h}^*$. Therefore,

$$\sqrt{d} \sum_{h=1}^{H} \mathbb{E}_{\pi*} ||\phi(s_h, a_h)||_{(\Sigma_{\text{off},h}^* + \Sigma_{\text{on},h}^*)^{-1}} \leqslant \sqrt{d} \sum_{h=1}^{H} \mathbb{E}_{\pi*} ||\phi(s_h, a_h)||_{\Sigma_{\text{off},h}^{*-1}},$$

and the inequality holds. $\qquad \square$

**Lemma 3** (First Error Bound for RAPPEL, Algorithm 1). *For every $\delta \in (0,1)$ and any partition $\mathcal{X}_{\text{off}}, \mathcal{X}_{\text{on}}$, with probability at least $1 - \delta$, RAPPEL achieves*

$$V_1^*(s) - V_1^{\widehat{\pi}}(s) \lesssim \sqrt{\frac{c_{\text{off}}(\mathcal{X}_{\text{off}}) d H^4}{N_{\text{off}}}} + \sqrt{\frac{d_{\text{on}} d H^4}{N_{\text{on}}}}, \text{ where}$$

$N \geqslant \max\left\{\alpha_{\text{on}}^4 d_{\text{on}}^{-4}, \alpha_{\text{off}}^4 c_{\text{off}}(\mathcal{X}_{\text{off}})^{-4}\right\} \max\{N^*, poly(d, H, c_{\text{off}}(\mathcal{X}_{\text{off}}), \log 1/\delta)\}$, *where we define the quantities* $\alpha_{\text{off}} = \frac{N_{\text{off}}}{N}, \alpha_{\text{on}} = \frac{N_{\text{on}}}{N}$, *and the minimal samples for coverage is*

$$N^* = \min_N C \cdot N \text{ s.t. } \inf_{\Lambda \in \Omega} \max_{\phi \in \Phi} \phi^\top \left(N(\Lambda + \bar{\lambda} I) + \Lambda_{\text{off}}\right)^{-1} \phi \leqslant \tilde{O}(\max\{d_{\text{on}}/N_{\text{on}}, c_{\text{off}}(\mathcal{X}_{\text{off}})/N_{\text{off}}\}).$$

*Proof.* Let $\mathcal{X}_{\text{off}}, \mathcal{X}_{\text{on}}$ be an arbitrary partition of $\mathcal{S} \times \mathcal{A} \times [H]$. Let us leave the choice of OPTCOV tolerance unspecified for the moment, and simply assume for now that we have data $\mathcal{D}$ collected under the success event of Lemma 16.

We now invoke Theorem 2 from Xiong et al. (2023) on this dataset. As we choose $N > \Omega(d^2 H^6)$, $\lambda = 1/H^2$ and $\beta_1 = O(\sqrt{d})$, we obtain the suboptimality gap decomposition below:

$$V_1^*(s) - V_1^{\widehat{\pi}}(s) \lesssim \sqrt{d} \sum_{h=1}^{H} \mathbb{E}_{\pi*} \left[ \|\phi(s_h, a_h)\|_{\Sigma_h^{*-1}} \mid s_1 = s \right].$$

This decomposition can be further decomposed into the sum of bonuses on the offline and online partitions $\mathcal{X}_{\text{off}}$ and $\mathcal{X}_{\text{on}}$, respectively:

$$\sqrt{d} \sum_{h=1}^{H} \mathbb{E}_{\pi^*} \left[ \|\phi(s_h, a_h)\|_{\Sigma_h^{*-1}} \mid s_1 = s \right]$$

$$= \sqrt{d} \sum_{h=1}^{H} \left( \mathbb{E}_{\pi^*} \left[ \|\phi(s_h, a_h)\|_{\Sigma_h^{*-1}} \mathbb{1}_{\mathcal{X}_{\text{on}}} \mid s_1 = s \right] + \mathbb{E}_{\pi^*} \left[ \|\phi(s_h, a_h)\|_{\Sigma_h^{*-1}} \mathbb{1}_{\mathcal{X}_{\text{off}}} \mid s_1 = s \right] \right)$$

$$= \sqrt{d} \sum_{h=1}^{H} \mathbb{E}_{\pi^*} \left[ \sqrt{\phi(s_h, a_h)^\top \Sigma_h^{*-1} \phi(s_h, a_h)} \mathbb{1}_{\mathcal{X}_{\text{on}}} \mid s_1 = s \right]$$

$$+ \sqrt{d} \sum_{h=1}^{H} \mathbb{E}_{\pi^*} \left[ \sqrt{\phi(s_h, a_h)^\top \Sigma_h^{*-1} \phi(s_h, a_h)} \mathbb{1}_{\mathcal{X}_{\text{off}}} \mid s_1 = s \right].$$

We can further upper bound the above expectations under the optimal policy $\pi^*$ by taking the maximum of the quadratic form over each partition, yielding

$$\sqrt{d} \sum_{h=1}^{H} \mathbb{E}_{\pi^*} \left[ \|\phi(s_h, a_h)\|_{\Sigma_h^{*-1}} \mid s_1 = s \right]$$

$$= \sqrt{d} \sum_{h=1}^{H} \max_{\phi_h \in \Phi_{\text{on}}} \sqrt{\phi_h^\top \Sigma_h^{*-1} \phi_h} \mathbb{1}_{\mathcal{X}_{\text{on}}} + \sqrt{d} \sum_{h=1}^{H} \max_{\phi_h \in \Phi_{\text{off}}} \sqrt{\phi_h^\top \Sigma_h^{*-1} \phi_h} \mathbb{1}_{\mathcal{X}_{\text{off}}}$$

$$\leqslant \sqrt{d} \sum_{h=1}^{H} \max_{\phi_h \in \Phi_{\text{on}}} \sqrt{\phi_h^\top \Sigma_h^{*-1} \phi_h} + \sqrt{d} \sum_{h=1}^{H} \max_{\phi_h \in \Phi_{\text{off}}} \sqrt{\phi_h^\top \Sigma_h^{*-1} \phi_h}.$$

From Xiong et al. (2023), as $\left[ \mathbb{V}_h V_{h+1}^* \right](\cdot, \cdot) \in \left[ 1, H^2 \right]$, the weighted covariance matrix is uniformly upper bounded by the unweighted covariance matrix in the following manner:

$$\Sigma_h^{*-1} \preceq H^2 \Lambda_h^{-1},$$

which leads to our conclusion that

$$V_1^*(s) - V_1^{\hat{\pi}}(s) \lesssim \sqrt{d} \sum_{h=1}^{H} \max_{\phi_h \in \Phi_{\text{on}}} \sqrt{H^2 \phi_h^\top \Lambda_h^{-1} \phi_h} + \sqrt{d} \sum_{h=1}^{H} \max_{\phi_h \in \Phi_{\text{off}}} \sqrt{H^2 \phi_h^\top \Lambda_h^{-1} \phi_h}.$$

We now further bound the above two quadratic forms over the online and offline partitions respectively. By Lemma 1, the partial online coverage coefficient is bounded by the dimensionality of the online partition:

$$c_{\text{on}}(\mathcal{X}_{\text{on}}) = \inf_{\pi} \max_{\phi_h \in \Phi_{\text{on}}} \phi_h^\top \mathbb{E}_{\bar{\phi}_h \sim d_h^\pi} [\bar{\phi}_h \bar{\phi}_h^\top]^{-1} \phi_h \leqslant d_{\text{on}}.$$

As we have $N_{\text{on}}$ online episodes, the optimal covariates for online exploration would then yield

$$\inf_{\Lambda} \max_{\phi_h \in \Phi_{\text{on}}} \phi_h^\top \Lambda^{-1} \phi_h \lesssim c_{\text{on}}(\mathcal{X}_{\text{on}})/N_{\text{on}} \leqslant d_{\text{on}}/N_{\text{on}}.$$

Conversely, we also have access to $N_{\text{off}}$ episodes of offline data with the following guarantee that follows from an application of Matrix Chernoff:

$$\max_{\phi_h \in \Phi_{\text{off}}} \phi_h^\top \Lambda_{\text{off}}^{-1} \phi_h \lesssim c_{\text{off}}(\mathcal{X}_{\text{off}})/N_{\text{off}}.$$

Therefore, by Lemma 5, we can conclude that on its success event, running OPTCOV with tolerance $\tilde{O}(\max\{d_{\text{on}}/N_{\text{on}}, c_{\text{off}}(\mathcal{X}_{\text{off}})/N_{\text{off}}\})$, provides us covariates such that

$$\max_{\phi_h \in \Phi} \phi_h^\top \Lambda_h^{-1} \phi_h \lesssim \max \{c_{\text{off}}(\mathcal{X}_{\text{off}})/N_{\text{off}}, d_{\text{on}}/N_{\text{on}}\},$$

yielding the desired result.

It now remains to work out the burn-in cost from running OPTCOV. The following quantity of the minimal online samples any algorithm requires to establish coverage was first proposed in Wagenmaker and Pacchiano (2023):

$$N^* = \min_N C \cdot N \text{ s.t. } \inf_{\mathbf{\Lambda} \in \mathbf{\Omega}} \max_{\phi \in \Phi} \phi^\top \left( N(\mathbf{\Lambda} + \bar{\lambda} I) + \mathbf{\Lambda}_{\text{off}} \right)^{-1} \phi \leqslant \frac{\tilde{O}(\max\{d_{\text{on}}/N_{\text{on}}, c_{\text{off}}(\mathcal{X}_{\text{off}})/N_{\text{off}}\})}{6}.$$

We can use this as follows. Invoking Lemma 16, we see that OPTCOV incurs

$$\max\left\{ \left( \frac{N_{\text{off}}}{c_{\text{off}}(\mathcal{X}_{\text{off}})} \right)^{4/5}, \left( \frac{N_{\text{on}}}{d_{\text{on}}} \right)^{4/5} \right\} \max\{N^*, \text{poly}(d, H, c_{\text{off}}(\mathcal{X}_{\text{off}}), \log 1/\delta)\}$$

episodes of online exploration, for an overall burn-in cost of

$$N_{\text{off}} + N_{\text{on}} \geqslant \max\left\{ \frac{\alpha_{\text{on}}^4}{d_{\text{on}}^4}, \frac{\alpha_{\text{off}}^4}{c_{\text{off}}(\mathcal{X}_{\text{off}})^4} \right\} \max\{N^*, \text{poly}(d, H, c_{\text{off}}(\mathcal{X}_{\text{off}}), \log 1/\delta)\}$$

episodes, where $\alpha_{\text{off}} = \frac{N_{\text{off}}}{N_{\text{off}} + N_{\text{on}}}$ and $\alpha_{\text{on}} = \frac{N_{\text{on}}}{N_{\text{off}} + N_{\text{on}}}$.

Note that the more even the proportion of offline to online samples, the smaller $\alpha_{\text{off}}, \alpha_{\text{on}}$ are. In fact, as $\alpha_{\text{off}}^4, \alpha_{\text{on}}^4 \in [0.0625, 1]$, this term contributes no more than a constant factor that is no greater than 1 to the final sample complexity.

We then have that

$$V_1^*(s) - V_1^{\hat{\pi}}(s) \lesssim \inf_{\mathcal{X}_{\text{off}}, \mathcal{X}_{\text{on}}} \left( \sqrt{\frac{c_{\text{off}}(\mathcal{X}_{\text{off}})dH^4}{N_{\text{off}}}} + \sqrt{\frac{d_{\text{on}}dH^4}{N_{\text{on}}}} \right)$$

with probability at least $1-\delta$, when $N \geqslant \max\left\{ \frac{\alpha_{\text{on}}^4}{d_{\text{on}}^4}, \frac{\alpha_{\text{off}}^4}{c_{\text{off}}(\mathcal{X}_{\text{off}})^4} \right\} \max\{N^*, \text{poly}(d, H, c_{\text{off}}(\mathcal{X}_{\text{off}}), \log 1/\delta)\}$. □

**Lemma 4** (Second Error Bound for RAPPEL, Algorithm 1). *For every $\delta \in (0, 1)$ and any partition $\mathcal{X}_{\text{off}}, \mathcal{X}_{\text{on}}$, with probability at least $1 - \delta$, RAPPEL achieves*

$$V_1^*(s) - V_1^{\hat{\pi}}(s) \lesssim \sqrt{\frac{c_{\text{off}}(\mathcal{X}_{\text{off}})^2 dH^3}{N_{\text{off}}\alpha_{\text{off}}}} + \sqrt{\frac{d_{\text{on}}^2 dH^3}{N_{\text{on}}\alpha_{\text{on}}}}, \text{ where}$$

$N \geqslant \max\left\{ \alpha_{\text{on}}^4 d_{\text{on}}^{-4}, \alpha_{\text{off}}^4 c_{\text{off}}(\mathcal{X}_{\text{off}})^{-4} \right\} \max\{N^*, poly(d, H, c_{\text{off}}(\mathcal{X}_{\text{off}}), \log 1/\delta)\}$, *we define the quantities $\alpha_{\text{off}} = \frac{N_{\text{off}}}{N}$, $\alpha_{\text{on}} = \frac{N_{\text{on}}}{N}$, and the minimal samples for coverage is*

$$N^* = \min_N C \cdot N \text{ s.t. } \inf_{\mathbf{\Lambda} \in \mathbf{\Omega}} \max_{\phi \in \Phi} \phi^\top \left( N(\mathbf{\Lambda} + \bar{\lambda} I) + \mathbf{\Lambda}_{\text{off}} \right)^{-1} \phi \leqslant \tilde{O}(\max\{d_{\text{on}}/N_{\text{on}}, c_{\text{off}}(\mathcal{X}_{\text{off}})/N_{\text{off}}\}).$$

*Proof.* First, we set up some preliminaries. Following the same argument as the proof of Lemma 3, we can establish that, for arbitrary partition $\mathcal{X} = \mathcal{X}_{\text{on}} \cup \mathcal{X}_{\text{off}}$, we have

$$c_{\text{on}}(\mathcal{X}_{\text{on}}) \leqslant d_{\text{on}},$$

and running OPTCOV with tolerance $\tilde{O}(\max\{d_{\text{on}}/N_{\text{on}}, c_{\text{off}}(\mathcal{X}_{\text{off}})/N_{\text{off}}\})$, yields:

$$\max_{\phi_h \in \Phi} \phi_h^\top \mathbf{\Lambda}_h^{-1} \phi_h \lesssim \max\left\{ c_{\text{off}}(\mathcal{X}_{\text{off}})/N_{\text{off}}, d_{\text{on}}/N_{\text{on}} \right\}.$$

This incurs

$$\max\left\{ \left( \frac{N_{\text{off}}}{c_{\text{off}}(\mathcal{X}_{\text{off}})} \right)^{4/5}, \left( \frac{N_{\text{on}}}{d_{\text{on}}} \right)^{4/5} \right\} \max\{N^*, \text{poly}(d, H, c_{\text{off}}(\mathcal{X}_{\text{off}}), \log 1/\delta)\}$$

episodes of online exploration, for an overall burn-in cost of

$$N_{\text{off}} + N_{\text{on}} \geqslant \max\left\{ \frac{\alpha_{\text{on}}^4}{d_{\text{on}}^4}, \frac{\alpha_{\text{off}}^4}{c_{\text{off}}(\mathcal{X}_{\text{off}})^4} \right\} \max\{N^*, \text{poly}(d, H, c_{\text{off}}(\mathcal{X}_{\text{off}}), \log 1/\delta)\}$$

episodes.

To tighten the horizon dependence even further from the result of Lemma 3, we turn to the total variance lemma. i.e. Lemma C.5 in Jin et al. (2018), indicating that

$$\frac{1}{NH} \sum_{\tau \in \mathcal{D}} \sum_{h=1}^{H} \left[\mathbb{V}_h V_{h+1}^*\right](s_h^\tau, a_h^\tau) \lesssim \tilde{O}\left(H + \frac{H^2}{N}\right).$$

Then, we directly apply Lemma 17 with $\gamma = \max\{d_{\mathrm{on}}/N_{\mathrm{on}}, c_{\mathrm{off}}(\mathcal{X}_{\mathrm{off}})/N_{\mathrm{off}}\}$ and $\bar{\sigma} = H + H^2/N$, we will then obtain that

$$
\begin{aligned}
\sum_{h=1}^{H} \max_{\phi_h \in \Phi} \sqrt{\phi_h^\top \Sigma_h^{\star -1} \phi_h} &\leqslant \left(\frac{d_{\mathrm{on}}}{N_{\mathrm{on}}} + \frac{c_{\mathrm{off}}(\mathcal{X}_{\mathrm{off}})}{N_{\mathrm{off}}}\right) H \sqrt{N\left(H + \frac{H^2}{N}\right)} \\
&\leqslant \left(\frac{d_{\mathrm{on}}}{N_{\mathrm{on}}} + \frac{c_{\mathrm{off}}(\mathcal{X}_{\mathrm{off}})}{N_{\mathrm{off}}}\right) \sqrt{NH^3 + H^4} \\
&\leqslant \sqrt{\frac{c_{\mathrm{off}}(\mathcal{X}_{\mathrm{off}})^2 H^3}{N_{\mathrm{off}} \alpha_{\mathrm{off}}} + \frac{c_{\mathrm{off}}(\mathcal{X}_{\mathrm{off}})^2 H^4}{N_{\mathrm{off}}^2}} + \sqrt{\frac{d_{\mathrm{on}}^2 H^3}{N_{\mathrm{on}} \alpha_{\mathrm{on}}} + \frac{d_{\mathrm{on}}^2 H^4}{N_{\mathrm{on}}^2}} \quad (12) \\
&\lesssim \sqrt{\frac{c_{\mathrm{off}}(\mathcal{X}_{\mathrm{off}})^2 H^3}{N_{\mathrm{off}} \alpha_{\mathrm{off}}}} + \sqrt{\frac{d_{\mathrm{on}}^2 H^3}{N_{\mathrm{on}} \alpha_{\mathrm{on}}}}, \quad (13)
\end{aligned}
$$

which leads to our final result:

$$V_1^*(s) - V_1^{\widehat{\pi}}(s) \lesssim \inf_{\mathcal{X}_{\mathrm{off}}, \mathcal{X}_{\mathrm{on}}} \left(\sqrt{\frac{c_{\mathrm{off}}(\mathcal{X}_{\mathrm{off}})^2 d H^3}{N_{\mathrm{off}} \alpha_{\mathrm{off}}}} + \sqrt{\frac{d_{\mathrm{on}}^2 d H^3}{N_{\mathrm{on}} \alpha_{\mathrm{on}}}}\right),$$

where $\alpha_{\mathrm{off}} = N_{\mathrm{off}}/N$ and $\alpha_{\mathrm{on}} = N_{\mathrm{on}}/N$. $\qquad \square$

## C  Proof of Corollary 1

*Proof.* In tabular case, we set $\phi(s, a) = \mathbf{1}_{s,a}$ and $d = |\mathcal{S}| \cdot |\mathcal{A}|$. Let $N_h(s, a)$ be the number of visits to a specific state-action pair $(s, a, h)$. As the exploration algorithm OPTCOV ensures that

$$\max_{s,a,h} \frac{1}{N_h(s, a)} \leqslant \max\left(\frac{d_{\mathrm{on}}}{N_{\mathrm{on}}}, \frac{c_{\mathrm{off}}(\mathcal{X}_{\mathrm{off}})}{N_{\mathrm{off}}}\right),$$

we bound the error in the following way follows from Lemma 2,

$$
\begin{aligned}
V_1^*(s) - V_1^{\widehat{\pi}}(s) &\lesssim \sqrt{d} \sum_{h=1}^{H} \mathbb{E}_{\pi^*} \|\phi(s_h, a_h)\|_{(\Sigma_{\mathrm{off},h}^* + \Sigma_{\mathrm{on},h}^*)^{-1}} \\
&\leqslant \sqrt{|\mathcal{S}||\mathcal{A}|} \sum_{h=1}^{H} \sum_{s,a} d_h^\star(s, a) \sqrt{\frac{\left[\mathbb{V}_h V_{h+1}^*\right](s, a)}{N_h(s, a)}},
\end{aligned}
$$

where the last inequality follows from the fact that $\Sigma_h^\star = \mathrm{diag}\left(N_h(s, a) / \left[\mathbb{V}_h V_{h+1}^*\right](s, a)\right)_{s \in \mathcal{S}, a \in \mathcal{A}}$. We will then decompose the state-action space into $\mathcal{X}_{\mathrm{off}}$ and $\mathcal{X}_{\mathrm{on}}$, and bound the two parts seperately based on the tolerance level of OPTCOV,

$$
\begin{aligned}
V_1^*(s) - V_1^{\widehat{\pi}}(s) &\lesssim \sqrt{|\mathcal{S}||\mathcal{A}|} \sum_{h=1}^{H} \sum_{s,a} d_h^\star(s, a) \sqrt{\frac{\left[\mathbb{V}_h V_{h+1}^*\right](s, a)}{N_h(s, a)}} \mathbb{1}_{\mathcal{X}_{\mathrm{off}}} \\
&\quad + \sqrt{|\mathcal{S}||\mathcal{A}|} \sum_{h=1}^{H} \sum_{s,a} d_h^\star(s, a) \sqrt{\frac{\left[\mathbb{V}_h V_{h+1}^*\right](s, a)}{N_h(s, a)}} \mathbb{1}_{\mathcal{X}_{\mathrm{on}}} \\
&\leqslant \sqrt{\frac{|\mathcal{S}||\mathcal{A}| c_{\mathrm{off}}(\mathcal{X}_{\mathrm{off}})}{N_{\mathrm{off}}}} \sum_{h=1}^{H} \sum_{s,a} d_h^\star(s, a) \sqrt{\left[\mathbb{V}_h V_{h+1}^*\right](s, a)} \mathbb{1}_{\mathcal{X}_{\mathrm{off}}}
\end{aligned}
$$

$$+ \sqrt{\frac{|\mathcal{S}||\mathcal{A}|d_{\mathrm{on}}}{N_{\mathrm{on}}}} \sum_{h=1}^{H} \sum_{s,a} d_h^{\star}(s,a) \sqrt{\left[\mathbb{V}_h V_{h+1}^{*}\right](s,a)} \mathbb{1}_{\mathcal{X}_{\mathrm{on}}}$$

$$\leqslant \sqrt{|\mathcal{S}||\mathcal{A}|} \left( \sqrt{\frac{c_{\mathrm{off}}(\mathcal{X}_{\mathrm{off}})}{N_{\mathrm{off}}}} + \sqrt{\frac{d_{\mathrm{on}}}{N_{\mathrm{on}}}} \right) \sum_{h=1}^{H} \sum_{s,a} \sqrt{d_h^{\star}(s,a) \left[\mathbb{V}_h V_{h+1}^{*}\right](s,a)}.$$

As the optimal policy $\pi^{\star}$ executes a deterministic action $\pi^{\star}(s)$ for any state $s$, the inequality can be further bounded as

$$V_1^{*}(s) - V_1^{\widehat{\pi}}(s) \lesssim \sqrt{|\mathcal{S}||\mathcal{A}|} \left( \sqrt{\frac{c_{\mathrm{off}}(\mathcal{X}_{\mathrm{off}})}{N_{\mathrm{off}}}} + \sqrt{\frac{d_{\mathrm{on}}}{N_{\mathrm{on}}}} \right) \sum_{h=1}^{H} \sum_{s} \sqrt{d_h^{\star}(s,\pi^{\star}(s)) \left[\mathbb{V}_h V_{h+1}^{*}\right](s,\pi^{\star}(s))}$$

$$\leqslant \sqrt{H|\mathcal{S}|^2|\mathcal{A}|} \left( \sqrt{\frac{c_{\mathrm{off}}(\mathcal{X}_{\mathrm{off}})}{N_{\mathrm{off}}}} + \sqrt{\frac{d_{\mathrm{on}}}{N_{\mathrm{on}}}} \right) \sqrt{\sum_{h=1}^{H} \sum_{s} d_h^{\star}(s,\pi^{\star}(s)) \left[\mathbb{V}_h V_{h+1}^{*}\right](s,\pi^{\star}(s))}$$

$$\leqslant \sqrt{H|\mathcal{S}|^2|\mathcal{A}|} \left( \sqrt{\frac{c_{\mathrm{off}}(\mathcal{X}_{\mathrm{off}})}{N_{\mathrm{off}}}} + \sqrt{\frac{d_{\mathrm{on}}}{N_{\mathrm{on}}}} \right) \sqrt{\sum_{h=1}^{H} \mathbb{E}_{(s,a)\sim d_{\pi^{\star}}} \left[\mathbb{V}_h V_{h+1}^{*}\right](s,a)}$$

$$\leqslant \sqrt{H^3|\mathcal{S}|^2|\mathcal{A}|} \left( \sqrt{\frac{c_{\mathrm{off}}(\mathcal{X}_{\mathrm{off}})}{N_{\mathrm{off}}}} + \sqrt{\frac{d_{\mathrm{on}}}{N_{\mathrm{on}}}} \right), \tag{14}$$

where the last inequality follows from the proof of Lemma C.5. in Jin et al. (2018). $\qquad \square$

## D   On concentrability and coverability

**Lemma 1.** *For any partition $\mathcal{X}_{\mathrm{off}}, \mathcal{X}_{\mathrm{on}}$, we have that $c_{\mathrm{on}}(\mathcal{X}_{\mathrm{on}}) \leqslant d_{\mathrm{on}}$. Similarly, there exists a partition such that $c_{\mathrm{off}}(\mathcal{X}_{\mathrm{off}}) = O(d)$.*

*Proof.* This proof follows a similar strategy to that of Lemma B.10 in Wagenmaker and Jamieson (2023), except that we exploit the projections onto $d_{\mathrm{on}}$ to get a bound that depends on $d_{\mathrm{on}} \leqslant d$, instead of $d$. We wish to bound

$$c_{\mathrm{on}}(\mathcal{X}_{\mathrm{on}}) = \inf_{\pi} \max_{h} \frac{1}{\lambda_{d_{\mathrm{on}}}(\mathbb{E}_{d_h^{\pi}}[(\mathcal{P}_{\mathrm{on}}\phi_h)(\mathcal{P}_{\mathrm{on}}\phi_h)^{\top}])}.$$

$\mathcal{P}_{\mathrm{on}} \in \mathbb{R}^{d \times d}$ has rank $d_{\mathrm{on}} \leqslant d$, so we can decompose this with the thin SVD into $\mathcal{P}_{\mathrm{on}} = U_{\mathrm{on}} U_{\mathrm{on}}^{\top}$, where $U_{\mathrm{on}} \in \mathbb{R}^{d \times d_{\mathrm{on}}}$. It then holds that

$$\lambda_{d_{\mathrm{on}}}(\mathbb{E}_{d_h^{\pi}}[(\mathcal{P}_{\mathrm{on}}\phi_h)(\mathcal{P}_{\mathrm{on}}\phi_h)^{\top}]) = \lambda_{\min}(\mathbb{E}_{d_h^{\pi}}[(U_{\mathrm{on}}^{\top}\phi_h)(U_{\mathrm{on}}^{\top}\phi_h)^{\top}]),$$

and from Lemma 20 that

$$c_{\mathrm{on}}(\mathcal{X}_{\mathrm{on}}) = \inf_{\pi} \sup_{v_h \in \Phi_{\mathrm{on}}} v_h^{\top} U_{\mathrm{on}} E_{d_h^{\pi}}[(U_{\mathrm{on}}^{\top}\phi_h)(U_{\mathrm{on}}^{\top}\phi_h)^{\top}]^{-1} U_{\mathrm{on}}^{\top} v_h.$$

Apply Jensen's inequality to find that for any $v_h \in \Phi_{\mathrm{on}}$,

$$v_h^{\top} U_{\mathrm{on}} E_{d_h^{\pi}}[(U_{\mathrm{on}}^{\top}\phi_h)(U_{\mathrm{on}}^{\top}\phi_h)^{\top}] U_{\mathrm{on}}^{\top} v_h \geqslant v_h^{\top} U_{\mathrm{on}} \mathbb{E}_{\phi_h \sim d_h^{\pi}}[U_{\mathrm{on}}^{\top}\phi_h] \mathbb{E}_{\phi_h \sim d_h^{\pi}}[U_{\mathrm{on}}^{\top}\phi_h]^{\top} U_{\mathrm{on}}^{\top} v_h.$$

Then, we can bound

$$c_{\mathrm{on}}(\mathcal{X}_{\mathrm{on}}) = \inf_{\pi} \sup_{v_h \in \Phi_{\mathrm{on}}} v_h^{\top} U_{\mathrm{on}} E_{d_h^{\pi}}[(U_{\mathrm{on}}^{\top}\phi_h)(U_{\mathrm{on}}^{\top}\phi_h)^{\top}]^{-1} U_{\mathrm{on}}^{\top} v_h$$

$$\leqslant \inf_{\rho} \sup_{v_h \in \Phi_{\mathrm{on}}} v_h^{\top} U_{\mathrm{on}} \left( \mathbb{E}_{\pi \sim \rho} \left[ \mathbb{E}_{\phi_h \sim d_h^{\pi}}[U_{\mathrm{on}}^{\top}\phi_h] E_{\phi_h \sim d_h^{\pi}}[U_{\mathrm{on}}^{\top}\phi_h^{\top}] \right] \right)^{-1} U_{\mathrm{on}}^{\top} v_h.$$

By Kiefer-Wolfowitz (Lattimore et al., 2020), this is bounded by $d_{\mathrm{on}}$.

Similarly,

$$\inf_{\mathcal{X}_{\text{off}}, \mathcal{X}_{\text{on}}} c_{\text{off}}(\mathcal{X}_{\text{off}}) = \inf_{\mathcal{X}_{\text{off}}, \mathcal{X}_{\text{on}}} \max_h \frac{1}{\lambda_{d_{\text{off}}}(\mathbb{E}_{\mu_h}[(\mathcal{P}_{\text{off}}\phi_h)(\mathcal{P}_{\text{off}}\phi_h)^\top])}$$

$$= \inf_{\mathcal{X}_{\text{off}}, \mathcal{X}_{\text{on}}} \max_h \frac{1}{\lambda_{\min}(\mathbb{E}_{\mu_h}[(U_{\text{off}}^\top\phi_h)(U_{\text{off}}^\top\phi_h)^\top])}$$

$$\leqslant O(d).$$

where the upper bound is achieved when, for instance, we choose $\mathcal{X}_{\text{off}}$ such that $\Phi_{\text{off}} =$ Span $\left((v_{h,1}, ..., v_{h,k_h})_{h \in [H]}\right)$, where $v_{h,i}$ is the $i$-th largest eigenvector of $\mathbb{E}_\mu[\phi_h\phi_h^\top] \approx \frac{1}{N_{\text{off}}} \sum_{\tau \in \mathcal{D}_{\text{off}}} \phi_h(s_h^\tau, a_h^\tau)\phi_h(s_h^\tau, a_h^\tau)^\top$, and $v_{h,k_h}$ is the eigenvector corresponding to the largest eigenvalue $\lambda_{h,k_h} \geqslant \Omega(1/k_h)$. The largest eigenvalue $\lambda_{h,1}$ is always $\Omega(1/d)$ for non-null features, so there always exists such a partition where $d_{\text{off}}$ is at least 1.

$\square$

Informally, one can choose the offline partition to be the span of the large eigenvectors of the covariance matrix, so the smallest eigenvalue of the projected covariance matrix, i.e. the partial all policy concentrability coefficient, is no larger than the dimension of the partition.

**Lemma 5** (Maximum Eigenvalue Bound with OPTCOV). *On any partition $\mathcal{X}_{\text{off}}, \mathcal{X}_{\text{on}}$, if we run OPTCOV with tolerance $\tilde{O}(\max\{d_{\text{on}}/N_{\text{on}}, c_{\text{off}}(\mathcal{X}_{\text{off}})/N_{\text{off}}\})$, on this partition we also have that*

$$\max_{\phi_h \in \Phi} \phi_h^\top \Lambda_h^{-1} \phi_h \lesssim \max\{c_{\text{off}}(\mathcal{X}_{\text{off}})/N_{\text{off}}, d_{\text{on}}/N_{\text{on}}\}.$$

*Proof.* By Lemma 1, for any partition, we have that

$$c_{\text{on}}(\mathcal{X}_{\text{on}}) = \inf_\pi \max_{\phi_h \in \Phi_{\text{on}}} \phi_h^\top \mathbb{E}_{\bar{\phi}_h \sim d_h^\pi}[\bar{\phi}_h \bar{\phi}_h^\top]^{-1} \phi_h \leqslant d_{\text{on}},$$

Applying Matrix Chernoff, we have that with probability at least $1 - \delta$,

$$\max_{\phi_h \in \Phi_{\text{off}}} \phi_h^\top \Lambda_{h,\text{off}}^{-1} \phi_h \leqslant \max_{\phi_h \in \Phi_{\text{off}}} \phi_h^\top \mathbb{E}_{\bar{\phi}_h \sim \mu_h}[\bar{\phi}_h \bar{\phi}_h^\top + N_{\text{off}}^{-1}\mathbf{I}]^{-1}\phi_h N_{\text{off}}^{-1}\left(1 - \sqrt{\frac{2}{N_{\text{off}}}\log\left(\frac{4d}{\delta}\right)}\right)^{-1},$$

and similarly for $c_{\text{on}}(\mathcal{X}_{\text{on}})$ we also have that

$$\inf_\pi \max_{\phi_h \in \Phi_{\text{on}}} \phi_h^\top \Lambda_{h,\pi}^{-1} \phi_h \leqslant \inf_\pi \max_{\phi_h \in \Phi_{\text{on}}} \phi_h^\top \mathbb{E}_{\bar{\phi}_h \sim \mu_h}[\bar{\phi}_h \bar{\phi}_h^\top]^{-1}\phi_h N_{\text{on}}^{-1}\left(1 - \sqrt{\frac{2}{N_{\text{on}}}\log\left(\frac{4d}{\delta}\right)}\right)^{-1}.$$

As $\Lambda_{h,\text{off}} + \Lambda_{h,\text{on}} = \Lambda_h$, we have

$$\max_{\phi_h \in \Phi} \phi_h^\top \Lambda_h^{-1}\phi_h = \max\left\{\max_{\phi_h \in \Phi_{\text{off}}} \phi_h^\top \Lambda_h^{-1}\phi_h, \max_{\phi_h \in \Phi_{\text{on}}} \phi_h^\top \Lambda_h^{-1}\phi_h\right\}$$

$$\lesssim \max\left\{c_{\text{off}}(\mathcal{X}_{\text{off}})/N_{\text{off}}, \max_{\phi_h \in \Phi_{\text{on}}} \phi_h^\top \Lambda_h^{-1}\phi_h\right\},$$

where the last step follows from the choice of partition. So it suffices to run OPTCOV with tolerance $\tilde{O}(\max\{d_{\text{on}}/N_{\text{on}}, c_{\text{off}}(\mathcal{X}_{\text{off}})/N_{\text{off}}\})$,

to find that there exists at least one partition such that

$$\max_{\phi_h \in \Phi} \phi_h^\top \Lambda_h^{-1}\phi_h \lesssim \max\{c_{\text{off}}(\mathcal{X}_{\text{off}})/N_{\text{off}}, d_{\text{on}}/N_{\text{on}}\}.$$

$\square$

**Lemma 6** (Coverability Coefficient Is Bounded In Tabular MDPs). *If the underlying MDP is tabular, for any partition $\mathcal{X}_{\text{off}}, \mathcal{X}_{\text{on}}$, we have that $c_{\text{on}}(\mathcal{X}_{\text{on}}) \leqslant d_{\text{on}}$.*

*Proof.* First, we write the concentrability coefficient in terms of densities.

$$c_{\text{on}}(\mathcal{X}_{\text{on}}) = \min_{\pi} \max_{h} \frac{1}{\lambda_{d_{\text{on}}}(\mathbb{E}_{d_h^{\pi}}[(\mathcal{P}_{\text{on}}\phi_h)(\mathcal{P}_{\text{on}}\phi_h)^{\top}])}$$

$$\leqslant \min_{\pi} \max_{h} \frac{\mathbb{1}_{\mathcal{X}_{on}}}{\min_{s,a} d_h^{\pi}(s,a)\mathbb{1}_{\mathcal{X}_{on}}}$$

$$\leqslant \min_{\pi} \max_{h,s,a} \frac{\mathbb{1}_{\mathcal{X}_{on}}}{d_h^{\pi}(s,a)\mathbb{1}_{\mathcal{X}_{on}}}.$$

By the same trick that Xie et al. (2022a) use in their Lemma 3,

$$\frac{\mathbb{1}_{\mathcal{X}_{\text{on}}}}{d_h^{\pi}(s,a)\mathbb{1}_{\mathcal{X}_{\text{on}}}} \leqslant \frac{\mathbb{1}_{\mathcal{X}_{\text{on}}}}{\sup_{\pi''} d_h^{\pi''}(s,a)\mathbb{1}_{\mathcal{X}_{\text{on}}} / \sum_{s',a'} \sup_{\pi'} d_h^{\pi'}(s',a')\,\mathbb{1}_{\mathcal{X}_{\text{on}}}}$$

$$\leqslant \frac{\sum_{s,a} \sup_{\pi} d_h^{\pi}(s,a)\,\mathbb{1}_{\mathcal{X}_{\text{on}}}}{\sup_{\pi} d_h^{\pi}(s,a)\,\mathbb{1}_{\mathcal{X}_{\text{on}}}}$$

$$\leqslant d_{\text{on}}.$$

$\square$

# E  Proofs for Algorithm 2

## E.1  Setup

We consider the same state-action space splitting framework of Tan and Xu (2024). Let $\mathcal{X}_{\text{on}} \cup \mathcal{X}_{\text{off}} = [H] \times \mathcal{S} \times \mathcal{A}$. Then, their images under the feature map $\Phi_{\text{off}} = \text{Span}(\phi(\mathcal{X}_{\text{off},h}))_{h\in[H]} \subseteq \mathbb{R}^d$ and $\Phi_{\text{on}} = \text{Span}(\phi(\mathcal{X}_{\text{on},h}))_{h\in[H]} \subseteq \mathbb{R}^d$ are subspaces of $\mathcal{X}$ with dimension $d_{\text{off}}$ and $d_{\text{on}}$, respectively. We denote $\mathcal{P}_{\text{off}}, \mathcal{P}_{\text{on}}$ as the orthogonal projection operators onto these subspaces respectively. The partial offline all-policy concentrability coefficient

$$c_{\text{off}}(\mathcal{X}_{\text{off}}) = \max_{h} \frac{1}{\lambda_{d_{\text{off}}}(\mathbb{E}_{\mu_h}[(\mathcal{P}_{\text{off}}\phi_h)(\mathcal{P}_{\text{off}}\phi_h)^{\top}])},$$

is bounded by the inverse of the $d_{\text{off}}$-th largest eigenvalue of the covariance matrix of the projected feature maps onto the offline partition, where $\lambda_k$ is the $k$-th largest eigenvalue. Write $\mathbb{1}_{\mathcal{X}_{\text{on}}}$ as shorthand for $\mathbb{1}((s,a,h) \in \mathcal{X}_{\text{on}})$, and similarly for $\mathbb{1}_{\mathcal{X}_{\text{off}}}$.

Now, we work through the analysis of He et al. (2023) to ensure that their result holds in our setting, where the regret decomposes into online part $\|\mathbf{\Sigma}_{t,h}^{-1/2}\phi_h(s_h^{(t)}, a_h^{(t)})\mathbb{1}_{\mathcal{X}_{\text{on}}}\|_2$ and offline part $\|\mathbf{\Sigma}_{t,h}^{-1/2}\phi_h(s_h^{(t)}, a_h^{(t)})\mathbb{1}_{\mathcal{X}_{\text{off}}}\|_2$ respectively, instead of $\|\mathbf{\Sigma}_{t,h}^{-1/2}\phi_h(s_h^{(t)}, a_h^{(t)})\|_2$.

## E.2  High-probability events

We define several "high probability" events which are similar to those defined in He et al. (2023).

- We define $\widetilde{w}_{t,h}$ as the solution of the weighted ridge regression problem for the squared value function

$$\widetilde{w}_{t,h} = \mathbf{\Sigma}_{t,h}^{-1} \sum_{i=1}^{t-1} \bar{\sigma}_{i,h}^{-2}\phi(s_h^{(i)}, a_h^{(i)})V_{t,h+1}^2(s_{h+1}^{(i)}). \tag{15}$$

- We define $\mathcal{E}$ as the event where the following inequalities hold for all $s, a, t, h \in \mathcal{S} \times \mathcal{A} \times [T] \times [H]$:

$$\left|\widehat{\mathbf{w}}_{t,h}^{\top}\phi(s,a) - [\mathbb{P}_h V_{t,h+1}](s,a)\right| \leqslant \bar{\beta}\sqrt{\phi(s,a)^{\top}\mathbf{\Sigma}_{t,h}^{-1}\phi(s,a)}, \tag{16}$$

$$\left|\widetilde{\mathbf{w}}_{t,h}^{\top}\phi(s,a) - [\mathbb{P}_h V_{t,h+1}^2](s,a)\right| \leqslant \widetilde{\beta}\sqrt{\phi(s,a)^{\top}\mathbf{\Sigma}_{t,h}^{-1}\phi(s,a)}, \tag{17}$$

$$\left| \breve{\mathbf{w}}_{t,h}^\top \phi(s,a) - \left[ \mathbb{P}_h \breve{V}_{t,h+1} \right](s,a) \right| \leqslant \bar{\beta}\sqrt{\phi(s,a)^\top \mathbf{\Sigma}_{t,h}^{-1} \phi(s,a)}, \tag{18}$$

$$\widetilde{\beta} = O\left( H^2\sqrt{d\lambda} + \sqrt{d^3 H^4 \log^2(dHN/(\delta\lambda))} \right), \bar{\beta} = O\left( H\sqrt{d\lambda} + \sqrt{d^3 H^2 \log^2(dHN/(\delta\lambda))} \right).$$

This is the "coarse event" as mentioned in their paper, where concentration holds for the value and squared value function with all three estimators.

- We define $\widetilde{\mathcal{E}}_h$ as the event that for all episodes $t \in [T]$, stages $h \leqslant h' \leqslant H$ and state-action pairs $(s,a) \in \mathcal{S} \times \mathcal{A}$, the weight vector $\widehat{\mathbf{w}}_{t,h}$ satisfies

$$\left| \widehat{\mathbf{w}}_{t,h'}^\top \phi(s,a) - \left[ \mathbb{P}_h V_{t,h'+1} \right](s,a) \right| \leqslant \beta\sqrt{\phi(s,a)^\top \mathbf{\Sigma}_{t,h'}^{-1} \phi(s,a)}, \tag{19}$$

where

$$\beta = O\left( H\sqrt{d\lambda} + \sqrt{d\log^2(1 + dNH/(\delta\lambda))} \right).$$

Furthermore, let $\widetilde{\mathcal{E}} = \widetilde{\mathcal{E}}_1$ denotes the event that (19) holds for all stages $h \in [H]$. This is the fine event where concentration for $\widehat{w}$ is tighter than that required in (16) to (18).

Equipped with these definitions, we recall the following lemmas from He et al. (2023):

**Lemma 7** (Lemma B.1, He et al. (2023)). *$\mathcal{E}$ holds with probability at least $1 - 7\delta$.*

**Lemma 8** (Lemma B.2, He et al. (2023)). *On the event $\mathcal{E}$ and $\widetilde{\mathcal{E}}_{h+1}$, for each episode $t \in [T]$ and stage $h$, the estimated variance satisfies*

$$\left| \left[ \overline{\mathbb{V}}_h V_{t,h+1} \right]\left( s_h^{(t)}, a_h^{(t)} \right) - \left[ \mathbb{V}_h V_{t,h+1} \right]\left( s_h^{(t)}, a_h^{(t)} \right) \right| \leqslant E_{t,h},$$

$$\left| \left[ \overline{\mathbb{V}}_h V_{t,h+1} \right]\left( s_h^{(t)}, a_h^{(t)} \right) - \left[ \mathbb{V}_h V_{h+1}^* \right]\left( s_h^{(t)}, a_h^{(t)} \right) \right| \leqslant E_{t,h} + D_{t,h}.$$

**Lemma 9** (Lemma B.3, He et al. (2023)). *On the event $\mathcal{E}$ and $\widetilde{\mathcal{E}}_{h+1}$, for any episode $t$ and $i > t$, we have*

$$\left[ \mathbb{V}_h\left( V_{i,h+1} - V_{h+1}^* \right) \right]\left( s_h^{(t)}, a_h^{(t)} \right) \leqslant D_{t,h}/\left( d^3 H \right).$$

**Lemma 10** (Lemma B.4, He et al. (2023)). *On the event $\mathcal{E}$ and $\widetilde{\mathcal{E}}_h$, for all episodes $t \in [T]$ and stages $h \leqslant h' \leqslant H$, we have $Q_{t,h}(s,a) \geqslant Q_h^*(s,a) \geqslant \breve{Q}_{t,h}(s,a)$. In addition, we have $V_{t,h}(s) \geqslant V_h^*(s) \geqslant \breve{V}_{t,h}(s)$.*

**Lemma 11** (Lemma B.5, He et al. (2023)). *On event $\mathcal{E}$, event $\widetilde{\mathcal{E}}$ holds with probability at least $1 - \delta$.*

### E.3 Regret decomposition

From He et al. (2023), based on Lemma B.4 of their paper, $Q_{t,h}(s_h^{(t)}, a_h^{(t)}) = V_{t,h}(s_h^{(t)}) \geqslant V_h^*(s_h^{(t)})$, i.e. optimism holds for all episodes and timesteps. Therefore,

$$\mathrm{Reg}(T) \lesssim \sum_{t=1}^T \sum_{h=1}^H \left\{ \left[ \mathbb{P}_h\left( V_{t,h+1} - V_{t,h+1}^{\pi^{(t)}} \right) \right]\left( s_h^{(t)}, a_h^{(t)} \right) - \left( V_{t,h+1}\left( s_{h+1}^{(t)} \right) - V_{t,h+1}^{\pi^{(t)}}\left( s_{h+1}^{(t)} \right) \right) \right\}$$
$$+ \beta \sum_{t=1}^T \sum_{h=1}^H \|\mathbf{\Sigma}_{t,h}^{-1/2} \phi_h(s_h^{(t)}, a_h^{(t)})\|_2.$$

Accordingly, given a partition $\mathcal{X}_{\mathrm{off}}, \mathcal{X}_{\mathrm{on}}$ of $[H] \times \mathcal{S} \times \mathcal{A}$, we can further decompose this into the fraction of episodes where each partition is visited,

$$\sum_{t=1}^T \sum_{h=1}^H \|\mathbf{\Sigma}_{t,h}^{-1/2} \phi_h(s_h^{(t)}, a_h^{(t)})\|_2 = \sum_{h,t} \|\mathbf{\Sigma}_{t,h}^{-1/2} \phi_h(s_h^{(t)}, a_h^{(t)})\mathbb{1}_{\mathcal{X}_{\mathrm{off}}}\|_2 + \sum_{h,t} \|\mathbf{\Sigma}_{t,h}^{-1/2} \phi_h(s_h^{(t)}, a_h^{(t)})\mathbb{1}_{\mathcal{X}_{\mathrm{on}}}\|_2.$$

He et al. (2023) define the events

$$\mathcal{E}_1 = \left\{ \forall h \in [H], \sum_{t=1}^{T} \sum_{h'=h}^{H} \left[ \mathbb{P}_h \left( V_{t,h+1} - V_{t,h+1}^{\pi^{(t)}} \right) \right] \left( s_h^{(t)}, a_h^{(t)} \right) \right.$$
$$\left. - \sum_{t=1}^{T} \sum_{h'=h}^{H} \left( V_{t,h+1} \left( s_{h+1}^{(t)} \right) - V_{t,h+1}^{\pi^{(t)}} \left( s_{h+1}^{(t)} \right) \right) \leqslant 2\sqrt{2H^3 T \log(H/\delta)} \right\},$$

$$\mathcal{E}_2 = \left\{ \forall h \in [H], \sum_{t=1}^{T} \sum_{h'=h}^{H} \left[ \mathbb{P}_h \left( V_{t,h+1} - \check{V}_{t,h+1} \right) \right] \left( s_h^{(t)}, a_h^{(t)} \right) \right.$$
$$\left. - \sum_{t=1}^{T} \sum_{h'=h}^{H} \left( V_{t,h+1} \left( s_{h+1}^{(t)} \right) - \check{V}_{t,h+1} \left( s_{h+1}^{(t)} \right) \right) \leqslant 2\sqrt{2H^3 T \log(H/\delta)} \right\},$$

which they show that by Azuma-Hoeffding, both hold with probability $1 - \delta$ each. As such, we have that

$$\mathrm{Reg}(T) \lesssim \sqrt{H^3 T \log(H/\delta)} + \sum_{h,t} \beta \|\mathbf{\Sigma}_{t,h}^{-1/2} \phi_h(s_h^{(t)}, a_h^{(t)}) \mathbb{1}_{\mathcal{X}_{\mathrm{off}}}\|_2 + \sum_{h,t} \beta \|\mathbf{\Sigma}_{t,h}^{-1/2} \phi_h(s_h^{(t)}, a_h^{(t)}) \mathbb{1}_{\mathcal{X}_{\mathrm{on}}}\|_2.$$

Here, we denote

$$\mathrm{Reg}_{\mathrm{off}}(T) = \sum_{h,t} \beta \|\mathbf{\Sigma}_{t,h}^{-1/2} \phi_h(s_h^{(t)}, a_h^{(t)}) \mathbb{1}_{\mathcal{X}_{\mathrm{off}}}\|_2, \qquad \mathrm{Reg}_{\mathrm{on}}(T) = \sum_{h,t} \beta \|\mathbf{\Sigma}_{t,h}^{-1/2} \phi_h(s_h^{(t)}, a_h^{(t)}) \mathbb{1}_{\mathcal{X}_{\mathrm{on}}}\|_2,$$

as the offline regret and online regret, respectively.

### E.4 Offline regret control

Now, we bound the regret on the offline partition. We first perform a similar argument to that in Tan and Xu (2024); Xie et al. (2022a) to show that the sum of bonuses can be controlled by the maximum eigenvalue of the inverse weighted average covariance matrix in Lemma 12. We will then show that the maximum eigenvalue can be nicely bounded in Lemma 13.

**Lemma 12** (Sum of Bonuses on Offline Partition). *For any partition $\mathcal{X}_{\mathrm{off}}, \mathcal{X}_{\mathrm{on}}$, we can bound the sum of bonuses on the offline partition with the following:*

$$Reg_{\mathrm{off}}(T) \lesssim \sum_{h=1}^{H} \sqrt{\frac{dN_{\mathrm{on}}^2}{N_{\mathrm{off}}} \max_{\phi_h \in \Phi_{\mathrm{off}}} \phi_h^\top \bar{\mathbf{\Sigma}}_{\mathrm{off},h}^{-1} \phi_h},$$

*where $\bar{\mathbf{\Sigma}}_{\mathrm{off},h} = (\mathbf{\Sigma}_{\mathrm{off},h} + \lambda \mathbf{I})/N_{\mathrm{off}}$ and $T = N_{\mathrm{on}}$.*

*Proof.* It is sufficient to show the following holds true

$$\sum_{t} \beta \|\mathbf{\Sigma}_{t,h}^{-1/2} \phi_h(s_h^{(t)}, a_h^{(t)}) \mathbb{1}_{\mathcal{X}_{\mathrm{off}}}\|_2 \leqslant \sqrt{\frac{dN_{\mathrm{on}}^2}{N_{\mathrm{off}}} \max_{\phi_h \in \Phi_{\mathrm{off}}} \phi_h^\top \bar{\mathbf{\Sigma}}_{\mathrm{off},h}^{-1} \phi_h},$$

then the desired inequality directly follows. With a direct calculation, one may observe that

$$\|\mathbf{\Sigma}_{t,h}^{-1/2} \phi_h(s_h^{(t)}, a_h^{(t)}) \mathbb{1}_{\mathcal{X}_{\mathrm{off}}}\|_2 = \sqrt{\phi_h^\top(s_h^{(t)}, a_h^{(t)}) \mathbf{\Sigma}_{t,h}^{-1} \phi_h(s_h^{(t)}, a_h^{(t)}) \mathbb{1}_{\mathcal{X}_{\mathrm{off}}}}$$
$$\lesssim \sqrt{\phi_h^\top(s_h^{(t)}, a_h^{(t)}) (\mathbf{\Sigma}_{\mathrm{off},h} + \lambda \mathbf{I})^{-1} \phi_h(s_h^{(t)}, a_h^{(t)}) \mathbb{1}_{\mathcal{X}_{\mathrm{off}}}},$$

where the last inequality holds as $\Sigma_{\mathrm{off},h} \preceq \Sigma_{t,h}$. As a result, we are able to bound the desired inequality with the maximum eigenvalue of the inverse weighted matrix,

$$\sum_{t} \|\mathbf{\Sigma}_{t,h}^{-1/2} \phi_h(s_h^{(t)}, a_h^{(t)}) \mathbb{1}_{\mathcal{X}_{\mathrm{off}}}\|_2 \leqslant N_{\mathrm{on}} \sqrt{\max_{\phi_h \in \Phi_{\mathrm{off}}} \phi_h^\top (\mathbf{\Sigma}_{\mathrm{off},h} + \lambda \mathbf{I})^{-1} \phi_h}$$
$$= \sqrt{N_{\mathrm{on}} \frac{N_{\mathrm{on}}}{N_{\mathrm{off}}} \max_{\phi_h \in \Phi_{\mathrm{off}}} \phi_h^\top \bar{\mathbf{\Sigma}}_{\mathrm{off},h}^{-1} \phi_h},$$

where $\bar{\boldsymbol{\Sigma}}_{\text{off},h} = (\boldsymbol{\Sigma}_{\text{off},h} + \lambda\mathbf{I})/N_{\text{off}}$. As $\beta = \tilde{O}(\sqrt{d})$, we obtain the bound we desired:

$$\sum_t \beta\|\boldsymbol{\Sigma}_{t,h}^{-1/2}\phi_h(s_h^{(t)}, a_h^{(t)})\mathbb{1}_{\mathcal{X}_{\text{off}}}\|_2 \leqslant \sqrt{dN_{\text{on}}\frac{N_{\text{on}}}{N_{\text{off}}}\max_{\phi_h \in \Phi_{\text{off}}} \phi_h^\top \bar{\boldsymbol{\Sigma}}_{\text{off},h}^{-1}\phi_h}.$$

$\square$

**Lemma 13** (Partial Concentrability Bound). *For any partition $\mathcal{X}_{\text{off}}, \mathcal{X}_{\text{on}}$, we have that*

$$\sum_{h=1}^{H} \max_{\phi_h \in \Phi_{\text{off}}} \sqrt{\phi_h^\top \bar{\boldsymbol{\Sigma}}_{\text{off},h}^{-1}\phi_h} \lesssim \sqrt{c_{\text{off}}(\mathcal{X}_{\text{off}})^2 H^3},$$

*when $N_{\text{on}}, N_{\text{off}} \geqslant \tilde{\Omega}(d^{13}H^{14})$, where we define $\bar{\boldsymbol{\Sigma}}_{\text{off},h} = (\boldsymbol{\Sigma}_{\text{off},h} + \lambda\mathbf{I})/N_{\text{off}}$.*

*Proof.* Similar to the definition of $\bar{\boldsymbol{\Sigma}}_{\text{off},h}$, we define $\bar{\boldsymbol{\Lambda}}_{\text{off},h} = (\boldsymbol{\Lambda}_{\text{off},h} + \lambda\mathbf{I})/N_{\text{off}}$ in a similar way. Then, one may observe that

$$\max_{\phi_h \in \Phi_{\text{off}}} \left(\phi_h^\top \bar{\boldsymbol{\Lambda}}_{\text{off},h}^{-1}\phi_h\right) = \max_{\phi_h \in \Phi_{\text{off}}} \left(\phi_h^\top \left(\frac{1}{N_{\text{off}}}\left(\sum_{n=1}^{N_{\text{off}}} \phi_{n,h}\phi_{n,h}^\top + \lambda\mathbf{I}\right)\right)^{-1}\phi_h\right)$$

$$\leqslant \max_{\phi_h \in \Phi_{\text{off}}} \phi_h^\top \mathbb{E}_{\mu_h}[\bar{\boldsymbol{\Lambda}}_{\text{off},h}]^{-1}\phi_h \left(1 - \sqrt{\frac{2}{N_{\text{off}}}\log\left(\frac{4d}{\delta}\right)}\right)^{-1},$$

where the last line holds by an application of the Matrix Chernoff inequality. Then, we may further bound the quantity with the partial offline all-policy concentrability coefficient,

$$\max_{\phi_h \in \Phi_{\text{off}}} \left(\phi_h^\top \bar{\boldsymbol{\Lambda}}_{\text{off},h}^{-1}\phi_h\right) \lesssim \inf_{\mathcal{X}_{\text{off}},\mathcal{X}_{\text{on}}} \max_h \frac{1}{\lambda_{d_{\text{off}}} (\mathbb{E}_\mu(\mathcal{P}_{\text{off}}\phi_h)(\mathcal{P}_{\text{off}}\phi_h)^\top)}$$

$$= \inf_{\mathcal{X}_{\text{off}},\mathcal{X}_{\text{on}}} \max_h \frac{1}{\lambda_{\min} (\mathbb{E}_\mu(U_{\text{off}}^\top\phi_h)(U_{\text{off}}^\top\phi_h)^\top)}$$

$$= c_{\text{off}}(\mathcal{X}_{\text{off}}).$$

To tighten the dependence of the regret of the offline partition on $H$, we again employ a truncation argument that used in Lemma 4. Recall that in Section B of the appendix in He et al. (2023), by the total variance lemma of Jin et al. (2019), it holds that

$$\sum_{t=1}^{T}\sum_{h=1}^{H} \sigma_{t,h}^2 \leqslant \tilde{O}\left(H^2T + d^{10.5}H^{16}\right).$$

Again, recall that we have

$$\sum_{h,t} \|\boldsymbol{\Sigma}_{t,h}^{-1/2}\phi_h(s_h^{(t)}, a_h^{(t)})\mathbb{1}_{\mathcal{X}_{\text{off}}}\|_2$$

$$\lesssim \sqrt{H^2 N_{\text{on}}\frac{N_{\text{on}}}{N_{\text{off}}}\max_{\phi_h \in \Phi_{\text{off}}} \left(\phi_h^\top \left(\frac{1}{N_{\text{off}}}\left(\sum_{n=1}^{N_{\text{off}}} \bar{\sigma}_{n,h}^{-2}\phi_{n,h}\phi_{n,h}^\top + \lambda\mathbf{I}\right)\right)^{-1}\phi_h\right)}.$$

As $\bar{\sigma}_{n,h}^2 = \max\left\{\sigma_{n,h}^2, H, 4d^6H^4\|\phi_{n,h}\|_{\Sigma_{n,h}^{-1}}\right\}$. Consider the sets

$$\mathcal{I}_1 = \left\{n \in [N_{\text{off}}] : \forall h : \bar{\sigma}_{n,h}^2 = \max(\sigma_{n,h}^2, H)\right\}, \qquad \mathcal{I}_2 = \mathcal{I}_1^c.$$

Here, $\mathcal{I}_2$ roughly correspond to the "bad" set of trajectories where there exists some timestep $h$ such that $\bar{\sigma}_{n,h}^2 > \max\{\sigma_{n,h}^2, H\}$, and $\mathcal{I}_1$ to be the "good" set of trajectories where the monotonic variance estimator is controlled.

We need to bound the cardinality of the latter before employing our truncation argument on the estimated variances. As we note that for all $n \in \mathcal{I}_2$ we have that $\max_{h \in [H]} \sqrt{\phi_{n,h}^\top \Sigma_{n,h}^{-1} \phi_{n,h}} \geqslant 1/(4d^6 H^2)$, which indicates that

$$\sum_{h=1}^{H} \min\left\{1, 16d^{12} H^4 \phi_{n,h}^\top \Sigma_{n,h}^{-1} \phi_{n,h}\right\} \geqslant 1,$$

and so we can conclude that

$$|\mathcal{I}_2| \leqslant \sum_{h=1}^{H} \sum_{n=1}^{N_{\text{off}}} \min\left\{1, 16d^{12} H^4 \phi_{n,h}^\top \Sigma_{n,h}^{-1} \phi_{n,h}\right\} \lesssim d^{13} H^5 \log(1 + N/d),$$

by Lemma D.5 of Zhou and Gu (2022) and the fact that $\|\phi_{n,h}/\bar{\sigma}_{n,h}\|^2 \leqslant 1/H^2$. As we require in Theorem 2 that $N_{\text{on}}, N_{\text{off}} = \tilde{\Omega}(d^{13} H^{14})$, we come to the following result

$$|\mathcal{I}_2|/N_{\text{off}} \lesssim 8d^{13} H^5 \log(1 + N/d)/N_{\text{off}} = \tilde{o}(1), \quad |\mathcal{I}_1|/N_{\text{off}} = 1 - \tilde{o}(1).$$

Informally, this means that the proportion of trajectories in the "bad set" $\mathcal{I}_2$ is asymptotically zero, and the proportion in the "good set" $\mathcal{I}_1$ is asymptotically one. As for every $n \in \mathcal{I}_1$ we have that for any $h \in [H]$,

$$\max_{\phi_h \in \Phi_{\text{off}}} \left( \phi_h^\top \bar{\Sigma}_{\text{off},h}^{-1} \phi_h \right)$$
$$= \max_{\phi_h \in \Phi_{\text{off}}} \left( \phi_h^\top \left( \frac{1}{N_{\text{off}}} \left( \sum_{n=1}^{N_{\text{off}}} \bar{\sigma}_{n,h}^{-2} \phi_{n,h} \phi_{n,h}^\top + \lambda \mathbf{I} \right) \right)^{-1} \phi_h \right)$$
$$= \max_{\phi_h \in \Phi_{\text{off}}} N_{\text{off}} \left( \phi_h^\top \left( \sum_{n=1}^{N_{\text{off}}} \bar{\sigma}_{n,h}^{-2} \phi_{n,h} \phi_{n,h}^\top + \lambda \mathbf{I} \right)^{-1} \phi_h \right)$$
$$\leqslant \max_{\phi_h \in \Phi_{\text{off}}} N_{\text{off}} \left( \phi_h^\top \left( \sum_{n \in \mathcal{I}_1} \frac{\phi_{n,h} \phi_{n,h}^\top}{\sigma_{n,h}^2 + H} + \lambda \mathbf{I} \right)^{-1} \phi_h \right).$$

Now we invoke the total variance lemma. Recall that in Section B of the appendix in He et al. (2023), by the total variance lemma of Jin et al. (2019), if $N_{\text{off}} \geqslant \tilde{\Omega}(d^{10.5} H^{14})$, it holds that

$$\frac{1}{N_{\text{off}}} \sum_{n=1}^{N_{\text{off}}} \sum_{h=1}^{H} \sigma_{n,h}^2 = \tilde{O}\left( H^2 + d^{10.5} H^{16}/N_{\text{off}} \right) = \tilde{O}\left( H^2 \right).$$

With a direct application of Lemma 17, as we set $T = \tilde{O}(H)$ and $\gamma = c_{\text{off}}(\mathcal{X}_{\text{off}})/N_{\text{off}}$, we will then get to

$$\sum_{h=1}^{H} \max_{\phi_h \in \Phi_{\text{off}}} \sqrt{\phi_h^\top \Sigma_{\text{off},h}^{-1} \phi_h} \lesssim \frac{c_{\text{off}}(\mathcal{X}_{\text{off}}) H}{N_{\text{off}}} \sqrt{N_{\text{off}} H} = \sqrt{\frac{c_{\text{off}}(\mathcal{X}_{\text{off}}) H^3}{N_{\text{off}}}},$$

which indicates that

$$\sum_{h=1}^{H} \max_{\phi_h \in \Phi_{\text{off}}} \sqrt{\phi_h^\top \bar{\Sigma}_{\text{off},h}^{-1} \phi_h} \lesssim \sqrt{c_{\text{off}}(\mathcal{X}_{\text{off}})^2 H^3}.$$

$\square$

Now, from Lemmas 12 and 13, for any partition $\mathcal{X}_{\text{off}}, \mathcal{X}_{\text{on}}$, the offline regret satisfies

$$\text{Reg}_{\text{off}}(T) \lesssim \sum_{h=1}^{H} \sqrt{dN_{\text{on}} \frac{N_{\text{on}}}{N_{\text{off}}} \max_{\phi_h \in \Phi_{\text{off}}} \phi_h^\top \bar{\Sigma}_{\text{off},h}^{-1} \phi_h} \lesssim \sqrt{c_{\text{off}}(\mathcal{X}_{\text{off}})^2 dH^3 N_{\text{on}} \frac{N_{\text{on}}}{N_{\text{off}}}}.$$

## E.5 Online regret control

We will then bound the online term, $\mathrm{Reg}_{\mathrm{on}}(T)$. He et al. (2023) show in Lemma E.1 that it is possible to use Cauchy-Schwarz to bound this by

$$\mathrm{Reg}_{\mathrm{on}}(T) = \widetilde{O}\left(d^4 H^8 + \beta d^7 H^5 + \beta\sqrt{dHT + dH\sum_{t=1}^{T}\sum_{h=1}^{H}\sigma_{t,h}^2}\right),$$

and in Section B of the appendix, state that by the total variance lemma of Jin et al. (2019),

$$\sum_{t=1}^{T}\sum_{h=1}^{H}\sigma_{t,h}^2 \leqslant \widetilde{O}\left(H^2 T + d^{10.5}H^{16}\right)$$

We will seek to use the online partition to tighten the dimensional dependence in the first result accordingly.

**Lemma 14** (Modified Lemma E.1 in He et al. (2023))**.** *For any parameters $\beta' \geqslant 1$ and $C \geqslant 1$, and any partition $\mathcal{X}_{\mathrm{off}}, \mathcal{X}_{\mathrm{on}}$, the summation of bonuses on the online partition is upper bounded by*

$$\sum_{t=1}^{T}\min\left(\beta'\sqrt{\phi\left(s_h^{(t)}, a_h^{(t)}\right)^{\top}\mathbf{\Sigma}_{t,h}^{-1}\phi\left(s_h^{(t)}, a_h^{(t)}\right)}\mathbb{1}_{\mathcal{X}_{\mathrm{on}}}, C\right)$$

$$\leqslant 4d^4 H^6 C\iota + 10\beta' d_{\mathrm{on}}^5 H^4 \iota + 2\beta'\sqrt{2d_{\mathrm{on}}\iota\sum_{t=1}^{T}\left(\sigma_{t,h}^2 + H\right)}$$

*where $\iota = \log(1 + N/(d\lambda))$.*

*Proof.* For each horizon $h \in [H]$, we first note that the summation can be bounded by the sum of two terms, where the first term is tight-bounded and the second term stands for a tail event where $\phi^{T}\Sigma^{-1}\phi$ gets large.

$$\sum_{t=1}^{T}\min\left(\beta'\sqrt{\phi\left(s_h^{(t)}, a_h^{(t)}\right)^{\top}\mathbf{\Sigma}_{t,h}^{-1}\phi\left(s_h^{(t)}, a_h^{(t)}\right)}\mathbb{1}_{\mathcal{X}_{\mathrm{on}}}, C\right)$$

$$\leqslant \sum_{t=1}^{T}\beta'\min\left(\sqrt{\phi\left(s_h^{(t)}, a_h^{(t)}\right)^{\top}\mathbf{\Sigma}_{t,h}^{-1}\phi\left(s_h^{(t)}, a_h^{(t)}\right)}\mathbb{1}_{\mathcal{X}_{\mathrm{on}}}, 1\right)$$

$$+ C\sum_{t=1}^{T}\mathbb{1}\left\{\sqrt{\phi\left(s_h^{(t)}, a_h^{(t)}\right)^{\top}\mathbf{\Sigma}_{t,h}^{-1}\phi\left(s_h^{(t)}, a_h^{(t)}\right)}\mathbb{1}_{\mathcal{X}_{\mathrm{on}}} \geqslant 1\right\}.$$

We first bound $\sum_{t=1}^{T}\beta'\min\left(\sqrt{\phi\left(s_h^{(t)}, a_h^{(t)}\right)^{\top}\mathbf{\Sigma}_{t,h}^{-1}\phi\left(s_h^{(t)}, a_h^{(t)}\right)}\mathbb{1}_{\mathcal{X}_{\mathrm{on}}}, 1\right)$, using a variant of Lemma B.1 from Zhou and Gu (2022) in Lemma 18. With this, we have that

$$\sum_{t=1}^{T}\beta'\min\left(\sqrt{\phi\left(s_h^{(t)}, a_h^{(t)}\right)^{\top}\mathbf{\Sigma}_{t,h}^{-1}\phi\left(s_h^{(t)}, a_h^{(t)}\right)}\mathbb{1}_{\mathcal{X}_{\mathrm{on}}}, 1\right)$$

$$\leqslant \sum_{t=1}^{T}\beta'\min\left(\sqrt{\phi\left(s_h^{(t)}, a_h^{(t)}\right)^{\top}\left(\sum_{n=1}^{N_{\mathrm{off}}+t}(\phi_{n,h}\mathbb{1}_{\mathcal{X}_{\mathrm{on}}})(\phi_{n,h}\mathbb{1}_{\mathcal{X}_{\mathrm{on}}})^{\top} + \lambda\mathbf{I}_d\right)^{-1}\phi\left(s_h^{(t)}, a_h^{(t)}\right)}\mathbb{1}_{\mathcal{X}_{\mathrm{on}}}, 1\right)$$

$$\leqslant 10\beta' d_{\mathrm{on}}^5 H^4 \iota + 2\beta'\sqrt{2d_{\mathrm{on}}\iota\sum_{k=1}^{K}\left(\sigma_{k,h}^2 + H\right)},$$

where $\iota = \log(1 + N/(d\lambda))$.

From this, it suffices to follow the rest of the proof of Lemma E.1 from He et al. (2023) to bound the remaining term by

$$\sum_{t=1}^{T} \mathbb{1}\left\{\sqrt{\phi\left(s_h^{(t)}, a_h^{(t)}\right)^\top \mathbf{\Sigma}_{t,h}^{-1} \phi\left(s_h^{(t)}, a_h^{(t)}\right)} \mathbb{1}_{\mathcal{X}_{\mathrm{on}}} \geqslant 1\right\} \leqslant 4d^4 H^6 C\iota.$$

□

As a result, we obtain the following bound for the online regret

$$\mathrm{Reg}_{\mathrm{on}}(T) \lesssim d^7 H^9 + \beta\sqrt{d_{\mathrm{on}} d H^3 T}.$$

### E.6 Putting everything together

Combining our results in E.4 and E.5, we come to the bound of total regret that

$$\mathrm{Reg}(N_{\mathrm{on}}) \lesssim \sqrt{H^3 N_{\mathrm{on}} \log(H/\delta)} + \sqrt{c_{\mathrm{off}}(\mathcal{X}_{\mathrm{off}})^2 d H^3 N_{\mathrm{on}} \frac{N_{\mathrm{on}}}{N_{\mathrm{off}}}} + \sqrt{d_{\mathrm{on}} d H^3 N_{\mathrm{on}}} + d^7 H^9.$$

When we set $N_{\mathrm{on}}, N_{\mathrm{off}} = \tilde{\Omega}(d^{13} H^{14})$ and choose $\mathcal{X}_{\mathrm{off}}, \mathcal{X}_{\mathrm{on}}$ be the partition that minimize the right hand side, we have

$$\mathrm{Reg}(N_{\mathrm{on}}) \lesssim \inf_{\mathcal{X}_{\mathrm{off}}, \mathcal{X}_{\mathrm{on}}} \left(\sqrt{c_{\mathrm{off}}(\mathcal{X}_{\mathrm{off}})^2 d H^3 N_{\mathrm{on}} \frac{N_{\mathrm{on}}}{N_{\mathrm{off}}}} + \sqrt{d_{\mathrm{on}} d H^3 N_{\mathrm{on}}}\right),$$

proving Theorem 2.

## F OPTCOV from Wagenmaker and Jamieson (2023)

We lean on the OPTCOV algorithm from Wagenmaker and Pacchiano (2023) for reward-agnostic exploration , first proposed in Wagenmaker and Jamieson (2023), as well as the Frank-Wolfe subroutine used, for completeness.

---

**Algorithm 5** Collection of Optimal Covariates (OPTCOV), Wagenmaker and Pacchiano (2023)

---

1: **Input:** functions to optimize $(f_i)_i$, constraint tolerance $\epsilon$, confidence $\delta$.
2: **for** $i = 1, 2, 3, ...$ **do**
3:     Set the number of iterates $T_i \leftarrow 2^i$, episodes per iterate $K_i \leftarrow 2^i$.
4:     Play any policy for $K_i$ episodes to collect covariates $\mathbf{\Gamma}_0$ and data $\mathfrak{D}_0$.
5:     Initialize covariance matrix $\mathbf{\Lambda}_1 \leftarrow \mathbf{\Gamma}_0 / K$.
6:     **for** $t = 1, ..., T_i$ **do**
7:         Run FORCE (Wagenmaker et al., 2022) or another regret-minimizing algorithm on the exploration-focused synthetic reward $g_h^{(t)}(s,a) \propto \mathrm{tr}(-\nabla_{\mathbf{\Lambda}} f_i(\mathbf{\Lambda})|_{\mathbf{\Lambda}=\mathbf{\Lambda}_t \phi(s,a)\phi(s,a)^\top})$.
8:         Collect covariates $\mathbf{\Gamma}_t$, data $\mathfrak{D}_t$.
9:         Perform Frank-Wolfe update: $\mathbf{\Gamma}_{t+1} \leftarrow (1 - \frac{1}{t+1})\mathbf{\Lambda}_t + \frac{1}{t+1}\mathbf{\Gamma}_t / K_i$.
10:     **end for**
11:     Assign $\widehat{\mathbf{\Lambda}}_i \leftarrow \mathbf{\Lambda}_{T_i+1}, \mathfrak{D}_i \leftarrow \cup_{t=0}^{T_i}\mathfrak{D}_t$.
12:     **if** $f_i(\widehat{\mathbf{\Lambda}}_i) \leqslant K_i T_i \epsilon$ **then**
13:         **Return:** $\widehat{\Lambda}, K_i T_i, \mathfrak{D}_i$.
14:     **end if**
15: **end for**

---

The algorithm essentially performs the doubling trick to determine how many samples to collect, terminating when the minimum eigenvalue of the covariance matrix is above the set tolerance.

Wagenmaker and Pacchiano (2023) then prove the following guarantee for OPTCOV in the hybrid setting:

**Lemma 15** (Termination of OPTCOV, Lemma C.2 (Wagenmaker and Pacchiano, 2023))**.** *Let*

$$f_i(\mathbf{\Lambda}) = \frac{1}{\eta_i} \log \left( \sum_{\phi \in \Phi} e^{\eta_i \|\phi\|_{\mathbf{A}_i(\mathbf{\Lambda})^{-1}}^2} \right), \quad \mathbf{A}_i(\mathbf{\Lambda}) = \mathbf{\Lambda} + \frac{1}{T_i K_i} \mathbf{\Lambda}_{0,i} + \frac{1}{T_i K_i} \mathbf{\Lambda}_{\text{off}}$$

*for some $\mathbf{\Lambda}_{0,i}$ satisfying $\mathbf{\Lambda}_{0,i} \geq \mathbf{\Lambda}_0$ for all $i$, and $\eta_i = 2^{2i/5}$. Let $(\beta_i, M_i)$ denote the smoothness and magnitude constants for $f_i$. Let $(\beta, M)$ be some values such that $\beta_i \leq \eta_i \beta$, $M_i \leq M$ for all $i$. Then, if we run OPTCOV on $(f_i)_i$ with constraint tolerance $\epsilon$ and confidence $\delta$, we have that with probability at least $1 - \delta$, it will run for at most*

$$\max \left\{ \min_N 16\mathbf{N} \quad s.t. \quad \inf_{\mathbf{\Lambda} \in \mathbf{\Omega}} \max_{\phi \in \Phi} \phi^\top (N\mathbf{\Lambda} + \mathbf{\Lambda}_0 + \mathbf{\Lambda}_{\text{off}})^{-1} \phi \leq \frac{\epsilon}{6}, \right.$$
$$\left. \frac{\text{poly}(\beta, d, H, M, \log 1/\delta)}{\epsilon^{4/5}} \right\}.$$

*episodes, and will return data $\{\phi_\tau\}_{\tau=1}^N$ with covariance $\widehat{\mathbf{\Sigma}}_N = \sum_{\tau=1}^N \phi_\tau \phi_\tau^\top$ such that*

$$f_{\hat{i}} \left( N^{-1} \widehat{\mathbf{\Sigma}}_N \right) \leq N\epsilon$$

*where $\hat{i}$ is the iteration on which OPTCOV terminates.*

We use this to obtain a modified guarantee for OPTCOV that does not require a call to the CONDI-TIONEDCOV algorithm of Wagenmaker and Jamieson (2023).

**Lemma 16** (Modified Bound on OPTCOV, Theorem 4, Wagenmaker and Pacchiano (2023))**.** *Consider running OPTCOV with some $\epsilon_{\text{exp}} > 0$ and functions $f_i$ as defined in Lemma 15, instantiated with the regularization $\bar{\lambda} \geq 0$. Then with probability $1 - \delta$, this procedure will collect at most*

$$\max \left\{ \min_N C \cdot N \ s.t. \ \inf_{\mathbf{\Lambda} \in \mathbf{\Omega}} \max_{\phi \in \Phi} \phi^\top \left( N(\mathbf{\Lambda} + \bar{\lambda}I) + \mathbf{\Lambda}_{\text{off}} \right)^{-1} \phi \leq \frac{\epsilon_{\text{exp}}}{6}, \frac{\text{poly}(d, H, c_{\text{on}}(\mathcal{X}_{\text{on}}), \log 1/\delta)}{\epsilon_{\text{exp}}^{4/5}} \right\}$$

*episodes, and will produce covariates $\widehat{\mathbf{\Sigma}}$ such that*

$$\max_{\phi_h \in \Phi} \phi_h \left( \widehat{\mathbf{\Sigma}} + \bar{\lambda}I + \mathbf{\Lambda}_{\text{off}} \right)^{-1} \phi_h \leq \epsilon_{\text{exp}}.$$

*Proof.* This is essentially the proof of Theorem 4 in Wagenmaker and Pacchiano (2023), except where we chase around a few terms that differ in the analysis. By Lemma D.5 of Wagenmaker and Jamieson (2023), it suffices to bound the smoothness constants of $f_i(\mathbf{\Lambda})$ by

$$L_i = \frac{1}{\bar{\lambda}^2}, \quad \beta_i = \frac{2}{\bar{\lambda}^3} \left( 1 + \frac{\eta_i}{\bar{\lambda}} \right), \quad M_i = \frac{1}{\bar{\lambda}^2}.$$

Assume that the termination condition of OPTCOV is met for $\hat{i}$ satisfying

$$\hat{i} \leq \log \left( \text{poly} \left( \frac{1}{\epsilon_{\text{exp}}}, d, H, \log 1/\delta, c_{\text{on}}(\mathcal{X}_{\text{on}}), \bar{\lambda} \right) \right).$$

We assume this holds and justify it at the conclusion of the proof. For notational convenience, define

$$\iota := \text{poly} \left( \log \frac{1}{\epsilon_{\text{exp}}}, d, H, \log 1/\delta, c_{\text{on}}(\mathcal{X}_{\text{on}}), \bar{\lambda} \right).$$

Given this upper bound on $\hat{i}$, set

$$L = M := \frac{1}{\bar{\lambda}^2}, \quad \beta := \iota.$$

With this choice of $L, M, \beta$, we have $L_i \leq L$, $M_i \leq M$, $\beta_i \leq \eta_i \beta$ for all $i \leq \hat{i}$.

Now apply Lemma 15 with $\mathbf{\Lambda}_0 = \bar{\lambda} \cdot \mathbf{I}$ and get that, with probability at least $1 - \delta$, OPTCOV terminates after at most

$$\max \left\{ \min_N 16N \quad \text{s.t.} \quad \inf_{\mathbf{\Lambda} \in \mathbf{\Omega}} \max_{\boldsymbol{\phi} \in \Phi} \boldsymbol{\phi}^\top \left( N\mathbf{\Lambda} + \bar{\lambda} \cdot I + \mathbf{\Lambda}_{\text{off}} \right)^{-1} \boldsymbol{\phi} \leqslant \frac{\epsilon_{\text{exp}}}{6} \right.$$

$$\left. \frac{\text{poly}\left( d, H, \underline{\lambda}, c_{\text{off}}(\mathcal{X}_{\text{off}}), \log 1/\epsilon_{\text{exp}}, \log 1/\delta \right)}{\epsilon_{\text{exp}}^{4/5}} \right\}$$

episodes, and returns data $\{\boldsymbol{\phi}_\tau\}_{\tau=1}^N$ with covariance $\widehat{\mathbf{\Sigma}} = \sum_{\tau=1}^N \boldsymbol{\phi}_\tau \boldsymbol{\phi}_\tau^\top$ such that

$$f_{\hat{i}}\left( N^{-1}\widehat{\mathbf{\Sigma}} \right) \leqslant N\epsilon_{\text{exp}}$$

where $\hat{i}$ is the iteration on which OPTCOV terminates.

By Lemma D.1 of Wagenmaker and Jamieson (2023) we have

$$N \cdot \max_{\phi_h \in \Phi} \phi_h \left( \widehat{\mathbf{\Sigma}} + \mathbf{\Lambda}_{\hat{i},0} + \mathbf{\Lambda}_{\text{off}} \right)^{-1} \phi_h \leqslant f_{\hat{i}}\left( N^{-1}\widehat{\mathbf{\Sigma}} \right),$$

and the upper bound on the tolerance follows from Lemma D.8 of Wagenmaker and Jamieson (2023).

It remains to justify the bound on $\hat{i}$. We do so with the same argument that Wagenmaker and Pacchiano (2023) use. Note that by the definition of OPTCOV, if we run for a total of $\bar{N}$ episodes, we can bound $\hat{i} \leqslant \frac{1}{4}\log_2(\bar{N})$. However, we see that the bound on $\hat{i}$ given above upper bounds $\frac{1}{4}\log_2(\bar{N})$ for $\bar{N}$ the upper bound on the number of samples collected by OPTCOV stated above. Thus, the bound on $\hat{i}$ is valid. $\qquad\square$

## G  Miscellanous lemmas

**Lemma 17.** *Let $\Phi \subset \mathbb{R}^d$ be a linear subspace. Suppose $\{\phi_{h,n}\}_{h\in[H],n\in[N]} \in \Phi$ be a collection of unit vectors and $\{\sigma_{h,n}\}_{h\in[H],n\in[N]} \in \mathbb{R}_+$ be a collection of positive real numbers with mean $\bar{\sigma} = (NH)^{-1}\sum_{h,n}\sigma_{h,n}$. Suppose it holds that $\max_{h\in[H]}\max_{\phi_h\in\Phi}(\phi_h^T\Lambda_h^{-1}\phi_h) \leqslant \gamma$, then the following result satisfies*

$$\sum_{h=1}^H \max_{\phi_h\in\Phi} \sqrt{\phi_h^T\Sigma_h^{-1}\phi_h} \lesssim \gamma H\sqrt{N\bar{\sigma}},$$

*with*

$$\Lambda_h = \sum_{n=1}^N \phi_{h,n}\phi_{h,n}^T + \lambda I_d, \qquad \Sigma_h = \sum_{n=1}^N \frac{\phi_{h,n}\phi_{h,n}^T}{\sigma_{h,n}} + \lambda I_d.$$

*Proof.* First, we denote $\bar{\sigma}_h = N^{-1}\sum_n \sigma_{h,n}$. Informally, this implies that most individuals of $\sigma_{h,\cdot}$ is asymptotically on the order of $\bar{\sigma}_h$, with only a small amount of individuals being higher in order. To rule out the effect of the "large" ones, we group them into the following collection of sets:

$$\mathcal{E}_h(C_h) = \{n \in [N] : \sigma_{h,n} \geqslant C_h\bar{\sigma}_h\}.$$

Here, we leave the choice of the truncation level $C_h$ open for now, but note that we allow the truncation levels $C_h$ vary across different timesteps $h$ and related to $\bar{\sigma}_h$. It follows by definition that $\sum_{h=1}^H \bar{\sigma}_h = H\bar{\sigma}$. From an application of Markov's Inequality, the cardinality of set $\mathcal{E}_h(C_h)$ can be upper bounded as

$$|\mathcal{E}_h(C_h)| \leqslant \frac{N}{C_h}.$$

We now choose the truncation level $C_h$. To do so, we follow the steps below to quantify the effect induced by the trajectories with high variance (i.e. those that belong to $\mathcal{E}_h(C_h)$):

$$\min_{\phi_h\in\Phi} \phi_h^\top \Sigma_h^\star \phi_h \geqslant \min_{\phi_h\in\Phi} \phi_h^\top \left( \sum_{n=1}^N \frac{\phi_{h,n}\phi_{h,n}^T}{\sigma_{h,n}} \right) \phi_h$$

$$\geqslant \min_{\phi_h \in \Phi} \phi_h^\top \left( \sum_{n \in [N] \setminus \mathcal{E}_h(C_h)} \frac{\phi_{h,n} \phi_{h,n}^T}{\sigma_{h,n}} \right) \phi_h$$

$$\geqslant \frac{1}{C_h \bar{\sigma}_h} \min_{\phi_h \in \Phi} \phi_h^\top \left( \sum_{n \in [N] \setminus \mathcal{E}_h(C_h)} \phi_{h,n} \phi_{h,n}^T \right) \phi_h.$$

We now utilize a basic matrix inequality that for any matrix $A, B$, we have

$$\min_{\phi_h \in \Phi} \phi_h^\top A \phi_h \geqslant \min_{\phi_h \in \Phi} \phi_h^\top (A + B) \phi_h - \max_{\phi_h \in \Phi} \phi_h^\top B \phi_h,$$

which allows us to further bound $\min_{\phi_h \in \Phi} \phi_h^\top \Sigma_h^\star \phi_h$ as

$$\min_{\phi_h \in \Phi} \phi_h^\top \Sigma_h^\star \phi_h \geqslant \frac{1}{C_h \bar{\sigma}_h} \min_{\phi_h \in \Phi} \phi_h^\top \left( \sum_{n=1}^N \phi_{h,n} \phi_{h,n}^T + \lambda I_d \right) \phi_h$$

$$- \frac{1}{C_h \bar{\sigma}_h} \max_{\phi_h \in \Phi} \phi_h^\top \left( \sum_{n \in \mathcal{E}_h(C_h)} \phi_{h,n} \phi_{h,n}^T + \lambda I_d \right) \phi_h$$

$$\gtrsim \frac{1}{C_h \bar{\sigma}_h} \left( \gamma^{-1} - \frac{N}{C_h} - \lambda \right),$$

This leads to the following result:

$$\min_{\phi_h \in \Phi} \phi_h^\top \Lambda_h \phi_h = \min_{\phi_h \in \Phi} (\phi_h^\top \Lambda_h^{-1} \phi_h)^{-1} \gtrsim \left\{ \max \left( \frac{c_{\mathrm{off}}(\mathcal{X}_{\mathrm{off}})}{N_{\mathrm{off}}}, \frac{d_{\mathrm{on}}}{N_{\mathrm{on}}} \right) \right\}^{-1} = \gamma^{-1},$$

where the first equality holds because $\Lambda_h$ is a linear transformation on the subspace $\Phi$. Equivalently, this holds from the variational characterization of the eigenvalues and the fact that the largest absolute eigenvalue is equal to the inverse of the smallest absolute eigenvalue of the inverse. As a result, in order to rule out the effect of the "high variance trajectories", we select the truncation level $\delta_h$ such that $N/C_h = \Theta(\gamma^{-1})$, implying $C_h = \Theta(N\gamma)$. Hence, we obtain the following lower bound:

$$\min_{\phi_h \in \Phi} \phi_h^\top \Sigma_h^\star \phi_h \gtrsim \frac{1}{\gamma^2 N \bar{\sigma}_h}.$$

Finally, we note that

$$\sum_{h=1}^H \max_{\phi_h \in \Phi} \sqrt{\phi_h^\top \Sigma_h^{\star-1} \phi_h} = \sum_{h=1}^H \left( \min_{\phi_h \in \Phi} \sqrt{\phi_h^\top \Sigma_h^\star \phi_h} \right)^{-1} \lesssim \gamma \sqrt{N} \sum_{h=1}^H \sqrt{\bar{\sigma}_h} \leqslant \gamma H \sqrt{N \bar{\sigma}}.$$

$\square$

**Lemma 18** (Modified Lemma B.1 from Zhou and Gu (2022)). *Let $\mathcal{X}_{\mathrm{off}}, \mathcal{X}_{\mathrm{on}}$ be a partition of $\mathcal{S} \times \mathcal{A} \times [H]$, such that their images under the feature map, $\Phi_{\mathrm{off}}, \Phi_{\mathrm{on}}$ are subspaces of dimension $d_{\mathrm{off}}, d_{\mathrm{on}}$ respectively. Let $\{\sigma_k, \beta_k\}_{k \geqslant 1}$ be a sequence of non-negative numbers, $\alpha, \gamma > 0$, $\{\mathbf{x}_k\}_{k \geqslant 1} \subset \mathbb{R}^d$ and $\|\mathbf{x}_k\|_2 \leqslant L$. Let $\{\mathbf{Z}_k\}_{k \geqslant 1}$ and $\{\bar{\sigma}_k\}_{k \geqslant 1}$ be recursively defined as follows: $\mathbf{Z}_1 = \lambda \mathbf{I} + \mathbf{Z}_{\mathrm{off}}$ for some symmetric matrix $\mathbf{Z}_{\mathrm{off}}$, where $N = N_{\mathrm{off}} + K$, and we have*

$$\forall k \geqslant 1, \bar{\sigma}_k = \max \left\{ \sigma_k, \alpha, \gamma \|\mathbf{x}_k\|_{\mathbf{z}_k^{-1}}^{1/2} \right\}, \mathbf{Z}_{k+1} = \mathbf{Z}_k + \mathbb{1}_{\mathcal{X}_{\mathrm{on}}} \mathbf{x}_k \mathbf{x}_k^\top / \bar{\sigma}_k^2$$

*Let $\iota = \log \left( 1 + NL^2 / (d\lambda \alpha^2) \right)$. Then we have*

$$\sum_{k=1}^K \min \left\{ 1, \beta_k \|\mathbf{x}_k\|_{\mathbf{z}_k^{-1}} \mathbb{1}_{\mathcal{X}_{\mathrm{on}}} \right\} \leqslant 2 d_{\mathrm{on}} \iota + 2 \max_{k \in [K]} \beta_k \gamma^2 d_{\mathrm{on}} \iota + 2 \sqrt{d_{\mathrm{on}} \iota} \sqrt{\sum_{k=1}^K \beta_k^2 (\sigma_k^2 + \alpha^2)}.$$

*Proof.* The proof roughly follows that of Lemma B.1 in Zhou and Gu (2022), except that we have to make modifications as necessary to tighten the dimension dependence to $d_{\mathrm{on}}$ and incorporate the offline data.

Decompose the set $[K]$ into a union of two disjoint subsets $[K] = \mathcal{I}_1 \cup \mathcal{I}_2$,

$$\mathcal{I}_1 = \left\{ k \in [K] : \|\mathbf{x}_k/\bar{\sigma}_k\|_{\mathbf{Z}_k^{-1}} \mathbb{1}_{\mathcal{X}_{\mathrm{on}}} \geqslant 1 \right\}, \mathcal{I}_2 = [K]\backslash\mathcal{I}_1.$$

Then the following upper bound of $|\mathcal{I}_1|$ holds, where the projector $\mathcal{P}_{\mathrm{on}}$ onto $\Phi_{\mathrm{on}}$ has the decomposition $\mathcal{P}_{\mathrm{on}} = U_{\mathrm{on}} U_{\mathrm{on}}^\top$ by the thin SVD, and we write $\mathbf{u}_k = U_{\mathrm{on}}^\top \mathbf{x}_k$:

$$|\mathcal{I}_1| = \sum_{k \in \mathcal{I}_1} \min\left\{1, \|\mathbf{x}_k/\bar{\sigma}_k\|_{\mathbf{Z}_k^{-1}}^2 \mathbb{1}_{\mathcal{X}_{\mathrm{on}}}\right\}$$

$$\leqslant \sum_{k=1}^{K} \min\left\{1, \|\mathbf{x}_k/\bar{\sigma}_k\|_{\mathbf{Z}_k^{-1}}^2 \mathbb{1}_{\mathcal{X}_{\mathrm{on}}}\right\}$$

$$\leqslant \sum_{k=1}^{K} \min\left\{1, \bar{\sigma}_k^{-2}\mathbf{x}_k^\top \mathbf{Z}_k^{-1}\mathbf{x}_k \mathbb{1}_{\mathcal{X}_{\mathrm{on}}}\right\}$$

$$= \sum_{k=1}^{K} \min\left\{1, (U_{\mathrm{on}}U_{\mathrm{on}}^\top\mathbf{x}_k)^\top \mathbf{Z}_k^{-1}(U_{\mathrm{on}}U_{\mathrm{on}}^\top\mathbf{x}_k) \mathbb{1}_{\mathcal{X}_{\mathrm{on}}}\right\}$$

$$= \sum_{k=1}^{K} \min\left\{1, \mathbf{x}_k^\top U_{\mathrm{on}}U_{\mathrm{on}}^\top \mathbf{Z}_k^{-1} U_{\mathrm{on}}U_{\mathrm{on}}^\top\mathbf{x}_k \mathbb{1}_{\mathcal{X}_{\mathrm{on}}}\right\}$$

$$= \sum_{k=1}^{K} \min\left\{1, \mathbf{x}_k^\top U_{\mathrm{on}}U_{\mathrm{on}}^\top \left(\sum_{n=1}^{k} \mathbb{1}_{\mathcal{X}_{\mathrm{on}}}\bar{\sigma}_n^{-2}\mathbf{x}_n\mathbf{x}_n^\top + \lambda\mathbf{I}_d\right)^{-1} U_{\mathrm{on}}U_{\mathrm{on}}^\top\mathbf{x}_k \mathbb{1}_{\mathcal{X}_{\mathrm{on}}}\right\}.$$

By Lemma 20, we can take the $U_{\mathrm{on}}$ inside the inverse and conclude that

$$\sum_{k=1}^{K} \min\left\{1, \mathbf{x}_k^\top U_{\mathrm{on}}U_{\mathrm{on}}^\top \left(\sum_{n=1}^{k} \mathbb{1}_{\mathcal{X}_{\mathrm{on}}}\bar{\sigma}_n^{-2}\mathbf{x}_n\mathbf{x}_n^\top + \lambda\mathbf{I}_d\right)^{-1} U_{\mathrm{on}}U_{\mathrm{on}}^\top\mathbf{x}_k \mathbb{1}_{\mathcal{X}_{\mathrm{on}}}\right\}$$

$$= \sum_{k=1}^{K} \min\left\{1, \mathbf{x}_k^\top U_{\mathrm{on}} \left(\sum_{n=1}^{k} \mathbb{1}_{\mathcal{X}_{\mathrm{on}}}\bar{\sigma}_n^{-2}U_{\mathrm{on}}^\top\mathbf{x}_n\mathbf{x}_n^\top U_{\mathrm{on}} + \lambda\mathbf{I}_{d_{\mathrm{on}}}\right)^{-1} U_{\mathrm{on}}^\top\mathbf{x}_k \mathbb{1}_{\mathcal{X}_{\mathrm{on}}}\right\}.$$

Intuitively, this is because all the $\mathbb{1}_{\mathcal{X}_{\mathrm{on}}}U_{\mathrm{on}}^\top\mathbf{x}_n$ and $\mathbb{1}_{\mathcal{X}_{\mathrm{on}}}\mathbf{x}_n$ are both in $\Phi_{\mathrm{on}}$, and in that case the projection is just the identity.

Writing $\mathbf{u}_n = U_{\mathrm{on}}^\top\mathbf{x}_n$, and invoking Lemma D.5 of Zhou and Gu (2022) (which is a restatement of Lemma 11 of Abbasi-yadkori et al. (2011)) and the fact that $\|\mathbf{x}_k/\bar{\sigma}_k\|_2 \leqslant L/\alpha$, it holds that

$$\sum_{k=1}^{K} \min\left\{1, \mathbf{x}_k^\top U_{\mathrm{on}} \left(\sum_{n=1}^{k} \mathbb{1}_{\mathcal{X}_{\mathrm{on}}}\bar{\sigma}^{-2}U_{\mathrm{on}}^\top\mathbf{x}_n\mathbf{x}_n^\top U_{\mathrm{on}} + \lambda\mathbf{I}_{d_{\mathrm{on}}}\right)^{-1} U_{\mathrm{on}}^\top\mathbf{x}_k \mathbb{1}_{\mathcal{X}_{\mathrm{on}}}\right\}$$

$$\sum_{k=1}^{K} \min\left\{1, \mathbf{u}_k^\top \left(\sum_{n=1}^{k} \mathbb{1}_{\mathcal{X}_{\mathrm{on}}}\bar{\sigma}^{-2}\mathbf{u}_n\mathbf{u}_n^\top + \lambda\mathbf{I}_{d_{\mathrm{on}}}\right)^{-1} \mathbf{u}_k \mathbb{1}_{\mathcal{X}_{\mathrm{on}}}\right\}$$

$$\leqslant 2d_{\mathrm{on}}\iota,$$

as desired, and conclude that $|\mathcal{I}_1| \leqslant 2d_{\mathrm{on}}\iota$.

The rest of the proof follows Zhou and Gu (2022) more closely. By the same argument that Zhou and Gu (2022) use,

$$\sum_{k \in [K]} \min\left\{1, \beta_k \|\mathbf{x}_k\|_{\mathbf{z}_k^{-1}} \mathbb{1}_{\mathcal{X}_{\mathrm{on}}}\right\} \leqslant 2d_{\mathrm{on}}\iota + \sum_{k \in \mathcal{I}_2} \beta_k\bar{\sigma}_k \|\mathbf{x}_k/\bar{\sigma}_k\|_{\mathbf{z}_k^{-1}} \mathbb{1}_{\mathcal{X}_{\mathrm{on}}}.$$

Decompose $\mathcal{I}_2 = \mathcal{J}_1 \cup \mathcal{J}_2$, where

$$\mathcal{J}_1 = \{k \in \mathcal{I}_2 : \bar{\sigma}_k = \sigma_k \cup \bar{\sigma}_k = \alpha\}, \mathcal{J}_2 = \left\{k \in \mathcal{I}_2 : \bar{\sigma}_k = \gamma\sqrt{\|\mathbf{x}_k\|_{\mathbf{z}_k^{-1}}}\mathbb{1}_{\mathcal{X}_{\mathrm{on}}}\right\}.$$

Similar to Zhou and Gu (2022),

$$\sum_{k \in \mathcal{J}_1} \beta_k \bar{\sigma}_k \left\| \mathbf{x}_k / \bar{\sigma}_k \right\|_{\mathbf{z}_k^{-1}} \mathbb{1}_{\mathcal{X}_{\text{on}}} \leqslant \sum_{k \in \mathcal{J}_1} \beta_k \left( \sigma_k + \alpha \right) \mathbb{1}_{\mathcal{X}_{\text{on}}} \min \left\{ 1, \left\| \mathbf{x}_k / \bar{\sigma}_k \right\|_{\mathbf{z}_k^{-1}} \mathbb{1}_{\mathcal{X}_{\text{on}}} \right\}$$

$$\leqslant \sum_{k=1}^{K} \beta_k \left( \sigma_k + \alpha \right) \min \left\{ 1, \left\| \mathbf{x}_k / \bar{\sigma}_k \right\|_{\mathbf{z}_k^{-1}} \mathbb{1}_{\mathcal{X}_{\text{on}}} \right\}$$

$$\leqslant \sqrt{2 \sum_{k=1}^{K} \left( \sigma_k^2 + \alpha^2 \right) \beta_k^2} \sqrt{\sum_{k=1}^{K} \min \left\{ 1, \left\| \mathbf{x}_k / \bar{\sigma}_k \right\|_{\mathbf{z}_k^{-1}} \mathbb{1}_{\mathcal{X}_{\text{on}}} \right\}^2}$$

$$\leqslant 2 \sqrt{\sum_{k=1}^{K} \beta_k^2 \left( \sigma_k^2 + \alpha^2 \right)} \sqrt{d_{\text{on}} \iota},$$

and as for $k \in \mathcal{J}_2$ we have that $\bar{\sigma}_k = \gamma^2 \left\| \mathbf{x}_k / \bar{\sigma}_k \right\|_{\mathbf{z}_k^{-1}} \mathbb{1}_{\mathcal{X}_{\text{on}}}$,

$$\sum_{k \in \mathcal{J}_2} \beta_k \bar{\sigma}_k \left\| \mathbf{x}_k / \bar{\sigma}_k \right\|_{\mathbf{z}_k^{-1}} \mathbb{1}_{\mathcal{X}_{\text{on}}} = \gamma^2 \cdot \sum_{k \in \mathcal{J}_1} \beta_k \left\| \mathbf{x}_k / \bar{\sigma}_k \right\|_{\mathbf{z}_k^{-1}}^2 \mathbb{1}_{\mathcal{X}_{\text{on}}}$$

$$= \gamma^2 \cdot \sum_{k=1}^{K} \beta_k \min \left\{ 1, \left\| \mathbf{x}_k / \bar{\sigma}_k \right\|_{\mathbf{z}_k^{-1}}^2 \mathbb{1}_{\mathcal{X}_{\text{on}}} \right\} \leqslant 2 \max_{k \in [K]} \beta_k \gamma^2 d_{\text{on}} \iota.$$

Therefore,

$$\sum_{k=1}^{K} \min \left\{ 1, \beta_k \left\| \mathbf{x}_k \right\|_{\mathbf{z}_k^{-1}} \mathbb{1}_{\mathcal{X}_{\text{on}}} \right\} \leqslant 2 d_{\text{on}} \iota + 2 \max_{k \in [K]} \beta_k \gamma^2 d_{\text{on}} \iota + 2 \sqrt{d_{\text{on}} \iota} \sqrt{\sum_{k=1}^{K} \beta_k^2 \left( \sigma_k^2 + \alpha^2 \right)}.$$

$\square$

**Lemma 19** (Modified Version of Theorem 4.3, Zhou and Gu (2022))**.** *Let $\{\mathcal{G}_n\}_{n=1}^{N}$ be a filtration, and $\{\mathbf{x}_n, \eta_n\}_{n=1}^{N}$ be a stochastic process such that $\mathbf{x}_n \in \mathbb{R}^d$ is $\mathcal{G}_n$-measurable and $\eta_n \in \mathbb{R}$ is $\mathcal{G}_{n+1}$-measurable. Let $L, \sigma, \lambda, \epsilon > 0, \boldsymbol{\mu}^* \in \mathbb{R}^d$. Arrange the datapoints from the offline and online samples as follows, $1, ..., N_{\text{off}}, N_{\text{off}} + 1, ..., N_{\text{off}} + N_{\text{on}}$. For $n = 1, ..., N$, let $y_n = \langle \boldsymbol{\mu}^*, \mathbf{x}_n \rangle + \eta_n$ and suppose that $\eta_n, \mathbf{x}_n$ also satisfy*

$$\mathbb{E} \left[ \eta_n \mid \mathcal{G}_n \right] = 0, \mathbb{E} \left[ \eta_n^2 \mid \mathcal{G}_n \right] \leqslant \sigma^2, |\eta_n| \leqslant R, \left\| \mathbf{x}_n \right\|_2 \leqslant L.$$

*For $n = 1, ..., N$, let $\mathbf{Z}_n = \lambda \mathbf{I} + \sum_{i=1}^{n} \mathbf{x}_i \mathbf{x}_i^\top, \mathbf{b}_n = \sum_{i=1}^{n} y_i \mathbf{x}_i, \boldsymbol{\mu}_n = \mathbf{Z}_n^{-1} \mathbf{b}_n$, and*

$$\beta_n = 12 \sqrt{\sigma^2 d \log \left( 1 + n L^2 / (d\lambda) \right) \log \left( 32 (\log(R/\epsilon) + 1) n^2 / \delta \right)}$$
$$+ 24 \log \left( 32 (\log(R/\epsilon) + 1) n^2 / \delta \right) \max_{1 \leqslant i \leqslant n} \left\{ |\eta_i| \min \left\{ 1, \left\| \mathbf{x}_i \right\|_{\mathbf{z}_{i-1}^{-1}} \right\} \right\}$$
$$+ 6 \log \left( 32 (\log(R/\epsilon) + 1) n^2 / \delta \right) \epsilon.$$

*Then, for any $0 < \delta < 1$, we have with probability at least $1 - \delta$ that,*

$$\forall n = 1, ..., N, \left\| \sum_{i=1}^{n} \mathbf{x}_i \eta_i \right\|_{\mathbf{z}_n^{-1}} \leqslant \beta_n, \left\| \boldsymbol{\mu}_n - \boldsymbol{\mu}^* \right\|_{\mathbf{z}_n} \leqslant \beta_n + \sqrt{\lambda} \left\| \boldsymbol{\mu}^* \right\|_2$$

*Proof.* The proof is merely a small wrapper over Theorem 4.3 of Zhou and Gu (2022), where we adapt this to our setting in the same way that Tan and Xu (2024) do in Lemma 1 of their paper. That is, we pre-append the offline data to the online data, and generate the $\mathbf{Z_n}, \mathbf{b_n}, \mu_\mathbf{n}, \beta_\mathbf{n}$ accordingly.

As in Lemma 1 of Tan and Xu (2024), let $N = N_{\text{off}} + N_{\text{on}}$. Order the $N_{\text{off}}$ offline episodes arbitrarily, to form episodes $1, ..., N_{\text{off}}$, and then begin the online episodes from episode $N_{\text{off}} + 1, ..., N$. Then, we can directly apply Theorem 4.3 of Zhou and Gu (2022) to recover the desired result. $\square$

**Lemma 20.** *Suppose that $W = \mathbb{R}^m$ and $V = \mathbb{R}^n$, where $n < m$. Let $\boldsymbol{U} : W \mapsto V$ be a linear transformation and that $S = (\boldsymbol{U}^\top \boldsymbol{U})W$. As $\boldsymbol{v}, \boldsymbol{v}_1, \ldots, \boldsymbol{v}_n \in S$, we have*

$$\boldsymbol{v}^\top \boldsymbol{U}^\top \boldsymbol{U} \Big( \sum_{j=1}^k \boldsymbol{v}_i \boldsymbol{v}_i^\top + \lambda \boldsymbol{I}_m \Big)^{-1} \boldsymbol{U}^\top \boldsymbol{U} \boldsymbol{v} = \boldsymbol{v}^\top \boldsymbol{U}^\top \Big( \sum_{j=1}^k \boldsymbol{U} \boldsymbol{v}_i \boldsymbol{v}_i^\top \boldsymbol{U}^\top + \lambda \boldsymbol{I}_n \Big)^{-1} \boldsymbol{U} \boldsymbol{v}$$

*Proof.* For projection matrix $\boldsymbol{U}$, there exists orthogonal matrix $\boldsymbol{Q} \in \mathbb{R}^{m \times m}$ and diagonal matrix $\boldsymbol{D} = (\boldsymbol{I}_n, \boldsymbol{0}_{n \times (m-n)})$ such that $\boldsymbol{U} = \boldsymbol{D}\boldsymbol{Q}$. We further define $\boldsymbol{u} = \boldsymbol{U}\boldsymbol{v}$, $\tilde{\boldsymbol{v}} = \boldsymbol{Q}\boldsymbol{v}$, $\boldsymbol{u}_i = \boldsymbol{U}\boldsymbol{v}_i$ and $\tilde{\boldsymbol{v}}_i = \boldsymbol{Q}\boldsymbol{v}_i$ for $1 \leqslant i \leqslant n$. Then, we note that as $\boldsymbol{v} \in S$, we have $\boldsymbol{v} = \boldsymbol{U}^\top \boldsymbol{U} \boldsymbol{v} = \boldsymbol{Q}^\top \boldsymbol{\Lambda} \boldsymbol{Q} \boldsymbol{v}$, where $\boldsymbol{\Lambda} = \mathrm{diag}(\boldsymbol{I}_n, \boldsymbol{0}_{m-n})$, which is equivalent to $\tilde{\boldsymbol{v}} = \boldsymbol{\Lambda}\tilde{\boldsymbol{v}}$. As a result, we may conclude that $\tilde{\boldsymbol{v}}^\top = (\boldsymbol{u}^\top, \boldsymbol{0}_{m-n})$.

Therefore, with a direct calculation, one will see that

$$\Big( \sum_{j=1}^k \boldsymbol{v}_i \boldsymbol{v}_i^\top + \lambda \boldsymbol{I}_m \Big)^{-1} = \Big( \sum_{j=1}^k \boldsymbol{Q}^\top \tilde{\boldsymbol{v}}_i \tilde{\boldsymbol{v}}_i^\top \boldsymbol{Q} + \lambda \boldsymbol{I}_m \Big)^{-1}$$

$$= \boldsymbol{Q}^\top \Big( \sum_{j=1}^k \tilde{\boldsymbol{v}}_i \tilde{\boldsymbol{v}}_i^\top + \lambda \boldsymbol{I}_m \Big)^{-1} \boldsymbol{Q}$$

$$= \boldsymbol{Q}^\top \begin{pmatrix} \sum_{i=1}^k \boldsymbol{u}_i \boldsymbol{u}_i^\top + \lambda \boldsymbol{I}_n & \boldsymbol{0} \\ \boldsymbol{0} & \lambda \boldsymbol{I}_{m-n} \end{pmatrix}^{-1} \boldsymbol{Q}$$

$$= \boldsymbol{Q}^\top \begin{pmatrix} \big( \sum_{j=1}^k \boldsymbol{u}_i \boldsymbol{u}_i^\top + \lambda \boldsymbol{I}_n \big)^{-1} & \boldsymbol{0} \\ \boldsymbol{0} & \lambda^{-1} \boldsymbol{I}_{m-n} \end{pmatrix} \boldsymbol{Q}.$$

This will establish our desired conclusion

$$\text{LHS} = \boldsymbol{v}^\top \Big( \sum_{j=1}^k \boldsymbol{v}_i \boldsymbol{v}_i^\top + \lambda \boldsymbol{I}_m \Big)^{-1} \boldsymbol{v} = \tilde{\boldsymbol{v}}^\top \begin{pmatrix} \big( \sum_{j=1}^k \boldsymbol{u}_i \boldsymbol{u}_i^\top + \lambda \boldsymbol{I}_n \big)^{-1} & \boldsymbol{0} \\ \boldsymbol{0} & \lambda^{-1} \boldsymbol{I}_{m-n} \end{pmatrix} \tilde{\boldsymbol{v}}$$

$$= \boldsymbol{u}^\top \Big( \sum_{j=1}^k \boldsymbol{u}_i \boldsymbol{u}_i^\top + \lambda \boldsymbol{I}_n \Big)^{-1} \boldsymbol{u} = \text{RHS}.$$

$\square$

## H  Further details on the numerical experiments

The environment, as in Tan and Xu (2024), is a 6-piece wide Tetris board with pieces no larger than $2 \times 2$, where the action space consists of four actions, differentiated by the degree of rotation in 90 degree intervals and the reward is given by penalizing any increases in the height of the stack from a tolerance of 2 blocks. As in Tan and Xu (2024), we generate feature vectors by projecting the 640-dimensional one-hot state-action encoding onto a 60-dimensional subspace spanned by the top 60 eigenvectors of the feature covariance matrix under 200 trajectories from the uniform behavior policy.

All experiments were run on a single computer with an Intel i9-13900k CPU, 128 GB of RAM, and a NVIDIA RTX3090 GPU, in no more than a couple of hours.

