# OpenReview forum: "Hybrid Reinforcement Learning Breaks Sample Size Barriers In Linear MDPs"
_NeurIPS.cc/2024/Conference — NeurIPS 2024 poster_

### Official Review · Reviewer_ujdT · 2024-07-12

**Soundness:** 2
**Presentation:** 3
**Contribution:** 2
**Rating:** 6
**Confidence:** 3

**Summary:**

This paper studies hybrid RL in linear MDPs, aiming to address the problem of whether hybrid RL can improve upon the existing lower bounds established in purely offline and purely online settings, without relying on the single-policy concentrability assumption. By combining offline dataset with online interaction, authors introduce computationally efficient algorithms which achieve sharper error bound and regret bound in offline and online settings.

**Strengths:**

Existing theoretical works of hybrid RL under function approximation mainly focus on PAC guarantees which are not tight and require stringent concentrability assumptions on the quality of the behavior policy. Motivated by the question of whether hybrid RL is useful (i.e. improving upon the existing lower bounds established in purely offline and purely online RL without relying on the single-policy concentrability assumption, which is raised by Xie et al. (2022b)), authors provide computationally efficient algorithms for both PAC and regret-minimizing RL for linear MDPs, without relying on single-policy concentrability.


In particular, this work exhibits several interesting findings:
1.  Algorithmically, two types of hybrid RL algorithms are introduced: an online-to-offline algorithm, which involves reward-agnostic online exploration followed by a pessimistic offline algorithm to learn an optimal policy; and additionally, an offline-to-online method that utilizes offline data to warm start an online algorithm.

2. Theoretically, authors show both algorithms improve upon existing sample complexity, which is measured with respect to either the PAC framework or the regret.

This work demonstrates clear rationale and compelling motivation, while clearly articulates its main ideas throughout the draft. Thorough discussion of related works have been provided. It develops a better theoretical understanding of hybrid RL in linear MDPs, which potentially benefits future works for hybrid RL with functional approximation in this context.

**Weaknesses:**

Below are several potential improvements that authors are suggested to consider:

1. It is better to summarize and highlight the main technical novelties in the main text. There exist extensive theoretical studies in either linear MDPs or in hybrid RL. It is still not quite clear throughout the main text what the main technical challenges are in achieving the optimal policy in the studied context and what the technical novelties are in improving existing theoretical results. It seems to me that the proofs closely follow Xiong et al. (2023) in offline RL,  Wagenmaker and Pacchiano (2023) in reward-agnostic exploration and He et al. (2023) in linear MDPs. As a result, it is important to point out how the studied settings can be technically challenging, and what novel arguments are developed compared to existing works.

**Questions:**

1. Could you highlight the main technical techniques / reason why Alg. 2 can achieve better regret compared to Tan and Xu, 2024 and Amortila et al., 2024 for the linear MDP case?

2. In the studied setting, if we consider general function approximation or a more general class of MDPs with linear structure, do you envision the current analysis and results can be utilized to improve existing bounds?

3. Could you comment on the optimality of the provided bounds, whether dependence on the parameters involved can be further improved?

4. Could you provide an intuitive explanation of concentrability coefficient? And what is the range that indicates good concentrability? In line 135, does $d^*$ represent the occupancy measure of the optimal policy?

**Limitations:**

This is a theoretical work, no negative social impact.

---

> ### Author Rebuttal · Authors · 2024-08-06
>
> We thank the reviewer for their helpful comments. We are glad that you find our findings interesting, and further thank the reviewer for their kind words that our motivation is compelling, our rationale is clear, and that our work has potential downstream impact. We address the reviewer’s questions and concerns below. If you think the responses adequately address your concerns, we will be glad if you choose to increase your score.
>
> **Response to Weaknesses**
>
> **W1: It is better to summarize and highlight the main technical novelties in the main text.**
>
> A1: We thank the reviewer for raising this concern! We agree with the reviewer in that it is better to include this in the main text, and will do so in an updated version. Regarding the second half of the reviewer’s comment, although we use well known existing algorithms with intuitive modifications, obtaining regret guarantees in our paper is not trivial if we just simply follow the previous arguments. We highlight several technical innovations in our paper within the global rebuttal, and also elaborate further in our answer to Question 1 below.
>
> **Responses to Questions**
>
> **Q1: Could you highlight the main technical techniques / reason why Alg. 2 can achieve better regret compared to Tan and Xu, 2024 and Amortila et al., 2024 for the linear MDP case?**
>
> A1: There are two main reasons why we achieve better regret with Algorithm 2.
> 1. The first is in our choice of online learning algorithm – we use LSVI-UCB++, which possesses a minimax-optimal $\sqrt{d^2H^3N_{on}}$ online-only regret bound. In contrast, the GOLF algorithm that Tan and Xu (2024) use incurs an extra factor of H^2 due to the use of the squared Bellman error and lack of variance-weighting. Amortila et al. (2024) require a weight function class, incurring an extra dH dependence, in total having $\sqrt{d^3H^6N_{on}}$ regret.
> 2. The second is in our careful analysis in the proof of Theorem 2. The truncation argument in Lemma 13 allows us to incur only a H^3 dependence in the offline portion of the regret bound, while we sharpen the dimensional dependence in the online part of the bound by proving a sharper variant of Lemma B.1 from Zhou and Gu (2022) in Lemma 16, using this in Lemma 14 to reduce the dimensional dependence in the summation of bonuses enough to achieve the desired $d_{on}dH^3N_{on}$ term. Without the above two techniques, one could have used a simpler analysis to achieve a much looser $\sqrt{c_{off}(X_{off})^2 d^6 H^8 N_{on}^2/N_{off}} + \sqrt{d^2H^3}$ regret bound by using the maximum magnitude of the variance weights for the offline partition and the analysis from He et al. verbatim for the online partition. Doing so would have still yielded an improvement over Tan and Xu (2024) and Amortila et al. (2024), but would not have yielded the same improvement that we managed to achieve.
>
> **Q2: In the studied setting, if we consider general function approximation or a more general class of MDPs with linear structure, do you envision the current analysis and results can be utilized to improve existing bounds?**
>
> A2: Absolutely! While some of the fine-grained techniques used in the analysis may not translate to the general function approximation setting, we believe that many of the general techniques will still be very applicable. For instance, one could use a similar analysis in spirit to our analysis of Algorithm 2 in order to attain better regret bounds than that found by Tan and Xu (2024) by e.g. analyzing the OLIVE algorithm of Du et al. (2021) and sharpening dependencies whenever possible while decomposing the error onto the offline and online partitions.
>
> The analysis for a more general class of MDPs with linear structure is far more straightforward. We believe that for the case of linear mixture MDPs, for instance, one could use a similar analysis to Theorem 2 on the UCRL-VTR+ algorithm by Zhou et al. (2020) to achieve a similar result.
>
> **Q3: Could you comment on the optimality of the provided bounds, whether dependence on the parameters involved can be further improved?**
>
> A3: The error bound for Algorithm 1 is no worse than the offline-only minimax rate, as our result in equation 6 is no worse than the (nearly) minimax rate achieved by Xiong et al. (2023) that depends on the expected features under the optimal policy. Our guess is that it could maybe be sharpened to be as low as a $O(c^*_{off}(\mathcal{X}) d H^3)$ rate previously mentioned, but it is unclear to us how to achieve this even in the context of purely offline RL. To our knowledge, there is no $d^xH^y$-style minimax rate for offline RL in linear MDPs available in the literature that would inform us as to whether that is possible.
>
> Regarding the regret bound for Algorithm 2, the online $d_{on}dH^3$ portion of the regret bound is as good as it gets. It may be possible to shave off a factor of $c_{off}(X_{off})$ from the offline $c_{off}(\mathcal{X}_{off})^2dH^3$ portion, but it is unclear to us how that can be done. Despite that, we emphasize that the existing regret bound for Algorithm 2 is already no worse than the online-only minimax regret, and as such we already show provable gains over online-only learning in Theorem 2.
>
> **Q4: Could you provide an intuitive explanation of concentrability coefficient? And what is the range that indicates good concentrability? In line 135, does $d^{\star}$ represent the occupancy measure of the optimal policy?**
>
> A4: One can think of the concentrability coefficient as an “inflation factor’’ on the number of offline samples collected. $N_{off}/c_{off}(\mathcal{X}_{off})$, or the number of offline samples divided by the concentrability coefficient, may be thought of as an ``effective sample size’’. If the concentrability coefficient is low, think 1-10, that is as good as it gets – each offline sample is approximately as good as an online sample.
>
> $d^{\star}$ does indeed represent the occupancy measure of the optimal policy.

---

> ### Author Response · Authors · 2024-08-11
> **Rebuttal follow up**
>
> Dear Reviewer ujdT,
>
> Again, we thank you for your effort in reviewing our paper and for your helpful comments! We have carefully considered your questions and addressed them in our response. We would like to know whether our response has appropriately addressed your questions and concerns about our paper. If we have fully addressed your concerns, we would appreciate it if you considered increasing your score for our paper. Please let me know if you have further comments or concerns about our paper. Thanks!

---

> > ### Comment · Reviewer_ujdT · 2024-08-12
> >
> > I thank the authors for their detailed response.
> >
> > The studied settings are indeed interesting, and I do think the results of this paper will inspire future works in this direction. The current draft will absolutely benefit from incorporating the responses and clarification provided during rebuttal, which do contain a lot of useful information that was not captured in the initial draft.
> >
> > As a result, authors are suggested to carefully revise the draft and address the concerns raised by all reviewers. I am happy to raise my score for support, and look forward to your revision!

---

### Official Review · Reviewer_stbS · 2024-07-13

**Soundness:** 3
**Presentation:** 3
**Contribution:** 2
**Rating:** 6
**Confidence:** 3

**Summary:**

This paper studies the hybrid reinforcement learning problem in the linear MDP setting. It provides two algorithms (one focused on improving the offline error and the other on improving the online error) with theoretical analysis on their sample complexity. Though the algorithms are not optimal in terms of sample complexity with respect to some problem parameters, it appears to be the state-of-the-art in the literature for the linear setting.

**Strengths:**

The paper is clearly written; the problem formulation and assumptions are clearly stated. The theoretical results seem sound.

**Weaknesses:**

Aside from the suboptimality in $d$ and $H$ already pointed out by the authors, another weakness is that Algorithm 2 has a pretty large minimum requirement for the size of the offline dataset.

**Questions:**

It would be nice to discuss how Algorithm 1 compares with Algorithm 2, aside from the difference in approach (e.g., why isn’t one clearly better than the other).

Why is $|A|$ in Line 303? If that is not a typo, then it seems to be another weakness of Algorithm 2, since the action space might not be discrete.

---

> ### Author Rebuttal · Authors · 2024-08-06
>
> We thank the reviewer for their comments and questions. We are glad that the reviewer confirms the soundness of our results and that our result is state-of-the-art for the hybrid linear MDP setting. We answer your questions below.
>
> **Response to Weaknesses and Questions**
>
> **Q1: Aside from the suboptimality in d and H already pointed out by the authors, another weakness is that Algorithm 2 has a pretty large minimum requirement for the size of the offline dataset.**
>
> A1: We agree that the minimum requirement for the size of the offline dataset in the analysis is large. This issue is linked to the high burn-in cost that LSVI-UCB++ requires in general. This is hard to improve, as it arises from the use of the total variance lemma in the analysis (in various places). However, we believe that this issue pales in comparison to the similarly large requirement on online episodes that the burn-in cost imposes, as in many situations offline data is cheaper and more plentiful than online data (motivating the setting of hybrid RL in general). Fortunately, this issue seems to be limited to the analysis and does not necessarily occur in practice – within the new numerical simulations attached to the pdf in the global response, we find that with only 200 episodes of offline data, Algorithm 2 achieves a much-improved regret compared to the online-only LSVI-UCB++ within the Tetris environment of Tan and Xu (2024). Experimentally, this also extends to as few as 50 episodes, though we do not report those results for brevity.
>
> **Q2: It would be nice to discuss how Algorithm 1 compares with Algorithm 2, aside from the difference in approach (e.g., why isn’t one clearly better than the other).**
>
> A2: We thank the reviewer for raising this good question! Actually, Algorithm 1 and Algorithm 2 both enjoy their own advantages and also suffer from their own drawbacks. We will discuss them in detail here.
> - Algorithm 1 is an online-to-offline algorithm where the final policy is given by an offline RL algorithm, which is deterministic, and this allows the algorithm to be deployed in some critical regimes that randomization is not allowed. Besides, as Algorithm 1 performs reward agnostic exploration, we are able to use the combined dataset to learn tasks with different reward functions.
> - Algorithm 2 is an offline-to-online algorithm which is fully unaware of partition, so we don’t need to estimate $d_{\text{on}}$ at the beginning before performing the algorithm. The algorithm provides a regret guarantee, which can be deployed in some specific scenarios where minimizing the regret is important.
> - As for why neither algorithm achieves a much better error bound compared to the other, we bring the reviewer’s attention to the tabular case to illustrate. Li et al. (2023) use the online-to-offline approach to achieve an error of $\inf_{\sigma \in [0,1]} O(\sqrt{H^3SA\min(\sigma H, 1)} + \sqrt{H^3SC^*(\sigma)})$, while the result of Tan and Xu (2024) and our analysis suggests that one could possibly use a similar approach to our Algorithm 2 to achieve a regret bound not too far from $O(\sqrt{H^3Sc_{\text{off}}(X_{\text{off}}) N_{\text{on}}^2/N_{\text{off}}} + \sqrt{H^3SAN_{\text{on}}})$ by modifying the algorithm of Azar et al. (2017) to accommodate the use of offline data. As such, one can achieve sample size gains over purely offline and purely online learning with either approach. It is therefore our opinion that neither approach is conclusively better with regard to sample efficiency than the other, and considerations like those stated in the previous two bullet points should govern the deployment of either instead.
>
> **Q3: Why is |A| in Line 303? If that is not a typo, then it seems to be another weakness of Algorithm 2, since the action space might not be discrete.**
>
> A3: As the reviewer notes correctly, that is the cardinality of the action space. However, this does not correspond to a weakness of Algorithm 2. The action space need not be discrete, as one can similarly search over all possible actions with a continuous action space with e.g. projected gradient descent or a random search. We simply performed the computational complexity analysis, with results in Line 303, with a discrete action space for simplicity, in line with that of He et al. (2023). However, we agree that this was not stated as clearly as it perhaps should have been, and apologize for the confusion.

---

> ### Author Response · Authors · 2024-08-11
> **Rebuttal follow up**
>
> Dear Reviewer stbS,
>
> Again, we thank you for your effort in reviewing our paper and for your helpful comments! We have carefully considered your questions and addressed them in our response. We would like to know whether our response has appropriately addressed your questions and concerns about our paper. If we have fully addressed your concerns, we would appreciate it if you considered increasing your score for our paper. Please let me know if you have further comments or concerns about our paper. Thanks!

---

### Official Review · Reviewer_6qEJ · 2024-07-13

**Soundness:** 3
**Presentation:** 3
**Contribution:** 2
**Rating:** 6
**Confidence:** 3

**Summary:**

In this work, the authors develop sample and computationally efficient hybrid RL algorithms that are provably better than online-only and offline-only algorithms for linear MDPs. Without relying on the single-policy concentrability assumption, the authors take both online-to-offline and offline-to-online approaches to achieve no worse than optimal sample complexity, regardless of the quality of the behavior policy.

**Strengths:**

This work demonstrates a thorough discussion of all relevant literature, while the results are extensively compared with existing methods.

This work provides no worse than optimal sample complexity for two types of hybrid RL methods, which is non-trivial.

The presentation is generally clear, and the appendix is well-organized. The proof appears sound from a quick skim.

**Weaknesses:**

The assumption regarding Full Rank Projected Covariates seems to implicitly impose some constraints.

No experimentation or code is included with the work, making it difficult to examine whether the proposed algorithms are efficient in practice.

**Questions:**

Q: I am not familiar with Assumption 2: Full Rank Projected Covariates. Is it a common assumption? How practical is it in downstream applications?

Q: HYRULE seems to be a straightforward generalization of existing algorithms. Could you please list the challenges in proving its regret guarantee?

Q: It would be helpful if there is an experimental plan to verify the algorithms. A simple toy experiment plan should suffice.

Minor suggestion:

As mentioned in line 241, OPTCOV requires tolerance parameter. Should this parameter also listed in the input of RAPPEL (Algorithm 1)?

**Limitations:**

Same as weakness. This work focuses on the setting of linear MDPs, where the techniques may not be generalizable to other types of function approximations.

---

> ### Author Rebuttal · Authors · 2024-08-06
>
> We thank the reviewer for providing helpful comments to our paper! We also thank the reviewer for believing that our contribution is non-trivial, and for confirming the soundness of the proofs. We have revised our paper based on your suggestions, including new numerical experiments. If you think our responses adequately address your concerns, we will be glad if you choose to increase your score.
>
> **Response to Weaknesses and Questions**
>
> **Q1: I am not familiar with Assumption 2: Full Rank Projected Covariates. Is it a common assumption? How practical is it in downstream applications?**
>
> A1: This assumption is inherited from Wagenmaker and Jamieson (2022) and Wagenmaker and Pacchiano (2023). Wagenmaker and Jamieson (2022) stated that this is analogous to other explorability assumptions in the literature (Zanette et al. (2020), Hao et al. (2021), and Agarwal et al. (2021)). It essentially requires that there is some policy that collects covariates that span the entire feature space. In practice, this is achievable for any linear MDP via a transformation of the features that amounts to a projection onto the eigenspace corresponding to the nonzero singular values. For example, this is performed for the numerical simulations attached in the pdf file for the global response – as in Tan et al. (2024), the feature vectors are generated by projecting the 640-dimensional one-hot state-action encoding onto a 60-dimensional subspace spanned by the top 60 eigenvectors of the covariance matrix of the offline dataset.
>
> **Q2: HYRULE seems to be a straightforward generalization of existing algorithms. Could you please list the challenges in proving its regret guarantee?**
>
> A2: We agree that HYRULE is a straightforward generalization of LSVI-UCB++ in He et. al (2023), with $\Sigma_0$ initialized with the offline dataset. However, to achieve the regret guarantee in Theorem 2, we had to decompose the regret into the regret on the offline and online partitions. In the process, we faced the following challenges:
> 1. Bounding the regret on the offline partition was challenging, as we were not able to utilize the technique that was used in He et. al (2023). Instead, we bounded the regret with the maximum eigenvalue of $\Sigma_{off,h}^{-1}$. To maintain a H^3 dependence on the offline partition, we had to use a truncation argument in Lemma 13 that we also deployed in proving the regret guarantee of Algorithm 1 (RAPPEL).
> 2. Bounding the regret on the online partition allowed us to use an analysis that was close to that of He et. al (2023). However, directly following the argument of He et al. (2023) would have left us with a d^2H^3 dependence in Theorem 2. To reduce the dimensional dependence to d_{on}dH^3, we prove a sharper variant of Lemma B.1 from Zhou and Gu (2022) in Lemma 16, using this in Lemma 14 to reduce the dimensional dependence in the summation of bonuses enough to achieve the desired result.
> Without the above two techniques, one could have used a simpler analysis to achieve a far looser $\sqrt{c_{off}(X_{off})^2d^6H^8 N_{on}^2/N_{off}} + \sqrt{d^2H^3}$ regret bound by using the maximum magnitude of the variance weights for the offline partition and the analysis from He et al. verbatim for the online partition, but this would not have yielded the same improvement.
>
> **Q3: It would be helpful if there is an experimental plan to verify the algorithms. A simple toy experiment plan should suffice.**
>
> A3: We thank the reviewer for this helpful suggestion! We have added numerical simulations and report our results in the pdf file attached. The global response contains a more detailed summary of our findings, but in brief we show that:
> 1. Although reward-agnostic hybrid exploration with the uniform behavior policy achieves the best coverage throughout as expected, even reward-agnostic hybrid exploration with adversarially collected offline data achieves better coverage than online-only exploration. (Fig. 1)
> 2. When learning from a dataset collected by reward-agnostic exploration as in Algorithm 1, hybrid exploration outperforms offline-only learning and online-only reward-agnostic exploration when the behavior policy is adversarial. (Fig. 2)
> 3. Initializing a regret-minimizing online RL algorithm (LSVI-UCB++) with offline data from a uniform behavior policy as in Algorithm 2 yields lower regret than LSVI-UCB++ without an offline dataset. This shows that even a nearly minimax-optimal online learning algorithm can stand to benefit from being initialized with offline data. (Fig. 3)
>
> **Q4: As mentioned in line 241, OPTCOV requires tolerance parameter. Should this parameter also be listed in the input of RAPPEL (Algorithm 1)?**
>
> A4: We thank the reviewer for this suggestion! We agree that the parameter should be listed in the input of RAPPEL, and have made the change.
>
> **Q5: This work focuses on the setting of linear MDPs, where the techniques may not be generalizable to other types of function approximations.**
>
> A5: We agree that some of the specific techniques used may not translate to the general function approximation setting. That said, many of the general techniques and methods will still be very applicable. For instance, the offline-to-online approach is studied by Tan and Xu (2024) in the general function approximation setting, and one could use a similar analysis in spirit to our analysis of Algorithm 2 in order to attain better regret bounds than that found by Tan and Xu (2024) by e.g. analyzing the OLIVE algorithm of Du et al. (2021), sharpening dependencies whenever possible while decomposing the error onto the offline and online partitions. Another example lies in the online-to-offline approach in the context of general function approximation, where using a reward-agnostic exploration algorithm like RF-OLIVE or RF-GOLF, followed by a pessimistic offline algorithm like A-CRAB from Zhu et al. (2023) is one potential way forward that uses a similar analysis to our analysis of Algorithm 1.

---

> > ### Comment · Reviewer_6qEJ · 2024-08-11
> >
> > Thank authors for the careful responses and interesting additional experiments.  The clarification of the assumptions sounds reasonable and consistent with the experiment design. Given the thoroughness of the rebuttal, I have no further questions and am pleased to increase the score.

---

> ### Author Response · Authors · 2024-08-11
> **Rebuttal follow up**
>
> Dear Reviewer 6qEJ,
>
> Again, we thank you for your effort in reviewing our paper and for your helpful comments! We have carefully considered your questions and addressed them in our response. We would like to know whether our response has appropriately addressed your questions and concerns about our paper. If we have fully addressed your concerns, we would appreciate it if you considered increasing your score for our paper. Please let me know if you have further comments or concerns about our paper. Thanks!

---

### Official Review · Reviewer_SSHJ · 2024-07-13

**Soundness:** 3
**Presentation:** 3
**Contribution:** 3
**Rating:** 7
**Confidence:** 3

**Summary:**

The paper presents studies Hybrid Reinforcement Learning for linear MDPs, where Hybrid RL addresses the limitations of purely offline and online methods by combining offline data and online exploration. The paper introduces two specific algorithms: Reward-Agnostic Pessimistic PAC Exploration-initialized Learning (RAPPEL) and Hybrid Regression for Upper-Confidence Reinforcement Learning (HYRULE). RAPPEL is a online-to-offline approach where one first perform reward agnostic exploration to increase the coverage of the existing data and then perform offline RL method, and HYRULE performs offline-to-online method by starting LSVI_UCB++ with the offline data. Both methods shows improvement over preivous methods, and no worse than offline minimax and online minimax rate, respectively.

**Strengths:**

1. The paper is well-organized, and the literature review is thorough.

2. The proposed method clearly improves on the previous hybrid RL method for linear MDPs, and the online part of the rate indeed of Alg.2 matches the online minimax rate of linear mdps. Overall the improvement of the result is solid.

3. The improvement of the result leverages existing algorithms that are (relatively) well-known as subroutines with intuitive modifications (plus sharper analysis at certain part), which are easy to relate to the literature and intuitive.

**Weaknesses:**

1. There is no clear new techniques introduced in the paper.

2. The offline rates match the minimax rate only up to the coverage term - the minimax coveage is the single policy coverage, while the proposed algorithm and analysis depends on the all policy coverage.

**Questions:**

see above

---

> ### Author Rebuttal · Authors · 2024-08-06
>
> We thank the reviewer for providing helpful comments to our paper! We are glad that you believe our paper is well-organized with a thorough literature review and that our methods are a solid improvement upon previous algorithms in the literature. We have revised our paper based on the suggestions from the reviewers. Please feel free to check on our inclusion of numerical experiments and comments.
>
> **Response to Weaknesses and Questions**
>
> **Q1: There is no clear new techniques introduced in the paper.**
>
> A1: Thanks for raising this question! In fact, there are several technical innovations in our theoretical analysis compared to previous literature. To summarize:
> - Our results are accomplished by decomposing the error onto the offline and online partitions, sharpening the dimensional dependence to $d_{\text{on}}$ and $c_{off}(\mathcal{X}_{\text{off}})$ via projections onto those partitions.
>   - The former is accomplished by Kiefer-Wolfowitz in Algorithm 1, and by proving a sharper variant of Lemma B.1 from Zhou and Gu (2022) in Lemma 16, using this in Lemma 14 to reduce the dimensional dependence in the summation of bonuses enough to achieve the desired result.
> - We maintain a $H^3$ dependence for the error or regret for Algorithms 1 and 2, which is non-trivial.
>   - We accomplish this in Algorithm 1 and for the offline partition in Algorithm 2 by combining the total variance lemma and a novel truncation argument that rules out “bad” trajectories, which allows us to maintain a desirable $H^3$ dependence on both partitions for both algorithms.
> - Algorithmically, the reviewer notes that we use well-known existing algorithms with intuitive modifications to show that one can use common methods for RL in linear MDPs to achieve state-of-the-art sample complexity in the hybrid setting. Despite that, this is the first work to explore the online-to-offline approach in linear MDPs, and although algorithmic novelty is not our key focus, we believe that our work possesses some degree of algorithmic novelty as a result.
>
> **Q2: The offline rates match the minimax rate only up to the coverage term - the minimax coverage is the single policy coverage, while the proposed algorithm and analysis depends on the all policy coverage.**
>
> A2: We agree that a result depending on the partial single-policy concentrability coefficient would have been desirable, and in fact this is addressed in the discussion section of Tan and Xu (2024). That said, it is not entirely clear to us that a partial single-policy concentrability coefficient would be significantly better than a partial all-policy concentrability coefficient, particularly when we take the infimum over partitions in the way we and Tan and Xu (2024) do. This is because a good offline partition for the partial all-policy concentrability would correspond to the portion of the state-action space well-covered by the offline dataset, while the same for the partial single-policy concentrability would be well-covered by offline dataset and the optimal policy. The smaller size of the latter offline partition may be offset by the larger size of the latter’s online partition, and as such any gains may be limited in this analysis of the hybrid setting.
>
> We also note that our offline rates do indeed match the offline-only minimax rate – our result in equation 6 is no worse than the (nearly) minimax rate achieved by Xiong et al. (2023) that depends on the expected features under the optimal policy. That said, our result falls short of a $c^*_{\text{off}}(\mathcal{X}_{\text{off}})dH^3$ rate. However, it is unclear whether that rate is even possible, and if it is, it is unclear to us how it can be achieved. To our knowledge, there is no $d^xH^y$-style minimax rate for offline RL in linear MDPs available in the literature, and solving that problem would be out of the scope of this paper.

---

> ### Author Response · Authors · 2024-08-11
> **Rebuttal follow up**
>
> Dear Reviewer SSHJ,
>
> Again, we thank you for your effort in reviewing our paper and for your helpful comments! We have carefully considered your questions and addressed them in our response. We would like to know whether our response has appropriately addressed your questions and concerns about our paper. If we have fully addressed your concerns, we would appreciate it if you considered increasing your score for our paper. Please let me know if you have further comments or concerns about our paper. Thanks!

---

> > ### Comment · Reviewer_SSHJ · 2024-08-13
> >
> > I appreciate the author's efforts in addressing my concerns and improving the submission. I would appreciate if the techincal contributions can be highlighted more in the revised version. On that regard I would like to increase my score.

---

### Author Rebuttal · Authors · 2024-08-06

We thank the reviewers for their comments and suggestions on our paper. We also thank the reviewers for their kind comments that our findings were interesting and our results were sound and nontrivial.

**Technical Contributions**

For the benefit of everyone, we provide a summary of our technical contributions.
1. Our results are accomplished by decomposing the error onto the offline and online partitions.
2. We sharpen the dimensional dependence from $d$ to $d_{\text{on}}$ and $c_{\text{off}}(\mathcal{X}_{\text{off}})$ via projections onto those partitions.
  a. The former is accomplished in Algorithm 1 by Kiefer-Wolfowitz, and in Algorithm 2 by  proving a sharper variant of Lemma B.1 from Zhou and Gu (2022) in Lemma 16, using this in Lemma 14 to reduce the dimensional dependence in the summation of bonuses enough to achieve the desired result.
3. We maintain a $H^3$ dependence for the error or regret for Algorithms 1 and 2, which is non-trivial.
  a. We accomplish this in Algorithm 1 and for the offline partition in Algorithm 2 by combining the total variance lemma and a novel truncation argument that rules out “bad” trajectories.
4. Algorithmically, we use well-known existing algorithms with intuitive modifications to show that one can use common methods for RL in linear MDPs to achieve state-of-the-art sample complexity in the hybrid setting. This is also the first work to explore the online-to-offline approach in linear MDPs.

**Numerical Experiments**

Upon the request of one reviewer, we also provide a series of numerical experiments, in order to demonstrate the benefits of hybrid RL in the offline-to-online and online-to-offline settings. We implement Algorithms 1 and 2 on the scaled-down Tetris environment from Tan et al. (2024). This is a $6$-piece wide Tetris board with pieces no larger than $2 \times 2$, where the action space consists of four actions, differentiated by the degree of rotation in 90 degree intervals and the reward is given by penalizing any increases in the height of the stack from a tolerance of $2$ blocks. The offline dataset consists of $200$ trajectories generated from a uniform behavior policy. As in Tan et al. (2024), the feature vectors are generated by projecting the $640$-dimensional one-hot state-action encoding onto a $60$-dimensional subspace spanned by the top $60$ eigenvectors of the covariance matrix of the offline dataset.
1. Figure 1 depicts the coverage (defined by $1/\lambda_{\min}(\Lambda), 1/\lambda_{d_{\text{off}}}(\Lambda_{\text{off}}), 1/\lambda_{d_{\text{on}}}(\Lambda_{\text{on}})$) achieved by the reward-agnostic exploration algorithm, OPTCOV, when initialized respectively with $200$ trajectories from (1) a uniform behavioral policy, (2) an adversarial behavior policy obtained by the negative of the weights of a fully-trained agent under Algorithm 1, and (3) no offline trajectories at all for fully online learning. It shows that although hybrid RL with the uniform behavior policy achieves the best coverage throughout as expected, even hybrid RL with adversarially collected offline data achieves better coverage than online-only exploration. This demonstrates the potential of hybrid RL as a tool for taking advantage of poor quality offline data.
2. In Figure 2, one can observe that hybrid RL demonstrates strong benefits in the online-to-offline setting when the behavior policy is of poor quality. When applying offline learning to the hybrid dataset of $200$ trajectories and $100$ online trajectories, $300$ trajectories of adversarially collected offline data, and $300$ trajectories of online data under reward-agnostic exploration, we see that the hybrid dataset is most conducive for learning. Additionally, without a warm-start from offline data, online-only reward-agnostic exploration performs worse than the adversarially collected offline data due to significant burn-in costs. Hybrid RL therefore, in this instance, performs better than both offline-only and online-only learning alone.
3. In Figure 3, we compare the performances of LSVI-UCB++ and Algorithm 2. It can be seen from the figure that initializing a regret-minimizing online algorithm (LSVI-UCB++, He et al. (2023)) with an offline dataset as in Algorithm 2 yields lower regret than the same algorithm without an offline dataset. This shows that even a nearly minimax-optimal online learning algorithm can stand to benefit from being initialized with offline data.

---

### Decision · Program_Chairs · 2024-09-25

**Decision:**

Accept (poster)

**Comment:**

This paper on hybrid (online/offline) RL for linear MDPs presents online-to-offline and offline-to-online algorithms with improved theoretical guarantees.

Although observing that the algorithmic contribution is limited, being based on modifications of existing algorithms, the reviewers agree that the theoretical contribution is significant and sound, and praise the thorough discussion of related works.

Some important points were clarified in the discussion phase. In particular, for the paper to be ready for publication, it is fundamental that the authors make the following additions to the camera-ready version:

- Better highlight the technical contributions, as done in responding to multiple reviewers

- Add the new experiments, which were requested and later judged valuable by one reviewer

In addition, I strongly recommend to further clarify the point on "partial single-policy" vs "partial all-policy" concentrability. In the current version of the manuscript, the claim that theoretical guarantees are obtained "without single-policy concentrability" could be misleading. In fact, it is not clear whether partial all-policy concentrability is always preferable to full *single*-policy concentrability, nor whether a partial single-policy concentrability result would yield a significant improvement. A discussion on this is needed, but something on the lines of the answer to Q2 of Reviewer SSHJ should suffice.

The meaning of the terms "partial" and "full" in the paper should also be clearly explained at the beginning of the manuscript, since in the offline RL literature "partial" is sometimes used as a synonym of "single-policy" and "full" as a synonym of "all-policy". Clearly, this could be confusing for some readers without a proper disclaimer.